# Assessment of actual evapotranspiration over a semi-arid heterogeneous land surface by means of coupled low resolution remote sensing data with energy balance model: comparison to extra Large Aperture Scintillometer measurements

Sameh Saadi[1,2], Gilles Boulet[1], Malik Bahir[1], Aurore Brut[1], Émilie Delogu[1], Pascal Fanise[1], Bernard Mougenot[1], Vincent Simonneaux[1], and Zohra Lili Chabaane[2]

[1]Centre d'Etudes Spatiales de la Biosphère, Université de Toulouse, CNRS, CNES, IRD, UPS, Toulouse, France

[2] Université de Carthage / Institut National Agronomique de Tunisie/ LR17AGR01-GREEN-TEAM, Tunis, Tunisie;

*Correspondence to*: Sameh Saadi (saadi_sameh@hotmail.fr)

**Abstract.**

In semi-arid areas, agricultural production is restricted by water availability; hence efficient agricultural water management is a major issue. The design of tools providing regional estimates of evapotranspiration (ET), one of the most relevant water balance fluxes, may help the sustainable management of water resources.

Remote sensing provides periodic data about actual vegetation temporal dynamics (through the Normalized Difference Vegetation Index NDVI) and water availability under water stress (through the surface temperature $T_{surf}$) which are crucial factors controlling ET.

In this study, spatially distributed estimates of ET (or its energy equivalent, the latent heat flux LE) in the Kairouan plain (Central Tunisia) were computed by applying the Soil Plant Atmosphere and Remote Sensing Evapotranspiration (SPARSE) model fed by low resolution remote sensing data (Terra and Aqua MODIS). The work goal was to assess the operational use of the SPARSE model and the accuracy of the modeled i) sensible heat flux (H) and ii) daily ET over a heterogeneous semi-arid landscape with a complex land cover (*i.e.* trees, winter cereals, summer vegetables).

SPARSE was run to compute instantaneous estimates of H and LE fluxes at the satellite overpass time. The good correspondence ($R^2$= 0.60 and 0.63 and RMSE=57.89 $Wm^{-2}$ and 53.85 $Wm^{-2}$; for Terra and Aqua, respectively) between instantaneous H estimates and large aperture scintillometer (XLAS) H measurements along a path length of 4 km over the study area showed that the SPARSE model presents satisfactory accuracy. Results showed that, despite the fairly large scatter, the instantaneous LE can be suitably estimated at large scale (RMSE=47.20 $Wm^{-2}$ and 43.20 $Wm^{-2}$; for Terra and Aqua, respectively and $R^2$= 0.55 for both satellites). Additionally, water stress was investigated by comparing modeled (SPARSE) and observed (XLAS) water stress values; we found that most points were located within a 0.2 confidence interval, thus the general tendencies are well reproduced. Even though extrapolation of instantaneous latent heat flux values to daily totals was less obvious, daily ET estimates are deemed acceptable.

KEYWORDS: Evapotranspiration, Remote sensing, SPARSE model, scintillometer, water stress.

## 1 Introduction

In water scarce regions, especially arid and semi-arid areas, the sustainable use of water by resource conservation as well as the use of appropriate technologies to do so is a priority for agriculture (Amri et al., 2014; Pereira et al., 2002).

Water use rationalization is needed especially for countries actually suffering from water scarcity, or for countries that probably would suffer from water restrictions according to climate change scenarios. Indeed, the Mediterranean region is one of the most prominent "hot spots" in future climate change projections (Giorgi and Lionello, 2008) due to an expected larger warming than the global average and to a pronounced increase in precipitation inter-annual variability. The major part of the southern Mediterranean countries, among others Tunisia, already suffer from water scarcity and show a growing water deficit, due to the combined effect of the water needs growth (soaring demography and irrigated areas extension), and the reduction of resources (temporary drought and/or climate change). This implies that closely monitoring the water budget components is a major issue (Oki and Kanae, 2006).

The estimation of evapotranspiration (ET) is of paramount importance since it represents the preponderant component of the terrestrial water balance; it is the second largest component after precipitation (Glenn et al., 2007); hence ET quantification is a key factor for scarce water resources management. Direct measurement of ET is only possible at local scale (single field) using the eddy covariance method for example; whereas, it is much more difficult at larger scales (irrigated perimeter or watershed) due to the complexity not only of the hydrological processes (Minacapilli et al., 2007) but also of the hydro-meteorological processes. Indeed, at landscape scale, surface heterogeneity influences regional and local climate, inducing for example cloudiness, precipitation and temperature patterns differences between areas of higher elevation (hills and mountains surrounding the Kairouan plain) and the plain downstream. Moreover, at these scales, land cover is usually heterogeneous and this affects the land-atmosphere exchanges of heat, water and other constituents (Giorgi and Avissar, 1997). ET estimates for various temporal and spatial scales, from hourly to monthly to seasonal time steps, and from field to global scales, are required for hydrologic applications in water resource management (Anderson et al., 2011). Techniques using remote sensing (RS) information are therefore essential when dealing with processes that cannot be represented by point measurements only (Su, 2002).

In fact, the contribution of RS in vegetation's physical characteristics monitoring on large areas have been identified for years (Tucker, 1978); RS provides periodic data about some major ET drivers, amongst others, surface temperature and vegetation properties (e.g. Normalized Difference Vegetation Index NDVI and Leaf Area Index LAI) from field to regional scales (Li et al., 2009; Mauser and Schädlich, 1998). Many methods using remotely-sensed data to estimate ET are reviewed in Courault et al. (2005). ICARE (Gentine et al., 2007) and SiSPAT (Braud et al., 1995) are examples of complex physically based Land Surface Models (LSM) using RS data. They include a detailed description of the vegetation water uptake in the root zone, the interactions between groundwater, root zone and surface water. However, the lateral surface and subsurface flows are neglected. This can lead to inaccurate results when applied in areas where such interactions are important (Overgaard et al., 2006).

Moreover, RS can provide estimates of large area fluxes in remote locations, but those estimates are based on the spatial and temporal scales of the measuring systems and thus vary one from another. Hence, one solution is to upscale local micrometeorological measurements to larger spatial scales in order to acquire an optimum representation of land-atmosphere interactions (Samain et al., 2012). However, such up-scaling process is not always possible and results might not be reliable in comparison to the RS distributed products.

Water and energy exchange in the soil-plant-atmosphere continuum have been simulated through several land surface models (Bastiaanssen et al., 2007; Feddes et al., 1978). Among them, two different approaches use remote sensing data to estimate spatially distributed ET (Minacapilli et al., 2009): one is based on the soil water balance (SWB) and one that solves the surface energy budget (SEB). The SWB approach exploits only visible-near-infrared (VIS-NIR) observations to perceive the spatial variability of crop parameters. The SEB modeling approach uses visible (VIS), near-infrared (NIR) and thermal (TIR) data to solve the SEB equation by forcing remotely-sensed estimates of the SEB components (mainly the surface temperature $T_{surf}$). In fact, there is a strong link between water availability in the soil and surface temperature under water stress, hence, in order to estimate soil moisture status as well as actual ET at relevant space and timescales, information in the TIR domain (8–14 µm) is frequently used (Boulet et al., 2007). The SWB approach has the advantage of high resolution and frequency VIS-NIR remote sensing data availability against limited availability of high resolution thermal imagery for the SEB approach. Indeed, satellite data such as Landsat or Advanced Spaceborne Thermal Emission and Reflection Radiometer (ASTER) provide field scale (30–100 m) estimates of ET (Allen et al., 2011), but they have a low temporal resolution (16 day-monthly) (Anderson et al., 2011).

The RS-based SWB models provide estimates of ET, soil water content, and irrigation requirements in a continuous way. For instance, at field scale, estimates of seasonal ET and irrigation can be obtained by SWB modeling using high resolution remote sensing forcing as done in the study with the SAtellite Monitoring of IRrigation (SAMIR) model by Saadi et al. (2015) over the Kairouan plain. However, for an appropriate estimation of ET, the SWB model requires knowledge of the water inputs (precipitation and irrigation) and an assessment of the extractable water from the soil (mostly derived from the soil moisture characteristics: actual available water content in the root zone, wilting point and field capacity), whereas, significant biases are found mainly when dealing with large areas and long periods, due to the spatial variability of the water inputs uncertainties as well as the inaccuracy in estimating other flux components such as the deep drainage (Calera et al., 2017). Hence, the major limitation of the SWB method is the high number of needed inputs whose estimation is highly uncertain especially over a heterogeneous land surface due to hydrologic processes complexity. Moreover, spatially distributed SWB models, typically those using the Food and Agriculture Organization-FAO guidelines (Allen et al., 1998) for crop ET estimation, generally parameterize the vegetation characteristics on the basis of land use maps (Bounoua et al., 2015; Xie et al., 2008), and different parameters are used for different land use classes. Nevertheless, SWB modelers generally do not have the possibility to carry out remote sensing-based land use change mapping due to time, budget, or capacity constraints and use often very generic classes potentially leading to modeling errors (Hunink et al., 2017). In addition, the lack of data about the soil properties (controlling field capacity, wilting point and the water retention) as well as the actual root depths, lead to limited practical use of the SWB models (Calera et al., 2017). The same apply to the soil evaporation whose estimation generally rely on the FAO guidelines approach (Allen et al., 1998). Although, it was shown that under high evaporation conditions, the FAO-56 (Allen et al., 1998) daily evaporation computed on the basis of the readily

evaporable water (REW) is overestimated at the beginning of the dry down phase (*i.e.* the period after rain or irrigation where the soil moisture is decreasing due to evapotranspiration and drainage, Mutziger et al., 2005; Torres and Calera, 2010). Hence, to improve its estimation a reduction factor proposed by Torres and Calera (2010) was applied to deal with this problem in several studies (e.g. Odi-Lara et al., 2016; Saadi et al., 2015). Furthermore, SWB models such as SWAP (Kroes, 2017), Cropsyst (Stöckle et al., 2003), AquaCrop (Steduto et al., 2009) and SAMIR (Simonneaux et al., 2009) are able to take irrigation into account, either as an estimated amount provided by the farmer (as an input if available) or a predicted amount through a module triggering irrigation according to, say, critical soil moisture levels (as an output). However, the limited knowledge of the actual irrigation scheduling is a critical limitation for the validation protocol of irrigation requirements estimates by SWB modeling. Therefore, SWB modelers must deal with the lack of information about real irrigation which induces unreliable estimations.

Consequently, ET estimation at regional scale is often achieved using SEB approaches, by combining surface temperature from medium to low resolution (kilometer scale) remote sensing data with vegetation parameters and meteorological variables (Liou and Kar, 2014). Recently, many efforts have been made to feed remotely sensed surface temperature into ET modeling platforms in combination with other critical variables, e.g., NDVI and albedo (Kalma et al., 2008; Kustas and Anderson, 2009). A wide range of satellite-based ET models were developed, and these methods are reviewed in (Liou and Kar, 2014). The majority of SEB-based models are single-source models; their algorithms compute a total latent heat flux as the sum of the evaporation and the transpiration components using a remotely sensed surface temperature. However, separate estimates of evaporation and transpiration makes the dual-source models more useful for agrohydrological applications (water stress detection, irrigation monitoring etc.) (Boulet et al., 2015).

Contrarily to SWB models, most SEB models are run in their most standardized version, using observed remote sensing-based parameters such as albedo in conjunction with a set of input parameters taken from literature or *in situ* data. On the other hand, the SEB model validation with enough data in space and time is difficult to achieve, due to the limited availability of high resolution thermal images (Chirouze et al., 2014). Therefore, it is usually possible to evaluate SEB models results only at similar scale (km) to medium or low resolution images. Indeed, the pixel size of thermal remote sensing images, except for the scarce Landsat7 images (60 m), covers a range of 1000 m (Moderate Sensors Resolution Imaging Spectroradiometer MODIS), to the order of 4000 m (Geostationary Operational Environmental Satellite GEOS) . However, direct methods measuring sensible heat fluxes (eddy covariance for example) only provide point measurements with a footprint considerably smaller than a satellite pixel. Therefore, scintillometry techniques have emerged as one of the best tools aiming to quantify averaged fluxes over heterogeneous land surfaces (Brunsell et al., 2011). They provide area-averaged sensible heat flux over areas comparable to those observed by satellites (Hemakumara et al., 2003; Lagouarde et al., 2002). Scintillometry can provide sensible heat using different wavelengths (optical wavelength ranges), aperture sizes (15-30 cm) and configurations (long-path and short-path scintillometry) (Meijninger et al., 2002). The upwind area contributing to the flux (*i.e.* the flux footprint) varies as wind direction and atmospheric stability, and must be estimated for the surface measurements in order to compare them to SEB estimates of the flux which are representative of the pixel (Brunsell et al., 2011). Assessing the upwind area contributing to the flux can be done using several footprint models (Schmid, 2002). Although footprint analysis ensures ad hoc spatial intersecting area between ground measurements and satellite-based surface fluxes, the spatial

heterogeneity at subpixel scale should be further considered in validating low resolution satellite data (Bai et al., 2015). The LAS technique has been validated over heterogeneous landscapes against eddy covariance measurements (Bai et al., 2009; Chehbouni et al., 2000; Ezzahar et al., 2009) and also against modeled fluxes (Marx et al., 2008; Samain et al., 2012; Watts et al., 2000). Few studies dealt with eXtra Large Aperture Scintillometer (XLAS) data (Kohsiek et al., 2006; Kohsiek et al., 2002; Moene et al., 2006). Historical survey, theoretical background as well as recent works in applied research concerning scintillometry are reviewed in De Bruin and Wang (2017). Since the scintillometer provides large-scale area-average sensible heat flux (H_XLAS), the corresponding latent heat flux (LE_XLAS) can then be computed as the energy balance residual term (LE_XLAS =Rn-G-H_XLAS), hence, the estimation of a representative value for the available energy (AE =Rn-G) is always crucial for the accuracy of the retrieved values of LE_XLAS. This assumption is valid only under the similarity hypothesis of Monin-Obukhov (MO) (Monin and Obukhov, 1954), *i.e.* surface homogeneity and stationary flows. These hypothesis are verified in our study area where topography is flat, and landscape is heterogeneous only from an agronomic point of view since we find different land uses (cereals, market gardening and fruit trees mainly olive trees with considerable spacing of bare soil); however, this heterogeneity in landscape features at field scale is randomly distributed and there is no drastic change in height and density of the vegetation at the scale of the XLAS transect (*i.e.* little heterogeneity at the km scale, most MODIS pixels have similar NDVI values for instance).

In this study, spatially distributed estimates of surface energy fluxes (sensible heat H and latent heat fluxes LE) over an irrigated area located in the Kairouan plain (Central Tunisia) were obtained by the SEB method, using the Soil Plant Atmosphere and Remote Sensing Evapotraspiration (SPARSE) model (Boulet et al., 2015) fed by 1-km thermal data and 1-km NDVI data from MODIS sensors on Terra and Aqua satellites. The main objective of this paper is to compare the modeled H and LE simulated by the SPARSE model with, respectively, the H measured by the XLAS and the LE reconstructed from the XLAS measurements acquired during two years over a large, heterogeneous area. We explore the consistency between the instantaneous H and LE estimates at the satellite overpass time, the water stress estimates and also ET derived at daily time step from both approaches.

## 2    Experimental site and datasets

### 2.1  Study area

The study site is a semi-arid region located in central Tunisia, the Kairouan plain (9°23′−10°17′E, 35°1′−35°55′N, (Figure 1). The landscape is mainly flat, and the vegetation is dominated by agricultural production (cereals, olive groves, fruit trees, market gardening, Zribi et al., 2011). Water management in the study area is typical of semi-arid regions with an upstream sub-catchment that transfers surface and subsurface flows collected by a dam (the El Haouareb dam), and a downstream plain (Kairouan plain) supporting irrigated agriculture (Figure 1). Agriculture consumes more than 80% of the total amount of water extracted each year from the Kairouan aquifer (Poussin et al., 2008). Most farmers in the plain uses their own wells to extract water for irrigation (Pradeleix et al., 2015), while a few depends on public irrigation schemes based on collective networks of water distribution pipelines all linked to a main borehole. The crop intensification in the last decades, associated to increasing irrigation, has led to growing water demand, and an overexploitation of the groundwater (Leduc et al., 2004).

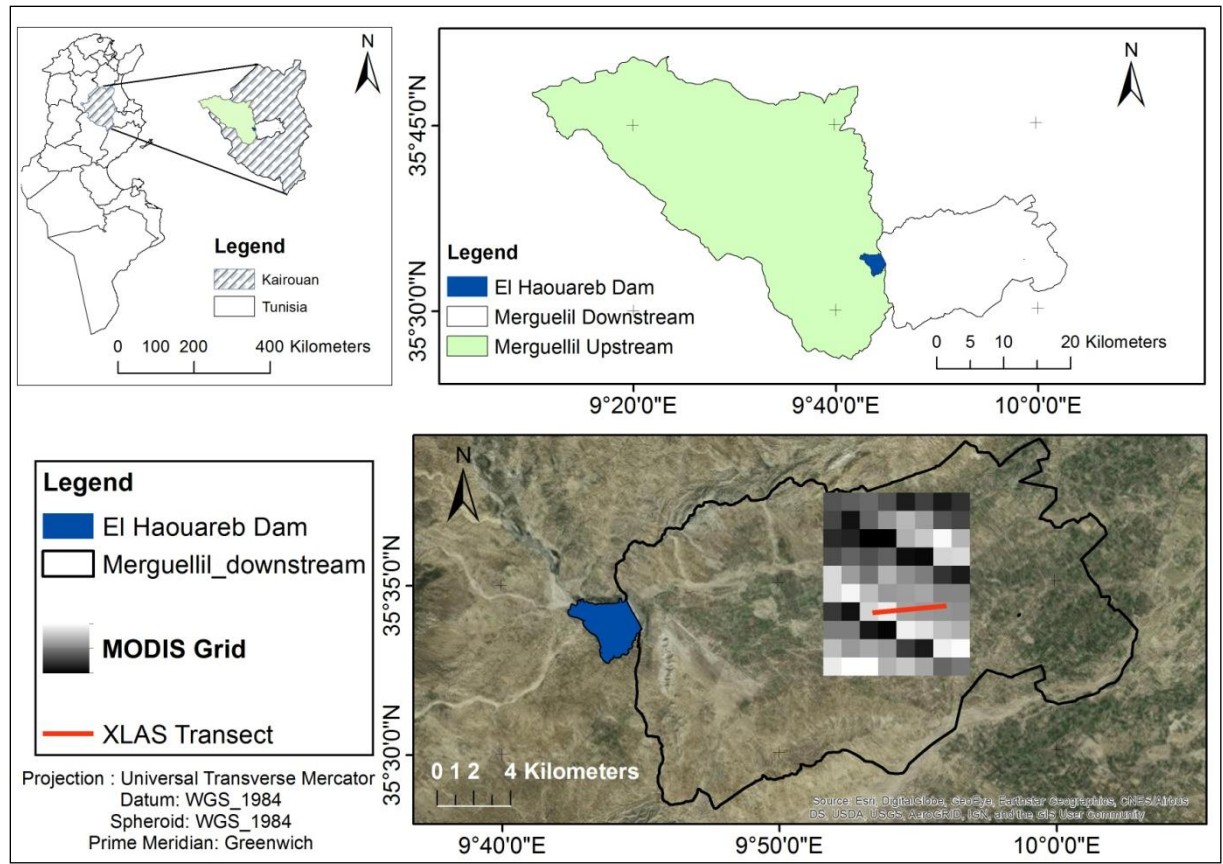

**Figure 1 : The study area: the downstream Merguellil sub-basin is the so called Kairouan plain; MODIS grid is the extracted 10 km × 8 km MODIS sub-image and in red the scintillomter XLAS transect**

### 2.2 Experimental set-up and remote sensing data

An optical Kipp and Zonen Extra Large Aperture Scintillometer (XLAS) was operated continuously for more than two years (1 March 2013 to 3 June 2015) over a relatively flat terrain (maximum difference in elevation of about 18 m). The scintillometer consists in a transmitter and a receiver both with an aperture diameter of 0.3 m, which allows longer path length. The wavelength of the light beam emitted by the transmitter is 940 nm. The transmitter was located on an eastern water tower (coordinates: 35° 34' 0.7" N; 9° 53' 25.19" E; 127 m above sea level) and the receiver on a western water tower (coordinates: 35° 34' 17.22" N; 9° 56' 7.30"E; 145 m above sea level) separated by a path length of 4 km (Figure 2).

The scintillometer transect was above mixed vegetation canopy: trees (mainly olive orchards) with some annual crops (cereals and market gardening) and the mean vegetation height is estimated about 1.17m along the transect. Both instruments were installed at 20 m height as recommended in the Kipp & Zonen instruction manual for LAS & XLAS (KIPP&ZONEN, 2007). At this height and for a 4-km path length, the devices are high enough to minimize measurement saturation and assumed to be above or close to the blending height where MO applied.

Furthermore, two automatic Campbell Scientific (Logan, USA) eddy covariance (EC) flux stations were also positioned at the same level on the two water tower top platforms. Half hourly turbulent fluxes in the western and the eastern EC stations were measured used a sonic anemometer CSAT3 (Campbell Scientific, USA) at a rate of 20 Hz and a sonic anemometer RM 81000 (Young, USA) at a rate of 10 Hz, respectively. The western

station data were more reliable with less measurement errors and gaps, hence, the western EC set-up was used to initialise friction velocity u* values and the Obukhov length Lo in the scintillometer flux computation (sect.3.1).

Half hourly standard meteorological measurements including incoming long wave radiation *i.e.* global incoming radiation ($Rg_{30}$), the incoming longwave radiation *i.e* atmospheric radiation ($R_{atm-30}$), wind speed ($u_{30}$), wind direction ($u_{d-30mn}$), air temperature ($T_{30}$) and relative humidity ($RH_{30}$) and barometric pressure ($P_{30}$) were recorded using an automated weather station installed in the study area (Figure 2), referred as the Ben Salem meteorological station (35° 33' 1.44" N; 9° 55' 18.11"E). Meteorological data were used either to force the SPARSE model or as input data in XLAS derived sensible and latent heat flux. The global incoming radiation was also used in the extrapolation method to scale instantaneous observed (sect. 3.3.2) and modeled (sect. 4.2) available energy as well as modeled sensible heat flux (sect. 4.2) to daily values.

In addition, an EC flux station, referred as the Ben Salem flux station (few tens of meters away from the meteorological station) was installed from November 2012 to June 2013 in an irrigated wheat field (Figure 2) measuring half hourly convective fluxes exchanged between the surface and the atmosphere ($H_{BS-30}$ and $LE_{BS-30}$) combined with measurements of the net radiation $Rn_{BS-30}$ and the soil heat flux $G_{BS-30}$. Net radiation and soil heat flux measurements were transferred to the meteorological station from June 2013 till June 2015. Since, there are no Rn and G measurements in the two water towers EC stations, $Rn_{BS}$ and $G_{BS}$ measurements were among the inputs data to derive sensible and latent heat fluxes from the XLAS measurements. In addition, measured available energy ($AE_{BS}=Rn_{BS}—G_{BS}$) and $H_{BS}$ were used to calibrate the extrapolation relationship of the available energy and the sensible heat flux, respectively (sect. 3.3.2 and 4.2).

Remotely sensed data were acquired for the study period (1[st] September 2012 to 30[th] June 2015) at the resolution of the MODIS sensor at 1 km, embarked on board of the satellites Terra (overpass time around 10:30 local solar time) and Aqua (overpass time around 13:30 local solar time). Downloaded MODIS products were (*i*) MOD11A1 and MYD11A1 for Terra and Aqua, respectively (surface temperature $T_{surf}$, surface emissivity $\varepsilon_{surf}$ and viewing angle $\phi$), (*ii*) MOD13A2 and MYD13A2 for Terra and Aqua, respectively (NDVI) and (*iii*) MCD43B1, MCD43B2 and MCD43B3 (albedo $\alpha$). These MODIS data provided in sinusoidal projection were reprojected in UTM using the MODIS Reprojection Tool. Then, sub-images of 10 km × 8 km centered on the XLAS transect (Figure 1) were extracted. The daily MODIS $T_{surf}$ and viewing angle, 8-day MODIS albedo, and 16-day MODIS NDVI contain some missing or unreliable data; hence, days with missing data (35% of all dates) in MODIS pixels regarding the scintillometer footprint (see later footprint computation in sect.3.2) were excluded. Albedo products (MCD43) are available every 8 days; the day of interest is the central date. Both Terra and Aqua data are used in the generation of this product, providing the highest probability for quality input data and designating it as a combined product. Moreover, the 1km/16days NDVI products (MOD13A2/MYD13A2) are available every 16 days and separately for Terra and Aqua. Algorithms generating this product operate on a per-pixel basis and require multiple daily observations to generate a composite NDVI value that will represent the full period (16 days). For both products, data are linearly interpolated over the available dates in order to get daily estimates. For each pixel, the quality index supplied with each product is used to select the best data.

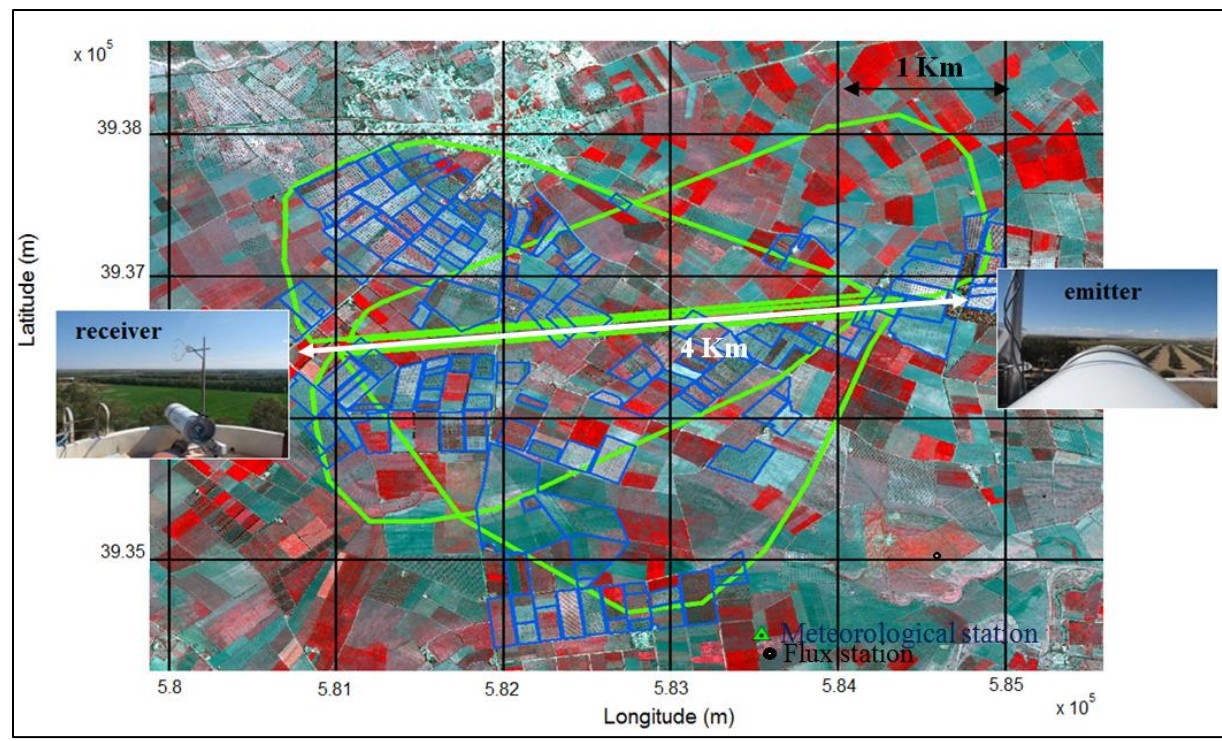


**Figure 2 : XLAS set-up: XLAS transect (white), for which the emitter and the receiver are located at the extremity of each white arrow, half-hourly XLAS footprint for selected typical wind conditions (green), MODIS grid (black), orchards (blue) and the location of the Ben Salem meteorological and flux stations. Background is a three color (red, green, blue) composite of SPOT5 bands 3 (NIR), 2 (VIS-red) and 1(VIS-green) acquired on 9th April 2013 and**

**showing in red the cereal plots.**

## 3    Extra Large aperture scintillometer (XLAS): data processing

### 3.1  Scintillometer derived fluxes

Scintillometer measurements are based on the scintillation theory; fluxes of sensible heat and momentum cause atmospheric turbulence close to the ground, and create, with surface evaporation, refractive index fluctuations

due mainly to air temperature and humidity fluctuations (Hill et al., 1980). The fluctuations intensity of refractive index is directly linked to sensible and latent heat fluxes. The light beam emitted by the XLAS transmitter towards the receiver is dispersed by the atmospheric turbulence. The scintillations representing the intensity fluctuations are analyzed at the XLAS receiver and are expressed as the structure parameter of the refractive index of air integrated along the optical path $C_{n^2}$ (m$^{-2/3}$) (Tatarskii, 1961). The sensitivity of the

scintillometer to $C_{n^2}$ along the beam is not uniform and follows a bell-shape curve due to the symmetry of the devices. This means that the measured flux is more sensitive to sources located towards the transect centre and is less affected by those close to the transect extremities.

In order to compute the XLAS sensible heat flux, $C_n^2$ was converted to the structure parameter of temperature turbulence $C_T^2$ (K$^2$m$^{-2/3}$) by introducing the Bowen ratio (ratio between sensible and latent heat fluxes), hereafter

referred to as β, which is a temperature /humidity correlation factor. Moreover, the height of the scintillometer beam above the surface varies along the path. In our study site, the terrain is very flat leading to little beam height variation across the landscape, except for what is induced by the different roughness of the individual fields. Since the interspaces between trees are large, the effective roughness of the orchards is not significantly different from that of annual crops fields. Consequently, $C_n^2$ and therefore $C_T^2$ are not only averaged horizontally

but vertically as well.

At visible wavelengths, the refractive index is sensitive to temperature fluctuations. Then, we can relate the $C_{n^2}$ to $C_{T^2}$ as follows:

$$C_{n^2} = \left(\frac{-0.78 \times 10^{-6} \times P}{T^2}\right)^2 C_{T^2} \left(1 + \frac{0.03}{\beta}\right)^2 \tag{1}$$

with T the air temperature (°K) and P the atmospheric pressure (Pa).

Green and Hayashi (1998) proposed another method to compute XLAS sensible heat flux (H_XLAS) assuming
full energy budget closure and using an iterative process without the need of β as an input parameter. This method is called the "β-closure method" (BCM, Twine et al., 2000). In the calculation algorithm, β is estimated iteratively with the BCM method, as described in Solignac et al. (2009) with initial guess using $Rn_{BS}$ and $G_{BS}$ from the Ben Salem flux station and initial $u_*$ coming from the western water tower EC station.

Then, the similarity relationship proposed by Andreas (1988) is used to relate the $C_{T^2}$ to the temperature scale $T_*$
in unstable atmospheric conditions as follows:

$$\frac{C_{T^2}(z_{LAS} - d)^{\frac{2}{3}}}{T_*^2} = 4.9\left(1 - 6.1\left(\frac{z_{LAS} - d}{L_O}\right)^{-\frac{2}{3}}\right) \tag{2}$$

And for stable atmospheric conditions:

$$\frac{C_{T^2}(z_{LAS} - d)^{\frac{2}{3}}}{T_*^2} = 4.9\left(1 + 2.2\left(\frac{z_{LAS} - d}{L_O}\right)^{\frac{2}{3}}\right) \tag{3}$$

where $L_O$ (m) the Monin Obukhov length, $Z_{LAS}$ (m) the scintillometer height, and d (m) the displacement height, which corresponds to 2/3 of the averaged vegetation height $z_v$.

$z_v$ accounts for the various heights within the selected footprint using angular zones originating from the centre
of the transect, and supported by high resolution remote sensing data (see Sect. 4.1).

Furthermore, considering the size of the surface changes in roughness (mean or effective vegetation height ~1.5m), the XLAS measurement height is assumed to be close to the blending height, or either higher. Thus, the fluxes measured by XLAS are area-averaged and MO similarity hypothesis can be applied in the flux algorithm computation.

Moreover, the fluxes measured with the XLAS were calculated for stability index ($Z_{LAS}/L_O$) comprised between - 2 and 0 (~73% of the cases). Then, according to Gruber and Fochesatto (2013), the sensitivity of the turbulent fluxes measurements to uncertainties in source measurements is rather similar between iterative algorithms or analytical solution, in this range of atmospheric stability.

From $T_*$ and the friction velocity $u_*$ (computed based on an iteration approach in the BCM method), the sensible
heat flux can be derived as follows:

$$H = -\rho c_p T_* u_* \tag{2}$$

where $\rho$ (kgm$^{-3}$) the density of air and $c_p$ (Jkg$^{-1}$K$^{-1}$) the specific heat of air at constant pressure.

H_XLAS was computed at a half hourly time step. Before flux computation, a strict filtering was applied to the XLAS data to remove outliers depending on weak demod signal. Negative night-time data were set to zero and

daytime flux missing data (one to three 30 mn-data) were gap filled using simple interpolation. Furthermore, half hourly H_XLAS aberrant values due to measurement errors and values higher than 400 $\text{Wm}^{-2}$, arising from measurement saturation, were ruled out (3% of the total measurement throughout the experiment duration). Finally, daily H_XLAS was computed as the average of the half hourly H_XLAS.

## 3.2 XLAS footprint computation

The footprint of a flux measurement defines the spatial context of the measurement and the source area that influences the sensors. In case of inhomogeneous surfaces like patches of various land covers and moisture variability due to irrigation, the measured signal is dependent on the fraction of the surface having the strongest influence on the sensor and thus on the footprint size and location. Footprint models (Horst and Weil, 1992; Leclerc and Thurtell, 1990) have been developed to determine what area is contributing to the heat fluxes as well as the relative weight of each particular cell inside the footprint limits. Contributions of upwind locations to the measured flux depend on the height of the vegetation, height of the instrumentation, wind speed, wind direction, and atmospheric stability conditions (Chávez et al., 2005).

According to the model of (Horst and Weil, 1992), for one-point measurement system, the footprint function $f$ relates the spatial distribution of surface fluxes, $F_0(x,y)$ to the measured flux at height $z_m$, $F(x,y,z_m)$, as follows:

$$F(x, y, z_m) = \int\limits_{-\infty}^{\infty} \int\limits_{-\infty}^{x} F_0(x', y') f(x - x', y - y', z_m) dx' dy' \tag{3}$$

The footprint function $f$ is computed as:

$$\bar{f}^y(x, z_m) = \frac{d\bar{z}}{dx} \frac{z_m}{\bar{z}^2} \frac{\bar{u}(z_m)}{\bar{u}(c\bar{z})} A e^{-(z_m/b\bar{z})^r} \tag{4}$$

where $\bar{u}(z)$ the mean wind speed profile and $\bar{z}$ the mean plume height for diffusion from a surface source. The variables $A$, $b$ and $c$ are scale factors and r a scale factor of the Gamma function. In the case of a scintillometer measurement, the footprint function has to be combined with the spatial weighting function $W(x)$ of the scintillometer to account for the sensor integration along its path. Thus, the sensible heat flux footprint mainly depends on the scintillometer effective height $Z_{LAS}$(Hartogensis et al., 2003), which includes the topography below the path and the transmitter and receiver heights, the wind direction and the Obukhov length $L_O$, which characterizes the atmospheric stability (Solignac et al., 2009). In a subsequent step, daily footprints were computed as a weighted sum of the half hourly footprints by the XLAS sensible heat flux.

In fact, there is an issue with the MODIS pixel heterogeneity and notably the distribution of the land use classes at the intersection between the square pixel and the XLAS footprint (Bai et al., 2015). Hence, in order to provide a first guess on these relative heterogeneities, land use classes within each MODIS pixel of the 10 km × 8 km sub-image were studied based on the land use map of the 2013-2014 season (Chahbi, 2016). The average footprint of all half hourly footprints for the whole study period was computed and overlaid on the MODIS grid in order to identify the MODIS pixels partially or totally covered by footprint (Figure 3).

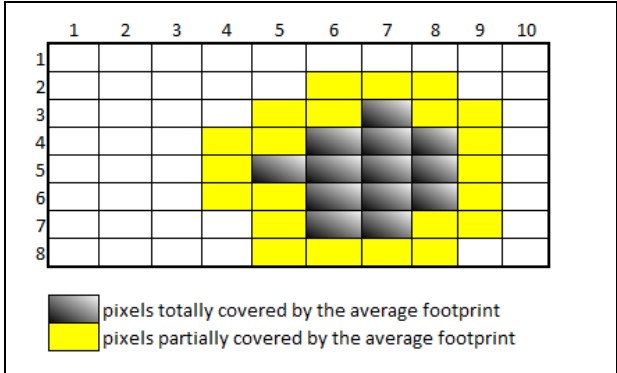

**Figure 3 : MODIS pixels partially or totally covered by XLAS source area**

The percentage of land use classes was computed for (i) the part of each pixel that lies within the footprint, and (ii) the complementary part of the pixel located outside of the footprint (Figure 4). Results show that difference in percentages of each land use classes for the pixel fractions located within or outside the footprint is low with 1.8%, 1.7%, 1.0% and 3.5% for cereals, market gardening, trees and bare soil, respectively. Moreover, the major

part of the area above transect is covered by fallow and orchards. The land use classes' partition inside the 13 MODIS pixels totally covered by the average footprint is comparable.

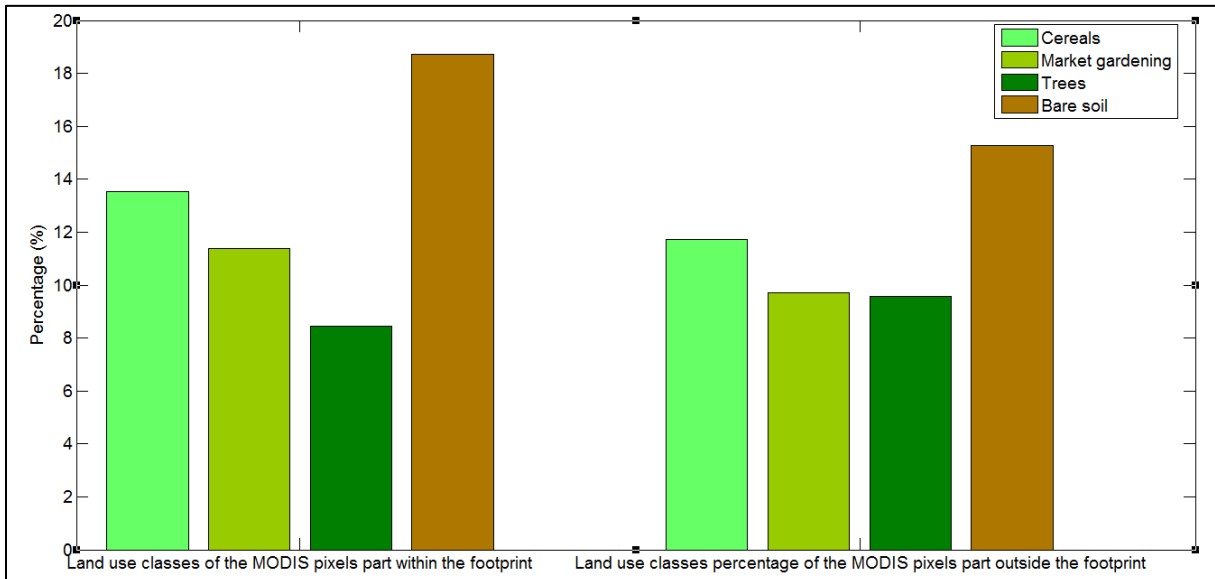

**Figure 4: Land use classes' percentage of the MODIS pixels within or outside the footprint**

### 3.3 XLAS derived latent heat flux

Instantaneous (LE_residual_XLAS$_{t\text{-FP}}$) and daily (LE_residual_XLAS$_{day\text{-FP}}$) XLAS derived latent heat flux (*i.e.* residual latent heat flux) of the XLAS upwind area were computed using the energy budget closure of the XLAS measured sensible heat flux (H_XLAS) with additional estimations of remotely sensed net surface radiation Rn and soil heat flux G, as available energy (AE=Rn-G), as follows:

$$LE\_residual\_XLAS_{t-FP} = AE_{t-FP} - H\_XLAS_t \qquad (5)$$

$$LE\_residual\_XLAS_{day-FP} = AE_{day-FP} - H\_XLAS_{day} \qquad (6)$$

H_XLAS$_t$ and H_XLAS$_{day}$ are respectively the instantaneous and daily measured H at the time of the satellite overpass interpolated from the half hourly fluxes measurements. Daily available energy within the footprint (AE$_{day-FP}$) was computed from instantaneous available energy (AE$_{t-FP}$) as detailed in Sect. 3.3.1 and Sect. 3.3.2. The subscripts "30", "day" and "t" refer to half hourly, daily and instantaneous (at the time of Terra and Aqua overpasses) variables, respectively; while the subscript "FP" means that the footprint is taken into account *i.e.* instantaneous or the daily (depending on time scale) footprint was multiplied by the variable.

### 3.3.1 Instantaneous available energy

Net surface radiation is the balance of energy between incoming and outgoing shortwave and longwave radiation fluxes at the land-atmosphere interface. Remotely sensed surface radiative budget components provide unparalleled spatial and temporal information, thus several studies have attempted to estimate net radiation by combining remote sensing observations with surface and atmospheric data. Net radiation equation can be written as follows:

$$\text{Rn} = (1 - \alpha)Rg + \varepsilon_{surf}R_{atm} - \varepsilon_{surf}\sigma T_{surf}^4 \tag{7}$$

where Rg the incoming shortwave radiation (W.m$^{-2}$), R$_{atm}$ the incoming longwave radiation (W.m$^{-2}$), $\alpha$ the albedo, $\varepsilon_{surf}$ the surface emissivity, T$_{surf}$ the surface temperature (°K) and $\sigma$ Stefan-Boltzmann constant (W.m$^{-2}$.K$^4$) .The soil heat flux G depends on the soil type and water content as well as the vegetation type (Allen et al., 2005).The direct estimation of G by remote sensing data is not possible (Allen et al., 2011), however, empirical relations can estimate the fraction $\xi$=G/Rn as a function of soil and vegetation characteristics using satellite image data, such as the LAI, NDVI, $\alpha$ and T$_{surf}$. Generally, G represents 5-20% of Rn during daylight hours (Kalma et al., 2008). In order to estimate the G/Rn ratio, several methods have been tested for various types of surfaces at different locations. The most common methods parameterize $\xi$ as a constant for the entire day or at satellite overpass time (Ventura et al., 1999), according to NDVI (Jackson et al., 1987; Kustas and Daughtry, 1990), LAI (Choudhury et al., 1987; Kustas et al., 1993; Tasumi et al., 2005), vegetation fraction (fc ) (Su, 2002), T$_{surf}$ and $\alpha$ (Bastiaanssen, 1995), or only T$_{surf}$ (Santanello Jr and Friedl, 2003). These empirical methods are suitable for specific conditions; therefore, estimating G, especially in this type of environment where NDVI values are low and thus G/Rn values are large, is a critical issue. The approach adopted here was drawn on Danelichen et al. (2014) who evaluated the parameterization of these different models in three sites in Mato Grosso state in Brazil and found that the model proposed by (Bastiaanssen, 1995) showed the best performance for all sites, followed by the model from Choudhury et al. (1987) and Jackson et al. (1987):

Bastiaanssen (1995):

$$G = \text{Rn}(T_{surf} - 273.16)(0.0038 + 0.0074\alpha)(1 - 0.98\text{NDVI}^4) \tag{8}$$

Choudhury et al. (1987):

$$G = 0.4Rn(\exp(-0.5LAI)) \tag{9}$$

Jackson et al. (1987)

$$G = 0.583Rn(\exp(-2.13NDVI)) \tag{10}$$

Hence, these three methods were tested for the Ben Salem flux station measurements, by comparing the measured $G_{BS-t}$ and the computed G using measured $Rn_{BS-t}$, $T_{surf-BS-t}$, $\alpha_{BS}$, $NDVI_{BS}$ and $LAI_{BS}$ at Terra and Aqua overpass time (results not shown). The best results are issued from Bastiaanssen (1995) method with a Root Mean Square Error (RMSE) of 0.09 (average value of the two satellites overpass time) followed by Jackson et al. (1987) and Choudhury et al. (1987) with RMSE values of 0.15 and 0.2, respectively. Moreover, daily measured $G_{BS-day}$ was computed and a G accumulation is generally found as it has been already mentioned by (Clothier et al., 1986) who showed that G is neither constant nor negligible on diurnal timescales, and can constitute as much as 50% of Rn over sparsely vegetated area. Since G estimation was the most uncertain variable, the three above methods were tested to compute the distributed remotely sensed AE. The Ben Salem meteorological station was used to provide $Rg_t$ and $R_{atm-t}$. Remote sensing variables $\alpha$, $T_{surf}$, $\varepsilon_{surf}$ and NDVI came from MODIS products. Remotely sensed LAI was computed from the MODIS NDVI using a single equation (Clevers, 1989) for all crops in the study area:

$$LAI = -\frac{1}{k} ln \left( \frac{NDVI_\infty - NDVI}{NDVI_\infty - NDVI_{soil}} \right) \tag{11}$$

The calibration of this relationship was done over the Yaqui irrigated perimeter (Mexico) during the 2007-2008 growing season using hemispherical LAI measured in all the studied fields (Chirouze et al., 2014). Calibration results gave the asymptotical values of NDVI, $NDVI_\infty = 0.97$ and $NDVI_{soil} = 0.05$, as well as the extinction factor k=1.13. As this relationship was calibrated over a heterogeneous land surface but on herbaceous vegetation only, its relevance for trees was checked. For that purpose, clump-LAI measurements on an olive tree, as well as allometric measurements *i.e.* mean distance between trees and mean crown size done using Pleiades satellite data (Mougenot et al., 2014;Touhami, 2013) were obtained. Clump LAI is the value of the LAI of an isolated element of vegetation (tree, shrub...); if this element occupies a fraction cover f and is surrounded by bare soil, then the clump LAI value is equal to the area average LAI divided by f. Hence, we checked that the pixels with tree dominant cover show LAI values close to what was expected (of the order of 0.3 to 0.4 given the interrow distance of 12 m on average).

Remote sensed available energy was computed for the 10 km × 8 km MODIS sub-images at Terra-MODIS and Aqua-MODIS overpass time, using the three methods estimating G. Since the measured heat fluxes $H\_XLAS_t$ represent only the weighted contribution of the fluxes from the upwind area to the tower (footprint), then instantaneous footprint at the time of Terra and Aqua overpass were selected among the two half hour preceding and following the satellite's time of overpass (lowest time interval) and then was multiplied by the instantaneous remote sensed available energy $AE_t$ to get the available energy of the upwind area $AE_{t-FP}$.

### 3.3.2   Daily available energy

Most methods using TIR domain data rely on once-a-day acquisitions, late morning (such as Terra-MODIS overpass time) or early afternoon (such as Aqua-MODIS overpass time). Thus, they provide a single instantaneous estimate of energy budget components. In order to obtain daily AE from these instantaneous measurements and to reconstruct hourly variations of AE, we considered that its evolution was proportional to another variable whose diurnal evolution can be easily known.

The extrapolation from an instantaneous flux estimate to a daytime flux assumes that the surface energy budget is "self-preserving" *i.e.* the relative partitioning among components of the budget remains constant throughout

the day. However, many studies (Brutsaert and Sugita, 1992; Gurney and Hsu, 1990; Sugita and Brutsaert, 1990) showed that the self-preservation method gives day-time latent heat estimates that are smaller than observed values by 5-10%. Moreover, (Anderson et al., 1997) found that the evaporative fraction computed from instantaneous measured fluxes tends to underestimate the daytime average by about 10%, hence, a corrected parameterization was used and a coefficient=1.1 was applied. Similarly, Delogu et al. (2012) found an overestimation of about 10% between estimated and measured daily component of the available energy thus, a coefficient =0.9 was applied. The corrected parameterization proposed by Delogu et al. (2012) was tested, but this coefficient did not give consistent results, therefore, the extrapolation relationship was calibrated in order to get accurate daily results of AE .

Thereby, the applied extrapolation method was tested using *in situ* Ben Salem flux station measurements. The incoming short wavelengths radiation was used to scale available energy from instantaneous to daily values; but only for clear sky days for which MODIS images can be acquired and remote sensing data used to compute AE are available. Clear sky days were selected based on the ratio of daily measured incoming short wavelengths radiation $Rg_{day}$ to the theoretical clear sky radiation Rso as proposed by the FAO-56 method (Allen et al., 1998). A day was defined as clear if the measured $Rg_{day}$ is higher than 85 % of the theoretical clear sky radiation at the satellite overpass time (Delogu et al., 2012).

Daily measured available energy $AE_{BS-day}$ computed as the average of half-hourly measured $AE_{BS-30}$, was compared to daily available energy ($AE_{BS-day-Terra}$ and $AE_{BS-day-Aqua}$) computed using the extrapolation method from instantaneous measured $AE_{BS-t-Terra}$ and $AE_{BS-t-Aqua}$ at Terra and Aqua overpass time, respectively (Equation 14).

$$AE_{BS-day-Terra} = a_{Terra} Rg_{day} \frac{AE_{BS-t-Terra}}{Rg_{t-Terra}} + b_{Terra}$$

(1 4)

$$AE_{BS-day-Aqua} = a_{Aqua} Rg_{day} \frac{AE_{BS-t-Aqua}}{Rg_{t-Aqua}} + b_{Aqua}$$

where $Rg_{day}$ is the daily measured incoming short wavelengths radiation in the Ben Salem meteorological station; $Rg_{t-Terra}$ and $Rg_{t-Aqua}$ are the instantaneous incoming short wavelengths radiations measured at Terra and Aqua overpass time, respectively and $AE_{BS-t-Terra}$ and $AE_{BS-t-Aqua}$ are the instantaneous measured available energy in the Ben Salem flux station, at Terra and Aqua overpass time.

Results gave an overestimation of about 15 %. The corrected parameterizations of AE (Table 1), needed to remove the bias between measured ($AE_{BS-day}$) and computed AE ($AE_{BS-day-Terra}$ and $AE_{BS-day-Aqua}$), were applied to compute daily remotely sensed AE ($AE_{day}$) from instantaneous AE ($AE_t$) following the extrapolation method shown in equation 14.

**Table 1: Corrected parameterizations of available energy for the diurnal reconstitution**

| Terra | $a_{Terra}$ | 0.85 |
|---|---|---|
| | $b_{Terra}$ | -19.81 |
| Aqua | $a_{Aqua}$ | 0.87 |
| | $b_{Aqua}$ | -18.94 |

Then $AE_{day}$ was multiplied by the weighting coefficients ranging from zero and one of the corresponding daily footprint to get the daily available energy of the upwind area $AE_{day\text{-}FP}$. Finally, estimates of Terra and Aqua observed daily LE (LE_residual_XLAS$_{day\text{-}FP}$) were obtained based on the three methods used to compute G.

## 4  SPARSE model

### 4.1  Energy fluxes derived from SPARSE model

The SPARSE dual-source model solves the energy budgets of the soil and the vegetation. Here we use the "layer approach", for which the resistance network relating the soil and vegetation heat sources to a main reference level through a common aerodynamic level use a series electrical branching. Main unknowns are the component temperatures, *i.e.* soil ($T_s$) and vegetation ($T_v$) temperatures. Totals at the reference height (the measurement height of the meteorological forcing), as well as the longwave radiation budget, are also solved so that altogether a system of five equations can be built:

$$\begin{cases} H = H_s + H_v \\ LE = LE_s + LE_v \\ R_{ns} = G + H_s + LE_s \\ R_{nv} = H_v + LE_v \\ \varepsilon_{surf}\sigma T_{surf}^4 = \varepsilon_{surf}R_{atm} - R_{an} \end{cases} \tag{15}$$

where $R_{atm}$ the atmospheric radiation (Wm$^{-2}$), $R_{an}$ net longwave radiation which depends on Ts and Tv (Wm$^{-2}$), $T_{surf}$ and $\varepsilon_{surf}$ are respectively the surface temperature (°K) and emissivity as observed by the satellite; indexes "s" and "v" designate the soil and the vegetation, respectively.

The first two (Eq. (15)) express the continuity of the latent and sensible heat fluxes from the sources to the aerodynamic level through to the reference level, the third and the fourth (Eq. (15)) are the soil and vegetation energy budgets, and the fifth (Eq. (15)) relates the surface temperature $T_{surf}$ derived from observed MODIS surface temperature to $T_s$ and $T_v$ .

The SPARSE model system of equations is fully described in Boulet et al. (2015). SPARSE is similar to the TSEB model (Kustas and Norman, 1999) but includes the expressions of the aerodynamic resistances of Choudhury and Monteith (1988) and Shuttleworth and Gurney (1990). This system can be solved in a forward mode for which the surface temperature is an output (prescribed conditions), and an inverse mode when the surface temperature is an input derived from satellite observations or *in situ* measurements in the thermal infra-red domain (retrieval conditions). Figure 5 illustrates a diagram showing the flowchart of the model algorithm. System (15) is solved step-by-step by following similar guidelines as in the TSEB model: the first step assumes that the vegetation transpiration (LE$_v$) is maximum, and evaporation (LE$_s$) is computed. If this soil latent heat flux (LE$_s$) is below a minimum positive threshold for vegetation stress detection of 30 Wm$^{-2}$, the hypothesis that the vegetation is unstressed is no longer valid. In that case, the vegetation is assumed to suffer from water stress and the soil surface is assumed to be already long dry. Then, LE$_s$ is set to 30 Wm$^{-2}$. This value accounts for the small but non negligible vapor flow reaching the surface (Boulet et al., 1997). The system is then solved for vegetation latent heat flux (LE$_v$). If LE$_v$ is also negative, both LE$_s$ and LE$_v$ values are set to zero, whatever the value of $T_{surf}$. The system of equation can also be solved for $T_s$ and $T_v$ only if the efficiencies representing stress levels (dependent on surface soil moisture for the evaporation, and root zone soil moisture for the transpiration) are known. In that case the sole first four equations are solved. This prescribed mode allows computing all the

fluxes in known limiting soil moisture levels (very dry, e.g. fully stressed, and wet enough, e.g. potential). It limits unrealistically high values of component fluxes, latent heat flux values above the potential rates or sensible heat flux values above that of a non evaporating surface. The potential evaporation and transpiration rates used later on are computed using this prescribed mode with minimum surface resistance to evaporation and transpiration, respectively.

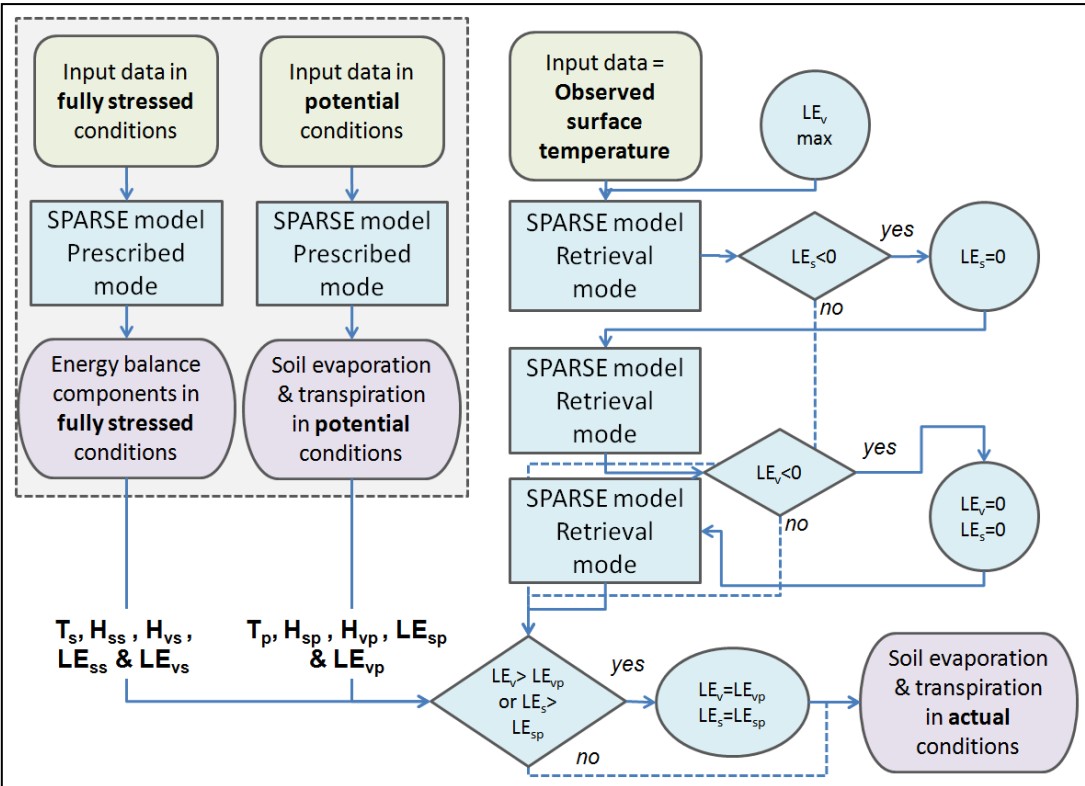

**Figure 5: Flowchart of the SPARSE algorithm; $T_s$, $H_{ss}$, $H_{vs}$ $LE_{ss}$ and $LE_{vs}$ are surface temperature, soil sensible heat flux, vegetation sensible heat flux, soil latent heat flux and vegetation latent heat flux in fully stressed conditions, respectively; $T_p$, $H_{sp}$, $H_{vp}$, $LE_{sp}$ and $LE_{vp}$ are surface temperature, soil sensible heat flux , vegetation sensible heat flux, soil latent heat flux and vegetation latent heat flux in potential conditions, respectively.**

Some of the model parameters were remotely sensed data while others were taken from the bibliography or measured *in situ*. Remotely sensed data fed into SPARSE are $T_{surf}$, $\varepsilon_{surf}$, $\phi$, NDVI, LAI and $\alpha$. A grid of the vegetation height ($z_v$) was also necessary as input in the SPARSE model; for herbaceous crops, vegetation height was interpolated with the help of NDVI time series between fixed minimum (0.05 m) and maximum (0.8 m) values, while for trees, the roughness length ($z_{om}$) was linked to the allometric measurements (mentioned before) and computed as a function of canopy area index, drag coefficient and canopy height using the drag partition approach proposed by Raupach (1994) for tall sparse vegetative environments. Then, since SPARSE deals with vegetation height and not roughness length, the same simple rule of the thumb as the one used in SPARSE was used to reconstruct $z_v$ for the tree cover types ($z_v=z_{om}/0.13$). In a final step, to get spatial vegetation height, $z_v$ was averaged over the MODIS pixels. *In situ* parameters used in SPARSE were mainly meteorological data: Rg, $R_{atm}$, Ta, Ha and u. No calibration was performed on the model parameters shown in Table 2.

**Table 2. SPARSE parameters**

| | Definition | Value | Data Sources |
|---|---|---|---|
| ***Remote sensing parameters*** | | | |
| NDVI | Normalized Difference Vegetation Index | | Satellite imagery |
| $T_{surf}$ (K) | Surface temperature (K) | | Satellite imagery |
| α | Albedo | | Satellite imagery |
| $\varepsilon_{surf}$ | Surface emissivity | | Satellite imagery |
| Φ (rad) | View zenith angle | | Satellite imagery |
| ***Meteorological parameters*** | | | |
| $R_g$ (Wm$^{-2}$) | Incoming solar radiation | | *In situ* data |
| $R_{atm}$ (Wm$^{-2}$) | Incoming atmospheric radiation | | *In situ* data |
| $T_a$ (K) | Air temperature at reference level | | *In situ* data |
| $RH_a$ (%) | Air relative humidity | | *In situ* data |
| $u_a$ (ms$^{-1}$) | Horizontal wind speed at reference level | | *In situ* data |
| ***Fixed parameters*** | | | |
| $z_a$ (m) | Atmospheric forcing height | 2.32 | *In situ* data |
| $z_v$ (m) | Vegetation height | | Derived from land cover |
| $\beta_{pot}$ | Evapotranspiration efficiency in full potential conditions | 1.000 | |
| $\beta_{stress}$ | Evapotranspiration efficiency in fully stressed conditions | 0.001 | |
| $r_{stmin}$ (sm$^{-1}$) | Minimum stomatal resistance | 100 | (Boulet et al., 2015) |
| $w$ (m) | Leaf width | 0.05 | (Braud et al., 1995) |
| $\varepsilon_v$ | Vegetation emissivity | 0.98 | (Braud et al., 1995) |
| $\alpha_v$ | Vegetation albedo | 0.25 | Estimation |
| ***Constants*** | | | |
| $\rho_{cp}$ (J.kg$^{-1}$.K$^{-1}$) | Product of air density and specific heat | 1170 | (Braud et al., 1995) |
| σ (W. m$^{-2}$.k$^4$) | Stefan–Boltzmann constant | 5.66. 10$^{-8}$ | (Braud et al., 1995) |
| γ (Pa.K$^{-1}$) | Psychrometric constant | 0.66 | (Braud et al., 1995) |
| $z_{om,s}$ (m) | Equivalent roughness length of the underlying bare soil in the absence of vegetation | 5.10$^{-3}$ | (Braud et al., 1995) |
| $n_{SW}$ | Coefficient in $r_{av}$ (Aerodynamic resistance between the vegetation and the aerodynamic level) | 2.5 | (Boulet et al., 2015) |
| ξ | Ratio between soil heat flux G and available net radiation on the bare soil $Rn_s$ | 0.4 | (Braud et al., 1995) |

The retrieval and prescribed modes of the SPARSE model were run for the 10 km × 8 km sub-images at the time of Terra and Aqua overpasses, to get instantaneous modeled fluxes H_SPARSE$_t$, LE_SPARSE$_t$ and AE_SPARSE$_t$ as well as sensible heat flux (H$_{s-t}$ =H$_{ss-t}$ +H$_{vs-t}$) in fully stressed conditions and latent heat (LE$_{p-t}$ =LE$_{sp-t}$ +LE$_{vp-t}$) and sensible heat (H$_{p-t}$ =H$_{sp-t}$ +H$_{vp-t}$) fluxes in potential conditions. Modeled values were then multiplied by the nearest half hourly footprint to the satellite overpass time, in order to get fluxes corresponding to the upwind area: H_SPARSE$_{t-FP}$, LE_SPARSE$_{t-FP}$, AE_SPARSE$_{t-FP}$, H$_{s-t-FP}$, H$_{p-t-FP}$ and LE$_{p-t-FP}$.

In a subsequent step, the retrieval and prescribed modes of SPARSE model was run at a half hourly time step using the half hourly meteorological measurements to get half hourly latent heat flux at potential conditions LE$_{p-30}$ and half hourly modeled available energy AE_SPARSE$_{30}$. The potential LE weighted by the corresponding half hourly footprint (LE$_{p-30-FP}$) is used later when computing daily LE based on the stress factor method while the half hourly AE weighted by the corresponding half hourly footprint (AE_SPARSE$_{30-FP}$) were used to compute daily LE based on the evaporative fraction method (section 4.2).

## 4.2 Reconstruction of daily modeled ET from instantaneous latent heat flux

Daily ET is usually required for applications in hydrology or agronomy for instance, whereas most SEB methods provide a single instantaneous latent heat flux because the energy budget is only computed at the satellite overpass time (Delogu et al., 2012). In order to scale daily ET from one instantaneous estimate, there are various methods relying on the preservation, during the day, of the ratio of the latent heat flux to a scale factor having known diurnal evolution.

- *Stress Factor (SF) method*

The stress factor SF (Eq. (16)) is assumed invariant during the same day, the diurnal modeled fluxes are accounted for by recovering the diurnal course of potential ET.

$$SF = 1 - \frac{LE\_SPARSE_{t-FP}}{LE_{p-t-FP}} \tag{16}$$

The daily modeled ET (LE_SPARSE$_{day-FP}$) can be expressed as the product of the instantaneous estimate of SF at the satellite overpass time and the daily potential evapotranspiration :

$$\boldsymbol{LE\_SPARSE_{day-FP} = (1 - SF)LE_{p-day-FP}} \tag{17}$$

LE$_{p-day-FP}$ was calculated as the sum of the half hourly modeled latent heat fluxes at potential conditions LE$_{p-30-FP}$.

- *Evaporative Fraction method*

The evaporative fraction (EF) self-preservation is a valid assumption under dry conditions but no longer under wet conditions (Hoedjes et al., 2008). For these conditions, assuming a constant EF underestimates actual EF and therefore ET (Lhomme and Elguero, 1999). Indeed, according to Gentine et al. (2007), the diurnal shape of EF depends on both atmospheric forcing and surface conditions. Therefore EF was computed every 30 minutes using the following empirical parameterization (Delogu et al., 2012):

$$EF_{30} = \left[1.2 - \left(0.4\frac{Rg_{30}}{1000} + 0.5\frac{RH_{30}}{100}\right)\right]\left(\frac{EF_{SPARSE-t}}{EF_{met-t}}\right) \tag{18}$$

where Rg$_{30}$ and RH$_{30}$ are respectively the half hourly incoming short wavelengths radiations and relative humidity EF$_{SPARSE-t}$ and EF$_{met-t}$ are respectively SPARSE EF (Eq. (19)) and computed EF using the evaporative fraction method (Eq. (20)) at the satellite overpass time.

$$EF_{SPARSE-t} = \frac{LE\_SPARSE_{t-FP}}{AE\_SPARSE_{t-FP}} \tag{19}$$

$$EF_{met-t} = \left[1.2 - \left(0.4\frac{Rg_t}{1000} + 0.5\frac{RH_t}{100}\right)\right] \tag{20}$$

where LE_SPARSE$_{t-FP}$ and AE_SPARSE$_{t-FP}$ are respectively the latent heat flux and the available energy modeled by SPARSE at the satellite overpass time; Rg$_t$ and RH$_t$ are respectively the incoming short wavelengths radiations and relative humidity measured at the time of the satellite overpass.

The half hourly modeled ET (LE_SPARSE$_{30-FP}$) was computed as the product of the half hourly EF estimate and the half hourly modeled available energy AE_SPARSE$_{30-FP}$ (Eq. 21). AE_SPARSE$_{30}$ was computed from instantaneous modeled available energy (AE_SPARSE$_t$) using the same approach detailed in Sect. 3.3.2 and applying equation (14) for a half hourly time step (instead of a daily time step). AE_SPARSE$_{30}$ was weighted by

the corresponding half hourly footprint to get the modeled AE of the upwind area AE_SPARSE$_{30\text{-FP}}$. The daily modeled ET (LE_SPARSE$_{\text{day-FP}}$) was computed as the sum of the half hourly LE_SPARSE$_{30\text{-FP}}$ (Eq. 22).

$$LE\_SPARSE_{30-FP} = EF_{30} \times AE\_SPARSE_{30-FP} \qquad (21)$$

$$LE\_SPARSE_{day-FP} = \sum LE\_SPARSE_{30-FP} \qquad (22)$$

• *Residual method*

Besides, daily modeled ET (LE_SPARSE$_{\text{day-FP}}$) was also estimated as a residual term of the surface energy budget using daily modeled sensible heat flux (H_SPARSE$_{\text{day-FP}}$) and available energy (AE_SPARSE$_{\text{day-FP}}$) as follows:

$$\mathbf{LE\_SPARSE_{day-FP} = AE\_SPARSE_{day-FP} - H\_SPARSE_{day-FP}} \qquad (23)$$

H_SPARSE$_{\text{day}}$ was computed from modeled sensible heat flux (H_SPARSE$_t$) following the same extrapolation 565 method used for the available energy (see Sect. 3.3.2). The corrected parameterizations of H were got from the comparison of daily measured sensible heat flux H$_{\text{BS-day}}$ computed as the average of half-hourly measured H$_{\text{BS-30}}$ and daily sensible heat flux (H$_{\text{BS-day-Terra}}$ and H$_{\text{BS-day-Aqua}}$) computed using the extrapolation method from instantaneous measured H$_{\text{BS-t-Terra}}$ and H$_{\text{BS-t-Aqua}}$ at Terra and Aqua overpass time, respectively (Eq. 24).

$$H_{BS-day-Terra} = a'_{Terra} Rg_{day} \frac{H_{BS-t-Terra}}{Rg_{t-Terra}} + b'_{Terra}$$

$$\qquad (24)$$

$$H_{BS-day-Aqua} = a'_{Aqua} Rg_{day} \frac{H_{BS-t-Aqua}}{Rg_{t-Aqua}} + b'_{Aqua}$$

where H$_{\text{BS-t-Terra}}$ and H$_{\text{BS-t-Aqua}}$ are the instantaneous measured sensible heat flux in the Ben Salem flux station. 570 Therefore, the corrected parameterizations of H (Table 3), needed to remove the bias between measured (H$_{\text{BS-day}}$) and computed H (H$_{\text{BS-day-Terra}}$ and AE$_{\text{BS-day-Aqua}}$), were applied to compute daily modeled H ( H_SPARSE$_{\text{day}}$) from instantaneous modeled H (H_SPARSE$_t$) following the extrapolation method shown in equation 21. Finally, H_SPARSE$_{\text{day}}$ was weighted by the corresponding daily footprint to get the daily modeled H of the upwind area H_SPARSE$_{\text{day-FP}}$.

**Table 3: Corrected parameterizations of sensible heat flux for the diurnal reconstitution**

| Terra | a'$_{Terra}$ | 1.02 |
|---|---|---|
| | b'$_{Terra}$ | -17.31 |
| Aqua | a'$_{Aqua}$ | 1.00 |
| | b'$_{Aqua}$ | -14.83 |

## 5 Water stress estimates

Water stress estimation is crucial to deduce the root zone soil moisture level using remote sensing data, (Hain et al., 2009). Water stress results in a drop of actual evapotranspiration below the potential rate. Its intensity is 580 usually represented by a stress factor as defined in Sect. 4.2, ranging between 0 (unstressed surface) and 1 (fully stressed surface).

Modeled values of SF at the time of Terra and Aqua overpass ($SF_{mod}$) have been computed from modeled potential LE ($LEp_{-t-FP}$) as follows:

$$SF_{mod} = 1 - \frac{LE\_SPARSE_{t-FP}}{LE_{p-t-FP}} \qquad (25)$$

where $LE\_SPARSE_{t-FP}$ and $LE_{p-tFP}$ are the modeled latent heat fluxes in actual and potential conditions, respectively.

Furthermore, surface water stress factor derived from XLAS measurement, named $SF_{obs}$, at the time of Terra and Aqua overpass was computed as follows (Su, 2002):

$$SF_{obs} = \frac{H\_XLAS_t - H_{p-t-FP}}{H_{s-t-FP} - H_{p-t-FP}} \qquad (26)$$

where $H_{s-t-FP}$ and $H_{p-t-FP}$ are the modeled sensible heat flux in actual and potential conditions, respectively; and $H\_XLAS_t$ is the XLAS sensible heat flux at the satellite overpass time.

## 6    Results and discussion

### 6.1 XLAS and model derived instantaneous sensible heat fluxes

Our primary focus is the comparison between scintillometer measurements and the modeled sensible heat fluxes computed using the Terra and Aqua remotely sensed data. The scintillometer H at the time of the two satellites overpass ($H\_XLAS_t$) are interpolated from the half hourly H measurements. Heat flux determination was possible for typically about 87% of the daytime measurements during the summer, availability of XLAS heat flux values was lower during the cold season due to poor visibility and/or stable stratification.

H_SPARSE was weighted by the XLAS footprint in order to be able to compare the modeled values ($H\_SPARSE_{t-FP}$) with the XLAS measurements ($H\_XLAS_t$). Therefore, due to XLAS and remote sensing data availability, we got 175 and 118 values for Terra and Aqua respectively. In order to highlight H inter-seasonality between the drier 2012-2013 and the wetter 2013-2014 seasons, we present an example of two days each in one season, DOY 2013-082 shows H value ranging between 163 $Wm^{-2}$ and 342 $Wm^{-2}$ while DOY 2014-208 shows H value ranged between 97 $Wm^{-2}$ and 311 $Wm^{-2}$ (Figure 6). The colored area shows the modeled flux and the contours shows the surface source area contributing to the scintillometer measurements. The Day 2013-82 (23[th] March 2013) is chosen in the cold season while day 208-2014 (27[th] July 2014) is in the warm season to focus on land cover impact on $T_{surf}$ and thus on modeled H, (trees and cereals in winter vs. only irrigated trees and market gardening in summer). Moreover, the first day experiences a strong southern wind while there is a light northern wind during the second day. Generally, a little number of MODIS pixels brings a high contribution to the signal; among them two are hot pixels (pixel with high $T_{surf}$ and low NDVI) in which the land use is mainly arboriculture.

Prediction performance is assessed using RMSE and the coefficient of determination ($R^2$). Results for the sensible heat flux are illustrated in figure 7 and show good agreement between modeled and measured H at the time of satellites overpass. This is illustrated by linear regressions of $H\_SPARSE_{t-FP} = 1.065 \ H\_XLAS_t -14.788$ ($R^2 = 0.6$; RMSE = 57.89 $Wm^{-2}$) and $H\_SPARSE_{t-FP} = 1.12 \ H\_XLAS_t -10.57$ ($R^2 = 0.63$; RMSE = 53.85 $Wm^{-2}$) for Terra and Aqua, respectively. This result is of great interest considering that the SPARSE model was run with no prior calibration. However, we noted that bias is a function of the flux level and most outliers are

recorded for H greater than 200 Wm$^{-2}$. This can be explained by (i) the XLAS measurement saturation (according to the "Kipp & Zonen LAS and XLAS instruction manual" (KIPP&ZONEN, 2007), for a path length of 4 km and a scintillometer height of 20 m, saturation measurement problem starts from H values higher than 300 Wm$^{-2}$), (ii) uncertainties on the correction of stability using the universal stability function and (iii) potential

inconsistencies between the area average MODIS surface temperature and the air temperature measured locally at the meteorological station.

Whereas there are several studies dealing with large aperture scintillometer (LAS) data whose measurements are compared to modeled fluxes, in the few studies dealing with extra large aperture scintillometer (XLAS) data, the comparison is generally done with Eddy Covariance station measurements (Kohsiek et al., 2002; Moene et al.,

2006). Indeed, our results are in agreement with those found by Marx et al. (2008) who compared LAS-derived and satellite-derived H (SEBAL was applied with NOAA-AVHRR images providing maps of surface energy fluxes at a 1 km × 1 km spatial resolution), and found that modeled H is underestimated with a RMSE of 39 Wm$^{-2}$ for the site Tamale and 104 Wm$^{-2}$ for the site Ejura. Moreover, Watts et al.(2000) compared the satellite (AVHRR radiometer) estimates of H to those from LAS over semi-arid grassland in northwest Mexico

during the summer of 1997. They found RMSE values of 31 Wm$^{-2}$ and 43 Wm$^{-2}$ for LAS path lengths of 300 m and 600 m respectively and showed that LAS measurements are less good than those derived from a 3D sonic anemometer. They also suggested longer LAS path length (greater than 1.1 km) since the LAS is rather insensitive to the surface near the receiver and the emitter.



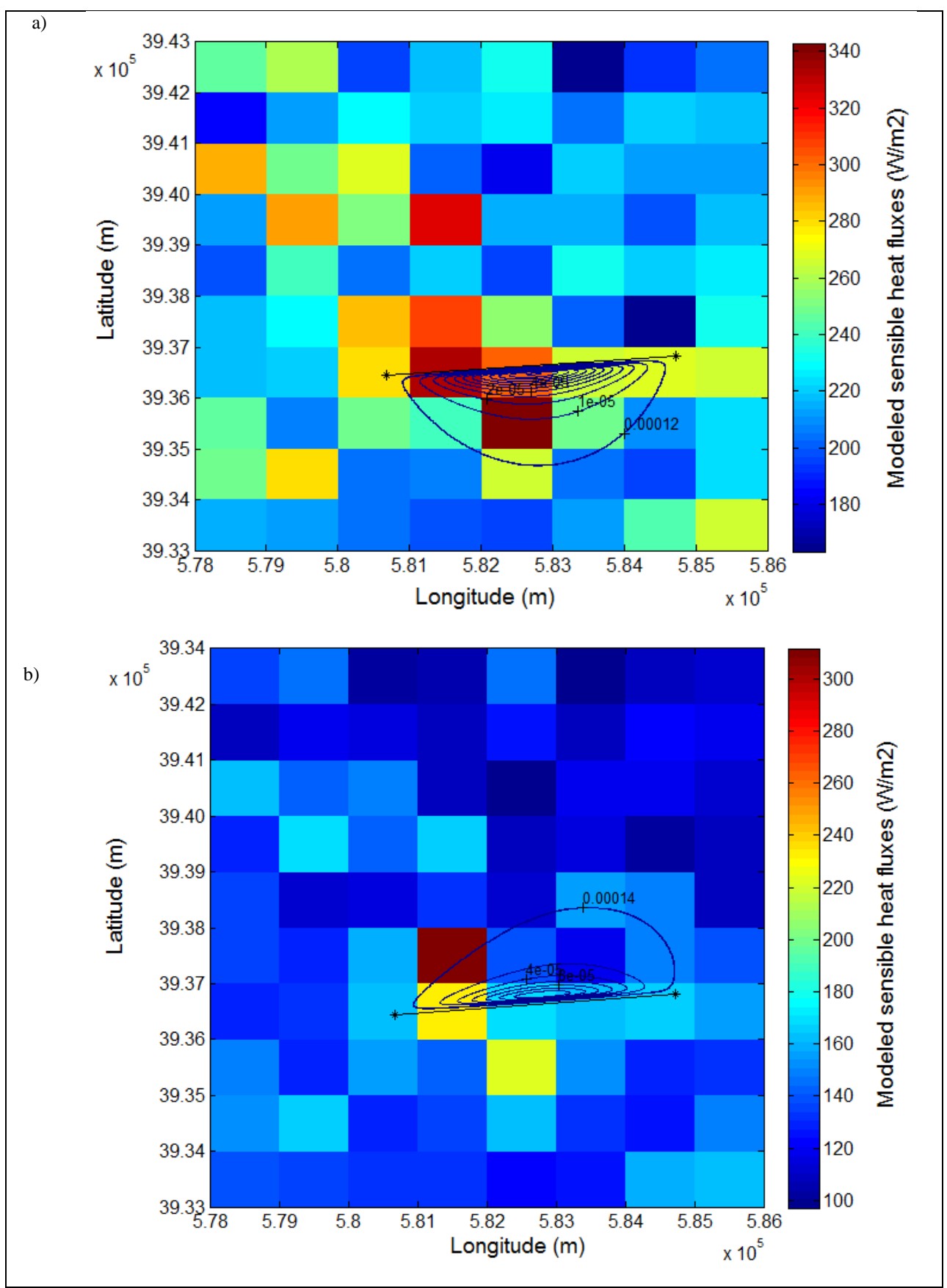

**Figure 6: Model derived sensible heat fluxes and footprints for (a) DOY 2013-082 at Aqua time overpass and (b) DOY 2014-208 at Terra time overpass. The colored area shows the modeled flux and the contours shows the surface source area contributing to the scintillometer measurements.**

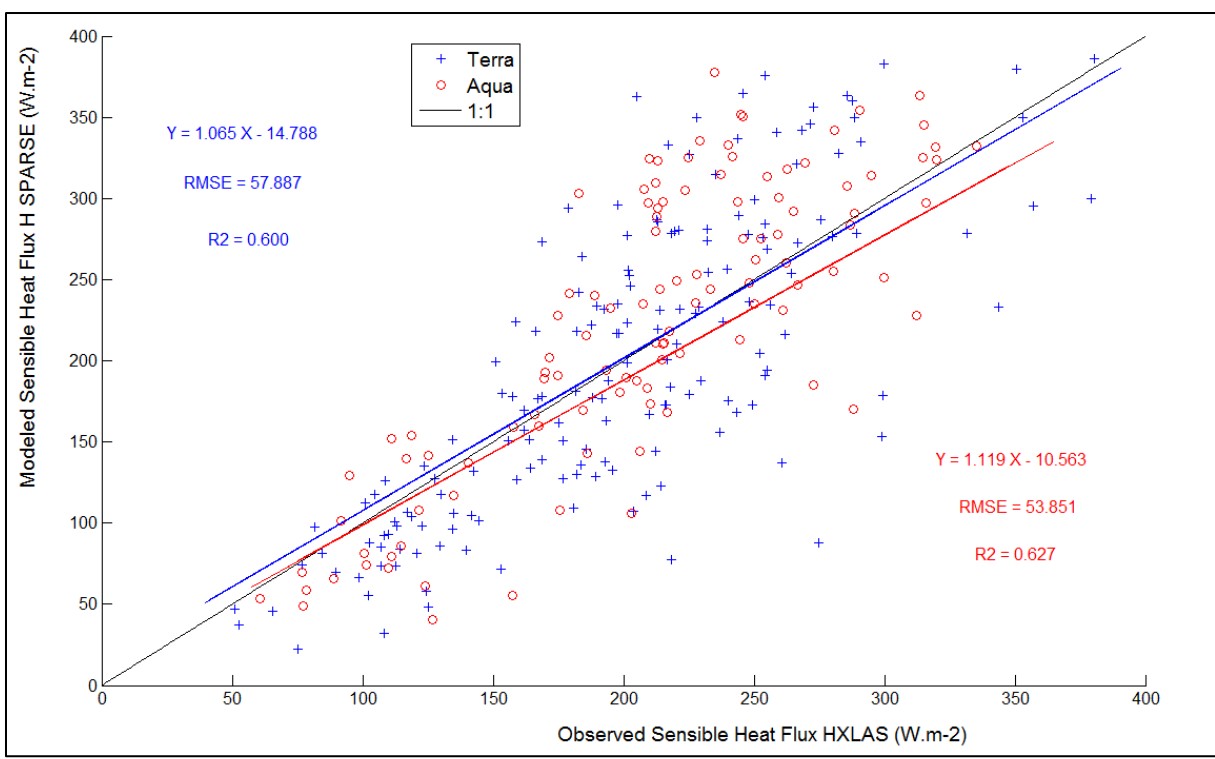

**Figure 7: Modeled vs. observed sensible heat fluxes at Terra and Aqua time overpass**

### 6.2 XLAS and model derived instantaneous latent heat fluxes

In a subsequent step, SPARSE derived LE (LE_SPARSE$_{t\text{-}FP}$) was compared to observed LE (LE_residual_XLAS$_{t\text{-}FP}$). Results are illustrated in figure 8 showing a good agreement between modeled and observed LE. However, these results are less good than for the H results, as shown by the linear regressions: LE_SPARSE$_{t\text{-}FP}$ =0.94 LE_residual_XLAS$_{t\text{-}FP}$ + 12.47 (RMSE = 47.20 Wm$^{-2}$) and

LE_SPARSE$_{t\text{-}FP}$ = 0.85 LE_residual_XLAS$_{t\text{-}FP}$ +11.51 (RMSE = 43.20 Wm$^{-2}$) for Terra and Aqua respectively,

with an overall $R^2$ of 0.55 for both satellites. We note a greater scatter for latent heat flux than for the sensible heat flux (Figure 7), which can be explained by the fact that LE is here a residual term affected by estimation errors in both AE and H. Despite this moderate discrepancy, the good agreement between both approaches indicates that the methodology adopted in SPARSE for estimating H and AE using MODIS imagery is appropriate for modeling latent heat fluxes.

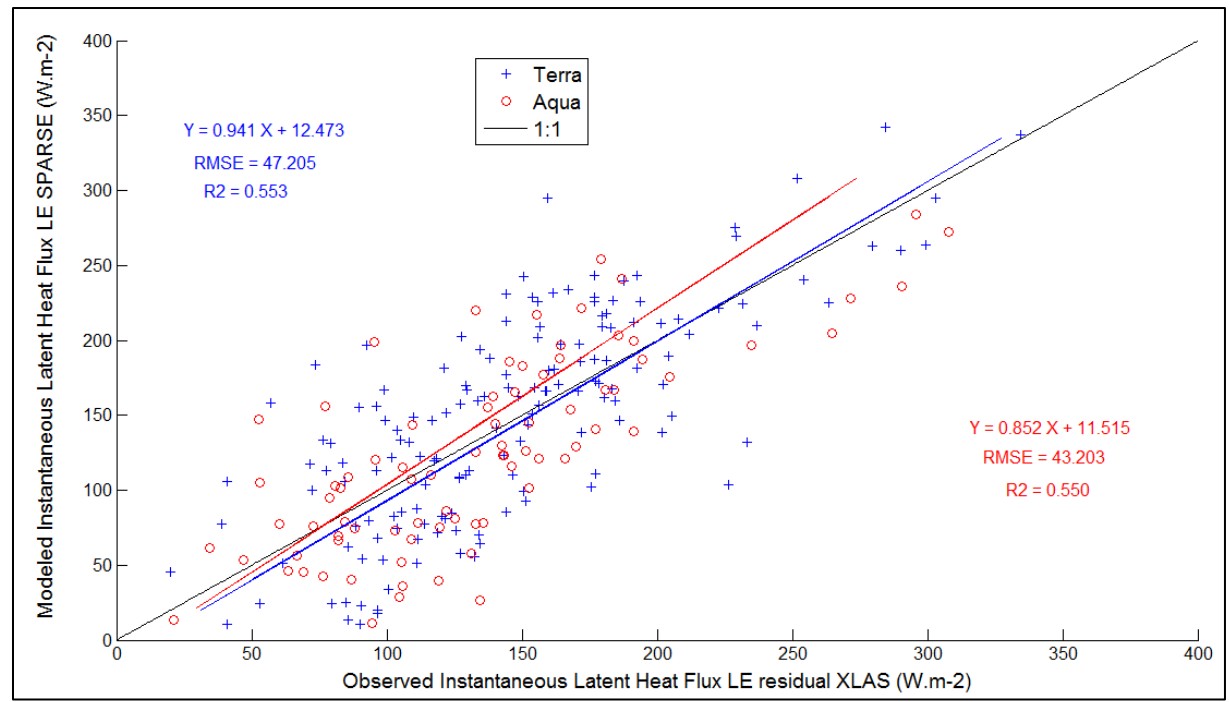


**Figure 8: Modeled vs. observed latent heat fluxes at Terra and Aqua time overpass**

### 6.3 Water stress

The scattered values of the Stress Factor as shown in figure 9 are consistent with previous studies such as Boulet et al. (2015). SEB retrieval of stress is limited by the scale mismatch between the instantaneous estimate of the
surface temperature during the satellite overpass (which can be influenced by high frequency turbulence) and the aggregated values of other forcing data which are derived from half hourly averages (Lagouarde et al., 2013; Lagouarde et al., 2015). However, general tendencies are well reproduced, with most points located within a 0.2 confidence interval (illustrated by dotted lines along the 1:1 line) as found by Boulet et al. (2015) at field scale, which is encouraging in a perspective of assimilating ET or SF in a water balance model for example. Moreover,
it is noted that results include small LE and $LE_p$ values having the same order of magnitude as the measurement uncertainty itself. Most outliers having greater water stress (~1) correspond to high evaporation from bare soil since the dominant land use in the study area is arboriculture, but also, this could be due to saturation of scintillation which led to an underestimation of H XLAS measurements as pointed by Frehlich and Ochs (1990) and Kohsiek et al. (2002).

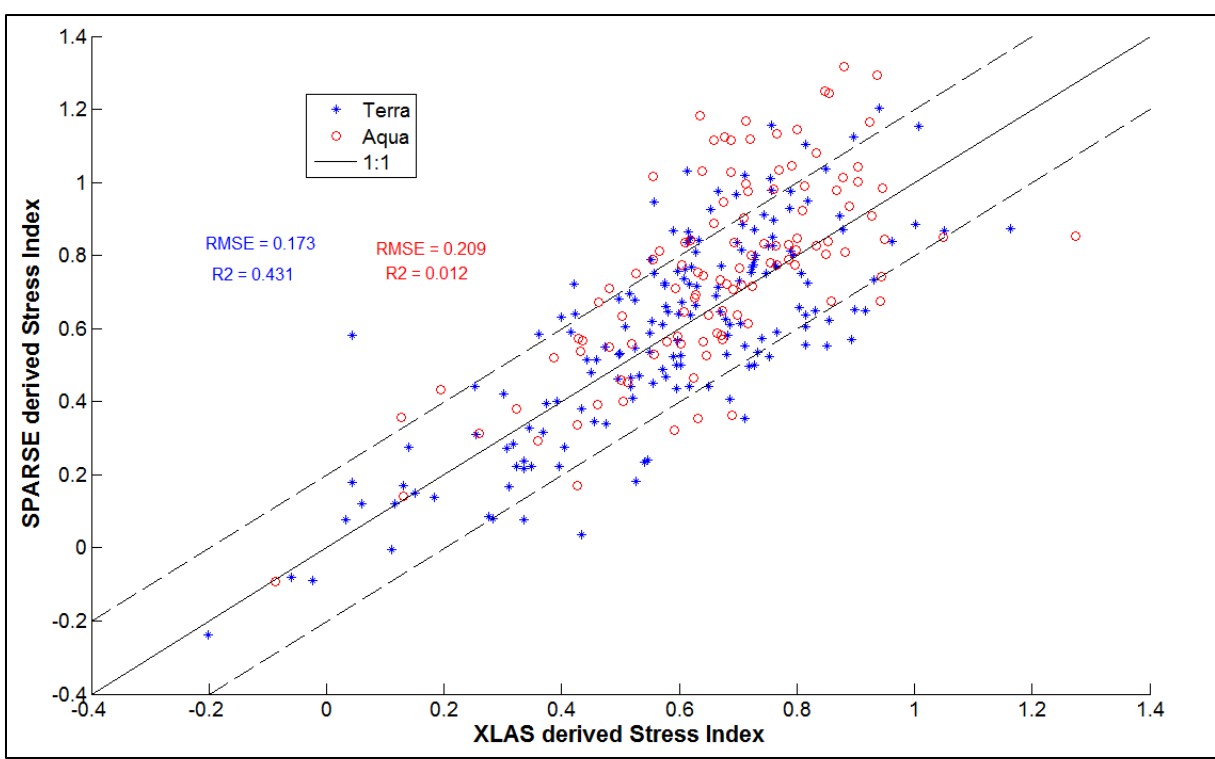


**Figure 9: Modeled vs. XLAS derived stress index SF at Terra and Aqua time overpass**

 Modeled and observed stress index at Terra and Aqua time overpass show a consistent evolution with daily rainfall (Figure 10), although the modeled stress show a greater dispersion than the observed one. During a rainy episode (or an eventual irrigation period), the surface temperature decreases towards the unstressed surface

temperature, thus marking an unstressed state, and SF tends to 0. Conversely, after a long dry down, the water stress appears and the surface temperature increases towards the equilibrium surface temperature computed by SPARSE under stressed conditions, and SF tends towards 1. Besides, it is noted that modeled stress indexes computed on the basis of Aqua MODIS's $T_{surf}$ are often greater than those computed used Terra MODIS's $T_{surf}$ due to higher $T_{surf}$ (higher global solar radiation) at the time of Terra overpass (around midday).


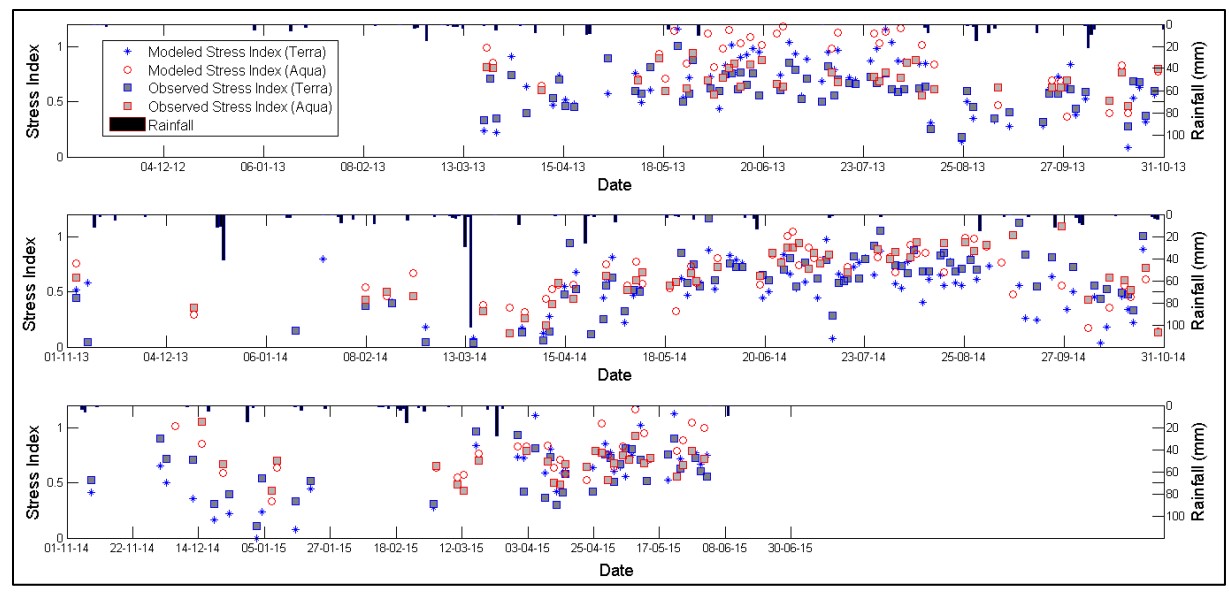


**Figure 10: Modeled and observed stress index evolution at Terra and Aqua time overpass compared to daily rainfall**

### 6.4 XLAS and model derived daily latent heat fluxes

Daily observed ET, *i.e.* LE_residual_XLAS$_{day\text{-}FP}$, was computed using the residual method; hence, six estimates of the daily observed ET were obtained by combining the two satellite datasets and three methods to compute G

and thus AE (see Sect. 3.3). Only the residual method was used to estimate daily observed ET for two reasons; on the first hand, to reduce the computations approach since, already, three methods to compute AE have been tested and on the other hand, the application of the EF method was not possible because we do not have a measured spatially distributed potential evapotranspiration (only point potential evapotranspiration data at the Ben Salem meteorological station are available). From daily observed ET estimates, minimum and maximum ET

were selected for each day and minimum and maximum daily ET time series were interpolated between successive days based on the self preservation of the ratio of AE to Rg as scale factor (Figure 11).

In addition, three methods were used to compute SPARSE daily ET for the Terra and Aqua overpasses (see Sect. 4.2), providing six estimates of the daily modeled ET. For each day average ET was plotted (260 days) with error bars figuring minimum and maximum values, along with precipitation to understand the rainfall impact on

the ET evolution (Figure 11).

Despite the uncertainty in reconstructing the daily ET from instantaneous ET, overall results show a good agreement between XLAS derived and SPARSE derived ET values with similar seasonal dynamics. Daily observed and modeled ET over the whole study period were both in the range of 0-4 mmday$^{-1}$ with an RMSE of 0.7 mmday$^{-1}$ which is consistent with the land use present in the XLAS path: mainly trees spaced by a

considerable fraction of bare soil, and less herbaceous soil-covering crops (see Sect.3.2). As expected, ET rates decrease significantly during dry periods (summers) since arid conditions limit the latent heat flux in favor of sensible heat flux and increase immediately after rainfall events due to the high amount of water evaporated from soil. The rainfall peaks that occurred on 3$^{rd}$ September 2013 (about 10 mm), 6$^{th}$ October 2013 (about 20 mm), 15$^{th}$ March 2014 (about 100 mm) and 22$^{nd}$ April 2014 (about 25 mm) are followed by well-reproduced

drydowns.

At seasonal scale, we note a good agreement between modeled and observed daily ET for the 2013-2014 and 2014-2015 seasons, especially when vegetation cover was more developed: from March to July 2014 and from

March to Mai 2015; these periods correspond to cereals vegetation peak in some plots (March-April) and to market gardening crops (e.g. tomato, water melon, pepper, etc.) cultivated generally from spring to the beginning of autumn in the interrow area of trees plots, which is a common farming practice in the Kairouan plain. However, the 2012-2013 season was dry compared with the two other ones, and less accurate results were obtained. Some points with little to null ET were recorded from May to July 2013 which can be explained by the very dry conditions and scattered vegetation cover with a considerable amount of bare soil. This behavior was not observed in the same period of 2014, because 2014 was a rainy year in comparison to 2013, therefore, even supposing that the farmers have the same attitude and cultivate the same crop types between the two years (which is not true in the context of our study area and farmers always change crop types), precipitations favor the growth of spontaneous vegetation over fallows which contribute to ET rise. On the other hand, since this year experiences more rain, farmers cultivate a larger part of the land and diversify the crop types; the vegetation cover is denser and contributes to an overall increase in ET. Overall, lower ET values are recorder in autumn (October and November) which correspond to evapotranspiration from trees only, since the latest summer crops (market gardening crops) have been already harvested and the winter crops (mainly cereals) are not yet sown.

Moreover, it can be seen that occasionally SPARSE overestimated ET. As example, three dates can be selected in August 2013 (15[th], 25[th] and 29[th] August 2013) for which modeled ET were 3.30 mm, 3.80 mm and 2.80 mm while maximum observed ET were 2.0 mm, 2.40 mm and 1.20 mm, respectively; broader amplitude between modeled (4.00 mm) and observed ET (1.40 mm) was also recorded on the 18[th] of May 2013. SPARSE also overestimates ET throughout ten days in August 2014 with an average difference of 1.1 mm and a maximum difference of 1.60 mm recorded in 23[rd] August 2014. These discrepancies are always recorded under wet conditions (minimum stress factor) which show the difficulty in representing accurately the conditions close to the potential ET. This might be related to the theoretical limit of the model for low vegetation stress especially when coupled with low evaporation efficiencies (*i.e.* dry soil surface) as already reported by Boulet et al. (2015) for senescent vegetation. Average difference between SPARSE and XLAS derived LE estimates when both are available indicate that SPARSE can predict evapotranspiration with accuracies approaching 5% of that of the XLAS.

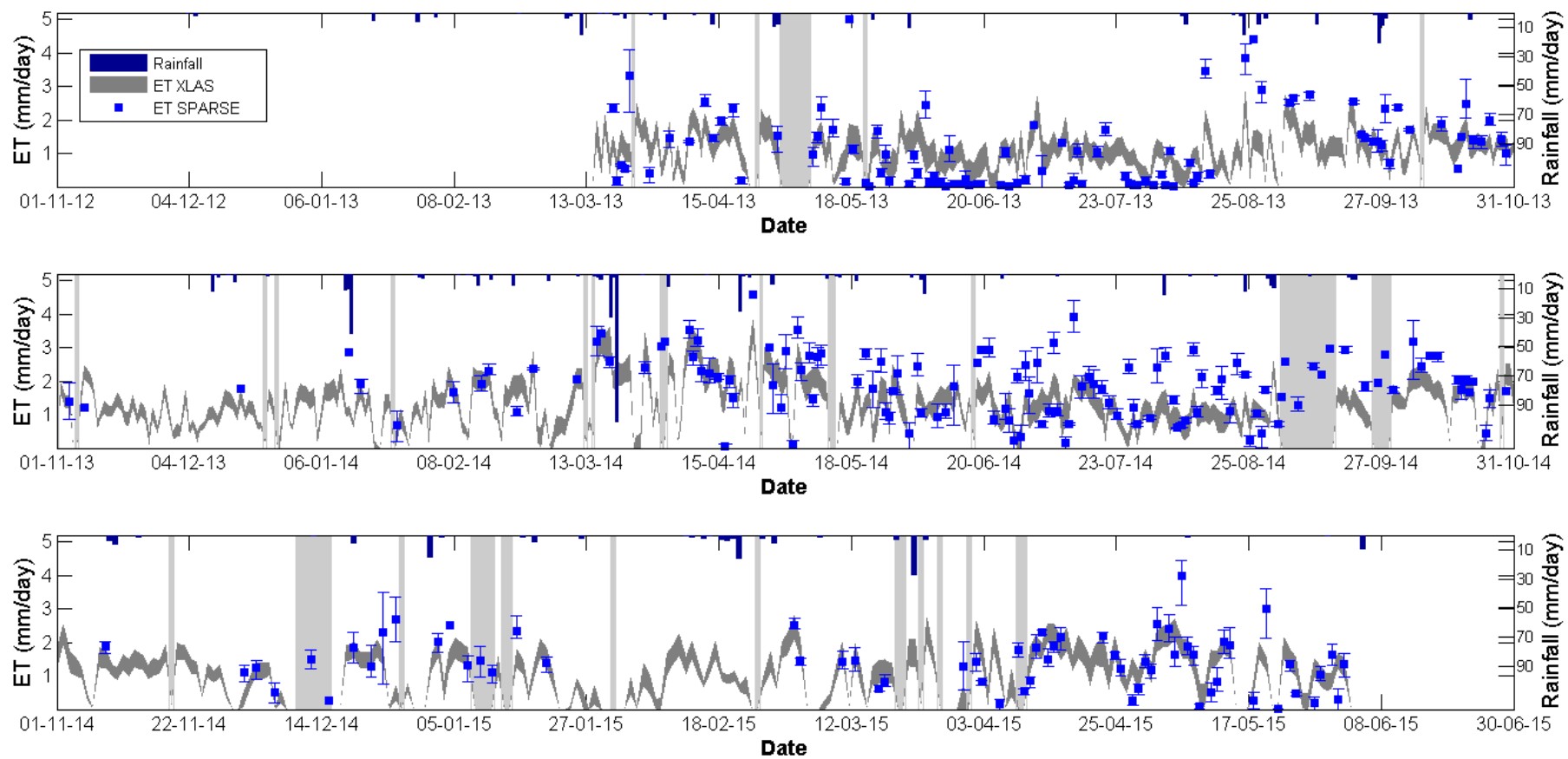

**Figure 11: Modeled vs. observed daily latent heat fluxes. Dark grey color shows minimum and maximum daily observed LE. Light grey vertical bars show gaps in XLAS data. Error bars for the modeled ET show the minimum and the maximum daily ET resulting from the three methods used to compute daily ET from instantaneous modeled ET.**

## 7    Conclusions

This study evaluated the performances of the SPARSE model forced by MODIS remote sensing products in an operational context (no model calibration) to estimate instantaneous and daily evapotranspiration. The validation protocol was based on an unprecedented dataset with an extra large aperture scintillometer. Indeed, up to our knowledge, this is the first work based on XLAS measurements acquired during more than 2 years, as compared to three months in previous works (Kohsiek et al., 2002; Moene et al., 2006). The estimates of the sensible heat

flux derived from the SPARSE model are in close agreement with those obtained from the XLAS. These results indicate that the XLAS can be fruitfully used to validate large-scale sensible heat flux derived from remote sensing data (and residual latent heat flux), in particular for the results obtained at the satellite overpass time, providing a feasible alternative to local micrometeorological techniques for measuring the sensible heat flux and validating satellite-derived estimates (*i.e.* eddy correlation). Furthermore, the extrapolation from instantaneous to

daily evapotranspiration is less obvious and three methods were tested based on the stress index, the evaporative fraction and the residual approach. The daily latent heat fluxes derived from the XLAS agreed rather well with those modeled using SPARSE model, which shows the potential of the SPARSE model in water consumption monitoring over heterogeneous landscape in semi-arid conditions, and especially to locate areas most affected by water stress. However, the precision in ET prediction with the SPARSE model is restricted by several

assumptions and uncertainties. For instance, the instantaneous remote sensing data and mainly $T_{surf}$ which is paramount in stress coefficient computation are assumed to be reliable. Moreover, there is an issue with the MODIS pixel heterogeneity and notably the distribution of components at the intersection between the square pixel and the XLAS footprint. Uncertainties are also due to half hourly forcing (meteorological and flux data) and XLAS data as well as to the extrapolation method from instantaneous to daily results. Furthermore, the

empirical estimation methods of soil heat flux G (three methods were tested) as well as the possible daily heat accumulation lead to possible errors in available energy estimation and in turn in residual LE estimation.

Even if overall results are encouraging, further work is needed to improve results by i) being most efficient in the SPARSE model application using calibrated input data specific to our study area, especially input parameters to which the model is particularly sensitive such as the mean leaf width and the minimum stomatal resistance, ii)

taking into account the heterogeneity of the 1km MODIS pixel by applying MODIS footprint, which is determined by the sensor's observation geometry and (iii) using a Land Surface Model applied at the field scale (Etchanchu et al., 2017) to analyze the scaling properties from the field to the footprint of the XLAS and the MODIS pixels similarly.

Finally, in a future work, we plan to take advantage of the complementarities between the Soil Water Balance

and Surface Energy Balance approaches (*i.e.* continuous but uncertain estimates using SWB due to poor soil water content control on one hand and sensitivity of SEB to the actual water stress on the other hand) to implement an assimilation scheme of the remotely sensed surface temperature into land surface models. In fact, in order to provide further information about distributed soil water status over the studied areas, the TIR-derived evapotranspiration products could be assimilated directly either in land surface or hydrological models.


**Author contribution:**

Sameh Saadi: data processing, data analysis and results interpretation.

Gilles Boulet: data analysis and results interpretation.

Malik Bahir: SPARSE inputs and XLAS data processing and analysis.

Aurore Brut: XLAS data processing and analysis.

Émilie Delogu: Daily evaporative fraction computation and analysis.

Bernard Mougenot and Zohra Lili Chabaane: site management.

Pascal Fanise: site instrumentation.

Vincent Simonneaux and Zohra Lili-Chabaane contributed with ideas and discussions.


**Competing interests:**

The authors declare that they have no conflict of interest.

**Acknowledgements**

The authors are thankful to the GDAs of Ben Salem I and Ben Salem II which enabled the scintillometer set-up
and access above the two water towers. Funding from the CNES/TOSCA program for the EVA2IRT project,
from the MISTRALS/SICMED program for the ReSAMEd project, from the ORFEO/CNES Program for
Pléiades images (© CNES 2012, Distribution Airbus DS, all rights reserved), and from the ANR/TRANSMED
program for the AMETHYST project (ANR-12-TMED-0006-01) as well as the mobility support from PHC
Maghreb program (N° 32592VE) are gratefully acknowledged,. This work has benefited also from the financial
support of the ARTS program ("Allocations de recherche pour une thèse au Sud") of IRD (Institut de Recherche
pour le Développement).

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
