# Peer review of "Assessment of actual evapotranspiration over a semi-arid heterogeneous land surface by means of coupled low resolution remote sensing data with energy balance model: comparison to extra Large Aperture Scintillometer measurements"

_Hydrology and Earth System Sciences, 2017_

## Referee Comment (RC1) · Anonymous Referee #1 · 7 Sep 2017

Interactive comment on "Assessment of actual evapotranspiration over a semi-arid heterogeneous land surface by means of coupled low resolution remote sensing data with energy balance model: comparison to extra Large Aperture Scintillometer measurements" MS No.: hess-2017-454. by Sameh Saadi et al.

The Authors present an extensive work (reinforced by experimental data) aimed to

assess the operational use of the Soil Plant Atmosphere and Remote Sensing Evapotraspiration (SPARSE) model and its accuracy by a comparison to the Scintillometric technique. I think that Authors address relevant scientific questions within the scope of HESS. Furthermore the paper is generally well organized and well written and therefore the paper could be taken into account for the final publication after a moderate revision. Particularly, The Authors should improve the part of "Results and discussion" (pag. 16-20) with a better description of the validation of SPARSE model carried out with by comparing H and AE estimations with flux station and XLAS scintillometer (see comments n°7, 11 and 12). My comments and questions are as follow:

1. Lines 33-44: The Authors corroborated "the good correspondence between instantaneous H estimates and large aperture scintillometer H measurements" reporting RMSE values expressed in W m-2. As stated by the Authors (Line 418) "For hydrological applications, daily ET is usually required. . . ." and in my opinion this means that for hydrological purposes the accuracy of daily evapotranspiration should be expressed in millimeters for day (mmd-1). Therefore in the abstract and through the paper this aspect should be considered and also critically analyzed. From my calculations the accuracy obtained by SPARSE model application should be around 1.6 mmd-1. Is this value "acceptable" ?

2. Lines 87-88: Is "irrigation requirements" (generally expressed in mmd-1) a prerogative only "of RS-based SWB models" ? Please, clarify.

3. Line 108: ". . .at the beginning of the process". Please clarify.

4. Lines 111-112: ". . .the lack of information about the actual irrigation scheduling adopted by the farmers is the critical limitation for SWB modeling". I believe that various SWB models (Swap, Cropsyst, FAO56, AcquaCrop) are able to consider both scheduled by farmer irrigation (as input) or predicted irrigation (as output). Please, clarify or modify.

5. Line 123: Insert ". . ." in dual-source models.

6. Lines 152-154: Clarify that the "layer" approach of SPARSE is essentially a "dual-source" scheme.

7. Line 187: The Authors should explain (also under a theoretical point of view) the choice to install Scintillometer at a 20 m height. About the experimental setup it is strange the absence of a "net radiometer" that, on the basis of the footprint analysis, could be installed in the average prevalent source area of footprint. The Authors could explain this fact.

8. Line 280: The terms "incoming solar radiation" and "incoming atmospheric radiation" are correct but could generate a misunderstanding. Please use the more classical "shortwave" and "longwave" terminology in eq. (9) and explain how RS data are generally used to solve balance equation of radiation (eq.9).

9. Line 367: About the "Temporal interpolation of albedo and NDVI" some brief details could be considered.

10. Line 455: Which method has been used to evaluate the "potential conditions", please clarify.

11. Lines 491-492: The Authors reported that ….."An overestimation of about 15% is found between estimated and measured daily available energy…..and the coefficients ……were applied to remove this bias". If I well understand the above procedure (remove of bias) is a sort of calibration of the output of modeled on the basis of observed flux station. Please clarify.

12. Lines 526-527: About the estimation of sensible heat flux the authors reported that "This result is of great interest considering that the SPARSE model was run with no prior calibration", but I feel a sort of contradiction with the bias removing procedure described in the above comment. Please clarify. Moreover I think that the Authors should describe the accuracy of model prior and after the bias correction.

13. Line 545: (Figure 7). Looking at the scatterplot it is clear a more dispersion for H

value greater than 150. Is there an explanation of this ?

14. Line 604: The Authors reported that "Daily observed and modeled ET over the whole study period were both in the range of 0-4 mm mm.day-1 which is consistent with the land use present in the XLAS pat". In my opinion this is a prosy comment, Trouble if not.

15. Line 616-617: The Authors reported that "Some points with little to null ET were recorded from May to July 2013 which can be explained by the very dry conditions and scattered vegetation cover with a considerable amount of bare soil". Why this behavior was not observed in the same period of 2014 ?

16. Line 863: Please check the (Minacapilli and Ciraolo, 2007) reference.

---

## Referee Comment (RC2) · Anonymous Referee #2 · 25 Sep 2017

**Review HESS**
Assessment of actual evapotranspiration over a semi-arid heterogeneous land surface by means of coupled low resolution remote sensing data with energy balance model: comparison to extra Large Aperture Scintillometer measurements
S. Saadi, G. Boulet, M. Bahir, A. Brut, B. Mougenot, P. Fanise, V. Simonneaux and Z. Chabaane

**Paper Summary**
This paper presents a multiscale study of evapotranspiration (ET) in semi-arid area of central Tunisia where agriculture production is limited by water availability. The analysis involves ecosystem fluxes obtained by means of eddy-covariance (EC) instrument and landscape fluxes through XLAS and satellite remote sensing. These observations have been combined to produce large-scale area-average estimates of ET and sensible heat fluxes through the use of a vegetation model driven by remote sensing input SPARSE. Results in terms of divergences in H and LE estimates between model and instrumentation are above 50 W/m2.
Considering the uncertainties and limitations bounding the present ET estimates on heterogeneous terrain; water stress was estimated within reasonable ranges.

**Review Summary**
This paper presents interesting results combining area-average large-scale observations of surface fluxes and modeling tools. Results in terms of the uncertainties on LE and H are within what has been observed on intensive field experiments in complex or perhaps, more precisely, highly complex environments as compared to the present one.
**This review finds this paper border line between rejection and major reviews.**
To substantiate this evaluation, this review finds an important number of instances in which the paper is not clear or need to be corrected or needs to provide better justification as well as improve clarity. Since the number of corrections found may take an important amount of time for the authors to resubmit that's why I'm giving the editor the options of rejection (with resubmission) or major correction. The results are definitively important but they need a much improvement.
Depending upon editor's decision I would like to see further:
1) Figures with better accuracy in their representation. For example, some of them seems to have been the result of quick spreadsheet plots but without including accurate axis ticks, grids, labels, etc.
2) Same as for the description of the figure captions and legends. The reader needs to understand a given figure by analyzing the figure and reading the information on the figure caption and legends.
3) A better explanation of the SPARSE methodology is needed, steps and the set of equations in the ET and H estimates. What the assumptions are and what is the physical framework? All of that is missing and therefore theoretically this paper is very weak. For example, from where the authors got a threshold value of 30 W/m2 to start the iteration? How convergence is achieved is a mystery here and how many iterations and how signal-to-noise ratio of RS data plays a role in that convergence? Which equation provides convergence we don't know.
4) I would like the authors to provide adequate justification to the use of formulas to deduce H based on LAS or XLAS. Particularly since the indicated formulas are valid only under the similarity hypothesis of Monin-Obukhov which implies homogenous surface and stationary

flows. No justification was provided as for how these conditions were tested to render valid the resulting HLAS flux.

5) when the authors discuss about uncertainties it is not clear what kind of uncertainties we are talking about and how have those been calculated? Moreover, uncertainties in heterogeneous terrain based on pure observations XLAS have not been computed. A reference is provided so that the authors can check on that.

6) Not clear where the EC flux comes into play. Also footprint functions for the scintillometers need to be accounted for. Reference on this element is provided below.

7) I would like the authors to provide an in-depth description of physical processes explaining the results in the final figures. Description of what is being presented in the figures is fine but we need more science here.

As an aside note the use of XLAS is not unique in this problem. A LAS can do 5 km max. optical beam path and resolve the same situation. What is critical with using XLAS is beyond 5 km optical path.

Bai, et al., 2015. "Characterizing the Footprint of Eddy Covariance System and Large Aperture Scintillometer Measurements to Validate Satellite-Based Surface Fluxes. *Geoscience and Remote Sensing Letters, IEEE,* 12(5), 943-947, 2015. doi: 10.1109/LGRS.2014.2368580.

**Comments in details**
Line 45 –off : please put references in chronologic order. This is the proper way to recognize previous work; unless specific discussions are provided which in those cases the trail of references needs to be broken down. This note is valid through the entire paper.

Line 50: About the claims about water scarcity related to climate change. -or better say climate variability: I wonder how compelling are these claims? – Can the authors substantiate in more details about this problem in this area? This is an important claim and need to be fully addressed by the authors to build context to this research and the methodologies being used.

Line 53: the use of "greatest" here tries to indicate what? "the larger" or "the most important"? This needs to be clearly understood without ambiguity and therefore we need to bring more specificity.

Line 56: I'll add complexity in the hydro-meteorological processes. As we move from ecosystem scale to landscape scales surface heterogeneity but also dynamic of the flow, cloudiness, precipitation come into play more aggressively. This also bring more context to the need of this study.

Line 61: I would disagree that "RS techniques becomes essential". Basically it has been demonstrated that plot (or ecosystem) exchanges within same complex canopies do verify consistent differences in sensible heat fluxes (the simplest and ubiquitous flux on earth) over distances that are much smaller than the RS footprint in particular MODIS. See Starkenburg et al., (2015).

Starkenburg et al. 2015: "Temperature regimes and turbulent heat fluxes across a heterogeneous canopy in an Alaskan boreal forest". *J. Geophys. Res. Atmos.*, 120: 1348–1360.
doi: 10.1002/2014JD022338

Now, I do agree that RS brings a mean to deduce, within certain ranges, an approximation of fluxes. What about mesoscale models? Or perhaps you wanted to indicated physical models using RS data as input? In any case, I think you should open this perspective here since there are other disciplines other than Remote Sensing Researchers that can also provide the same product.

Line 63: vegetation physical properties or characteristics?

Line 65: Authors use "plot" as one of the scales in which I assume results would be obtained. However, at no point plot-scale was defined. Please whenever plot is used for the first time in the Introduction section for example please clarify that. (excluding the abstract).

Line 87: please rephrase the text between parenthesis.

Line 93: Spell out FAO. If it is not being used anymore in the text, then no need to define an acronym.

Line 98-99: get rid of parenthesis here. What is inside is part of the phrase.

Line 102: FAO-56 put a reference here. Or make a short phrase explanation.

Line 103: what is "dry down"? please make sure you check consistency in all phrases.

Line 114: What's the meaning of adding quotes here? If single-source means single source, then no need for quotes. Quotes are used when you use a word or combination of words but you would like to indicate a different meaning.

Line 116: same as 114.

Line 117: comma missing before etc.

Line 128: add "they provide area-averaged sensible heat flux"

Line 130-131: incomplete phrase. And, can you elaborate a little bit more here?

Line 132: delete space before comma.

Line 133: representative of the pixel? It may be the case that for a particular MODIS data your scintillometer data intersects several pixels. Then we are talking about several pixels.

Line 140: **large-scale area-average** this is the proper measurement that one obtains from a scintillometer.

Lines 140-143: Here I need help. Are you indicating that to get ET large-scale area-average you use XLAS? But you need to assume a closure fraction or assume is 100% Energy Balance closure. As we increase surface heterogeneity and the atmospheric flow acquires an increased space-time variability then it is difficult to assume 100% energy balance closure. How you do then? Please explain how you treat and eventually circumvent this problem.
See for example Foken et al., (2006; 2010) and Foken (2008).
Foken, T., F. Wimmer, M. Mauder, C. Thomas, and C. Liebethal, 2006. Some aspects of the energy balance closure problem. *Atmos. Chem. Phys.,* 6, 4395–4402.
Foken, T., 2008: "The energy balance closure problem: An overview", *Ecol. Appl.,* 18(6), 1351–1367.
Foken, T., M. Mauder, C. Liebethal, F. Wimmer, F. Beyrich, J.-P. Leps, S. Raasch, H. A. R. DeBruin, W. M. L. Meijninger, and J. Bange, 2010: "Energy balance closure for the LITFASS-2003 experiment", *Theor. Appl. Climatol.*, *101*(1-2), 149-160, doi: 10.1007/s00704-009-0216-8.

Line 146: what is the "layer" approach? Can you be more explicit and detailed? If layer is the name of the approach, then no need to use quotes.

Line 147: when authors normally explain the use of electrical resistance as equivalent models really are not paying attention to the details. So then now you need to explain how you transform an electrical element such as a Resistor, which is a concentrated parameter into a distributed vegetation or soil representation. What are the assumption? Hypothesis? Regions where this approximation is valid and where it fails, etc. I'll give you a hint R=V/I where V(electrical voltage: what is imposed the potential) and I(electrical current, what flows between the boundaries). Then when you say you use Rsoil and Rveg. What are the analogs of V and I here? What R actually means? And how you walk out from the Ohm's Law for concentrated electrical parameters and transition to our problem where these parameters are distributed?
This comes from Norman and Kustas TSEB- way before SPARSE.
For example, here it is important to remark that vegetation information has to be at much higher resolution than the radiometric information to account for vegetation/forest variations for example the existence of clear areas within the forest or cultivars. How the authors account for that needs better explanations. And, what assumptions underlain these approximations?

Line 150: I wanted to be clear here that XLAS ONLY can deduce sensible heat not LE. Please make sure this thread is conveyed all the way through your work.

Line 158: put "(" to indicate the reference the cultivars are within the phrase.

Line 173: what "double device" means for you. Please be specific.

Figure 2: it is not clear where the XLAS emitter and receiver are specifically located. Put a dot or a symbol to indicate that. Photos actually say nothing here. Now I see that the CSAT is close to the XLAS receiver. I would caution the authors here that any interpretation between XLAS fluxes and EC-CSAT fluxes would not be representative since the EC system is closer to the XLAS receiver and/or transmitter for that matter is the same.
More importantly what is not clear here is what are the green contours indicating the footprint? And if these are EC footprint more likely are wrong.

Please specify what SPOT5 bands 1,2,3 are in terms of wavelengths and they are used in this work.

Line 196: I would write Extra Large Aperture Scintillometer (XLAS)

Line 198: Phrase: "Scintillometer is based on the scintillation method" what is this?

Line 198-200: What is the cause and what is the effect? This phrase is wrong please think about a little bit.

Line 205: replace "bean" by "beam"

Line 204: The reference that links scintillations and Cn2 is given by Tatarskii. We need to give the proper reference here. The fact that those references have been using it doesn't mean they were the ones given the foundation for this relationship. We need to make sure we give proper value to the actual references.

Line 206: symmetrical to what? What is that symmetry you are talking about?

Line 208: get rid of an extra space in the phrase.

Same line: "structure parameter of temperature" by structure parameter of temperature turbulence (refractive index in the case of CN2).

Line 210-212: here the authors mentions very cursory a very important problem which is the variation of Cn2 because of the beam height variation across the landscape. It seems this is one point you should be more cautious in bring some references and eventually limit your study on the basis of this sensitivity parameter.

Line 213: only sensitive to temperatures. Add a period in the phrase.

Eq. [1] you introduce here an approximation that then you'll use as an equality. Please explain and substantiate or directly correct the equation. Also, I wonder how much beta introduce error, in this case, a semi-arid environment.

Line 217: iterative methods have intrinsic convergence and resolution errors. You have to specify the convergence error and also how the average of Cn2 gives you a signal with enough SNR to keep the specific convergence factor. Now recently analytical methods have been developed that integrate the set of nonlinear equations in this casa Tatarskii and Monin-Obukhov similarity hypothesis set. See Gruber and Fochesatto, (2013).

Gruber M. A. and G. J. Fochesatto. 2013: "A New Sensitivity Analysis and Solution Method for Scintillometer Measurements of Area-Average Turbulent Fluxes" *Boundary-Layer Meteorology*, 149:65–83 DOI 10.1007/s10546-013-9835-9

Line 220: Zlas is a function where is that?

Andreas parameterization might not be valid for your site.- Can you justify here?

Zv: is the average canopy height but weighted by the extension of the plots?

Eq. [4] contains u* but it is not clarified here from where this is taken.

Here we can conclude that XLAS ONLY measures T* as a large-scale area-average variable but u* is a local variable or at least a variable measured at the scale of the EC system which is not the same as the XLAS. Explain please?

Line 225: rho is the air density and cp here are considered constants. Do they vary across the experiment?

Line 227: nomenclature is Number[space]unit. please correct all the way your text.

Line 228: change "circa" by "near". The correct use of "circa" in English is to indicate something that happened in the past (circa, 1000 AD) for example.

Line 230: how many "aberrant" values you have in the entire dataset. Please give more precision to the signal processing so that researchers can compare their work with yours in the future.

Line 247: and also gives the major sensitivity to H. See also (Gruber et al., 2014) for the specific analytic derivation of the sensitivity to the topography height.

Gruber, M. A., G.J. Fochesatto, O.K. Hartogensis, and M. Lysy. 2014: "Functional derivatives applied to error propagation of uncertainties in topography to large-aperture scintillometer-derived heat fluxes". *Atmos. Meas. Tech.*, 7, 2361-2371, doi:10.5194/amt-7-2361-2014, 2014.

Equations 7 and 8: assume closure of energy balance at 100% please explain how this is possible. And what are your assumptions that lead to this approximation and what is the uncertainty in this assumption.

Line 271: Here the authors give an estimation of G/Rn energy partition that is known to be variable not only across a given landscape but also across landscapes. This needs to be carefully estimated. This goes from 31% to very low values in dense canopies. Please be more specific and give values of this factors across all your landscapes.

Line 284: change "meteo" by "meteorological station".

Lines 280-290: Here the authors bring parameterizations of G. And certainly it is appreciated this compilation. However, it would be best to have a discussion of how one of these parameterization is or may result more optimal for this work. It seems all the formulas were found and then tossed in this article to see what happens. –So compare your environment with the environment in which those parameterizations were developed and then decide or make some arguments about how to best use or adapt any of these parameterizations.

Line 294: basically with the current satellite technology we cannot estimate diurnal cycles. However, you must know that at higher latitudes Aqua and Terra have at least six-passages a day.

Line 300: I don't understand why the authors propose a=1 and b=0 and then find motivation on

finding that actually these are not zero. The approximation of Rn by SW (Short Wave Downwelling) is known in micrometeorology and only works to some extent in clear skies when Rn is dominated by SW downwelling. I mean Rn can be negative but never SWdown. So, the way this paragraph is written possess a problem since it is not physically correct.

Line 304: How you weigh the 10x8 km images data by the footprint? What kind of functions are used here to compute the footprint. Please explain.

Line 310: replace the "temperature of soil" by "soil temperature".

Here you mention a "reference height" and simultaneously we are talking about a heterogeneous canopy and soil and canopy. Where is that reference height? And what are the assumptions and approximations you are taking by taking this assumption. For example, you are considering some variables at soil level but others at canopy level. How the reference height represents both? And what are the assumptions in terms of physical processes?

Eq. [15] you have here a radiative balance equation where it is assumed (without indication) that emissivity (on the left hand side ) is =1. Also this equation needs a reference level and a specific condition for the fluxes to be added and represented at the reference level. Please make sure you are accounting for all these so that the reader can fully understand what your assumptions are and where and under what conditions your analysis is valid.

Line 319-320: is SPARSE better than TSEB? Can you give a little bit more explanation here? TSEB has modes to trait vegetation ALEXI and DIS-ALEXI. Are you saying that by incorporating aerodynamic functions makes SPARSE better than TSEB? Please clarify here what's the extent and implication of your comment on the paper.

Line 325: from where you got the 30W/m2 minimum value? In some environments this will be three times G. Please justify this value.

Line 334:335: Here we need to be more specific. What data is from bibliography and what data comes from RS? Please be specific.

Line 343: Why you define an acronym MRT that is not used anymore? Acronyms that are not mentioned in the text anymore are unnecessary.

Line 343-347: this phrase is too long and badly constructed.

Line 349: We need more detail here. How many days or cases have been excluded from the entire dataset. We need to know how critical is this problem. Because if it is critical then it renders the method useless.

Line 355: k1.15 need space.

Line 357: explain clump-LAI measurements.

Delete the word "Bibliography" from Table 1. That column is for sources and a journal peer review is a source.

Line 379: "overpasses"

Line 383: The second step need a more substance. How come you are running a 30 min fluxes based on a single TIR input? This will result in diurnal cycle of fluxes that are totally biased. I would say that this approximation is only valid for time-intervals in which the turbulence conditions are not too different form the TIR observations.

Line 396: please revise the following wording "…complementary part to 1…"

Section 4.2 seems to go around and around the subject without going down to the specifics. I think is necessary to simplify the description of methods.

Line 407: how you define the wet conditions here? Rain through the day, a specific amount of mm? please be more specific here.

Eq. [21] assume 100% energy balance closure. You need to justify the use of this condition.

Line 429: "deduce" instead of "deduct".

Fig. 5. This figure is a very low quality without precision in the axis. Also we see only RS data here while it is announced XLAS data.

Line 475: "convolving" Convolution has a very specific meaning in mathematics. Please verify the use of this term here.

Same for the use of modelled or modeled. Both expressions are fine however if your choice is to use words in British English (in this case modelled) you have to be consistent all the way through your paper.

Line 477: "dots"? seriously?

Line 478: Why these two days? Please give the reasons why you are specifically using those days. This is important because when scientist reading your paper would like to reproduce your results they will find no framework to produce such comparisons.

Figure 6. I don't understand the coordinates (Y-axis and X-axis). Also the contours of XLAS footprint have no indications.

Line 482: what you mean by "hot pixel"? Please avoid jargon in the writing.

Line 489: In general models are calibrated based on EC systems and thus the deduced large-scale area-average fluxes derived from satellite remote sensing is controlled by LAS observations.

Line 490-500: In general, as the heterogeneity in vegetation, soil and eventually in topography leading to variables flows increases the divergence increases. There though cases in which even EC systems that are placed together at distance shorter than the convective ABL development verify more than 50/m2 differences (Starkenburg et al, 2015). So then results expressed here are within the range of reasonable values.

The only one physical explanation why the LAS path by being longer would give different results is when the heterogeneity is such that the BL that develops integrates patches of different thermodynamic and turbulent properties. Then, the mention of issue is interesting but without a correct explanation is useless.

Figure 7. contains features that are important to discuss since there is a change in the bias as function of the flux level. I wonder the authors to discuss this aspect from the physical aspects of the processes dominating this scale integration.

Figure 10. display several cases where there is a huge divergence in stress index particularly in April and July for both spacecraft.

Line 562: here the authors mentioned –uncertainties- but at no point in the paper we are discussing about this. As previously mentioned uncertainties come not only in EC and XLAS observations but also in the approximation used based on 100% closure in the energy balance. It is confusing and not clear definitively.

Line 565-570: give some explanation but actually is a description of the time-series. Can you provide a real-actual-explanation about what is the physical processes underlining this divergences and convergences.

Same from 570 to 575

Line 588: is this the actual explanation of why there is such divergence or is this another speculation?

Line 590-592: the error indicated here is extremely low now can you please indicate all-conditions in which this is valid and please circumvent this result to the specific interval of conditions in which this is actually valid.

Figure 11. From where and how you got errorbars in blue trace? Figure caption is not clear. We need a accurate description of the contents in the figure.

Line 610: "valorize" I wonder what the authors wanted to indicate here?

SVAT seems not to have been defined earlier.

---

## Author Comment (AC1) · 3 Oct 2017

Interactive comment on "Assessment of actual evapotranspiration over a semi-arid heterogeneous land surface by means of coupled low resolution remote sensing data with energy balance model: comparison to extra Large Aperture Scintillometer measurements" MS No.: hess-2017-454. by Sameh Saadi et al.

The Authors present an extensive work (reinforced by experimental data) aimed to assess the operational use of the Soil Plant Atmosphere and Remote Sensing Evapotraspiration (SPARSE) model and its accuracy by a comparison to the Scintillometric technique. I think that Authors address relevant scientific questions within the scope of HESS. Furthermore the paper is generally well organized and well written and there-fore the paper could be taken into account for the final publication after a moderate revision. Particularly, The Authors should improve the part of "Results and discussion" (pag. 16-20) with a better description of the validation of SPARSE model carried out with by comparing H and AE estimations with flux station and XLAS scintillometer (see comments n 7, 11 and 12). My comments and questions are as follow:

1. Lines 33-44: The Authors corroborated "the good correspondence between instantaneous H estimates and large aperture scintillometer H measurements" reporting RMSE values expressed in W m-2. As stated by the Authors (Line 418) "For hydrological applications, daily ET is usually required: : :." and in my opinion this means that for hydrological purposes the accuracy of daily evapotranspiration should be expressed in millimeters for day (mmd-1). Therefore in the abstract and through the paper this aspect should be considered and also critically analyzed. From my calculations the accuracy obtained by SPARSE model application should be around 1.6 mmd-1. Is this value "acceptable" ?

*Response:*

*Indeed, we agree with Reviewer 1 that for hydrological purposes the accuracy of daily evapotranspiration (ET) should be expressed in millimeters for day, however, mentioned RMSE in the abstract and through the paper are instantaneous sensible and latent heat fluxes estimates at the satellite overpass time and are not daily values, therefore, they are expressed in W.m-2. Since, they are instantaneous data, it should not be converted using this formula:*
*47.2 W.m-2\*0.0864/2.45= 1.66 mm/day*
*43.2 W.m-2\*0.0864/2.45= 1.52 mm/day*
*Therefore, we get an instantaneous LE error of about 0.1 mm/0.5.hour around the satellite overpass (around midday, at the max. ET rate)*
*Later (section 6.5), when dealing with daily ET, all values are in mm.day$^{-1}$, indeed, the model daily ET estimates accuracy (RMSE= 0.7 mm/day) will be added as it was mentioned for instantaneous results.*

2. Lines 87-88: Is "irrigation requirements" (generally expressed in mmd-1) a prerogative only "of RS-based SWB models" ? Please, clarify.

*Response:*

*Irrigation requirements are mainly estimated using RS-based SWB models, since irrigation is a component of the water balance equation on which is based SWB models. Indeed, the crop coefficient method (FAO56 method) is currently the main method used for scheduling irrigations around the world (Glenn et al., 2007).*
*Irrigation requirement was rarely directly estimated using SEB models. Indeed, SEB outputs are generally actual evapotranspiration (its energy equivalent LE) and if Irrigation is estimated, it should be computed as a residual term of the water balance equation. Exception exists, for example, (Courault et al., 1998) used surface temperature derived from NOAA data and a SVAT model called MAGRET to find parameters linked to the irrigation over the agricultural region "la Crau" in South-Eastern France ; the predicted parameters were the beginning and the end of irrigation, frequency and water quantity diverted.*

3. Line 108: ": : :at the beginning of the process". Please clarify.

*Response:*

*I did not find this expression in line 108 of the last article manuscript version "hess-2017-454-manuscript-version3_discussion" to which I am referring. Indeed, in this last version, the mentioned phrase was rectified as follows: "at the beginning of the dry down".*

4. Lines 111-112: ": : :the lack of information about the actual irrigation scheduling adopted by the farmers is the critical limitation for SWB modeling". I believe that var-ious SWB models (Swap, Cropsyst, FAO56, AcquaCrop) are able to consider both scheduled by farmer irrigation (as input) or predicted irrigation (as output). Please, clarify or modify.

*Response:*

*Indeed, several SWB models such as Swap, Cropsyst, FAO56, AcquaCrop and also the SAMIR model that we have already used (Saadi et al., 2015) are able to consider both methods to take irrigation into account: either an estimated amount provided by the farmer (as an input) or a predicted irrigation with a module to trigger irrigation according to, say, critical soil moisture levels (as an output). We have to clarify this part by saying that the lack of actual irrigation scheduling information does not impact the irrigation estimation by these models, since irrigation could be simulated by SWB models, but rather the validation protocol of irrigation requirements estimates (irrigation data is usually unavailable).*

5. Line 123: Insert ". . ." in dual-source models.

*Response:*

*In the version to which I am referring this expression is already put in inverted commas (line 116):*
*"However, separate estimates of evaporation and transpiration makes the "dual-source" models more useful for agrohydrological applications*

6. Lines 152-154: Clarify that the "layer" approach of SPARSE is essentially a "dual-source" scheme.

*Response:*

*The paragraph will be modified accordingly:*

*"In this study, (…) were obtained by the SEB method, using the "layer" approach (a resistance network that relates the soil and vegetation heat sources to a main reference level using a series electrical*

*branching) of the "dual-source" model; Soil Plant Atmosphere and Remote Sensing Evapotranspiration (SPARSE) (...)."*

7.      Line 187: The Authors should explain (also under a theoretical point of view) the choice to install Scintillometer at a 20 m height. About the experimental setup it is strange the absence of a "net radiometer" that, on the basis of the footprint analysis, could be installed in the average prevalent source area of footprint. The Authors could explain this fact.

*Response:*

*The choice to install Scintillometer at a 20 m height was based on the XLAS installation principle detailed in the "Kipp & Zonen LAS and XLAS instruction manual", indeed, the minimum installation height of the XLAS as function of the path length and for different surface conditions is graphically explained and shows that for a path length of 4km, the XLAS height of 20m is an adequate height since the XLAS is high enough to minimize measurement saturation and not too high to be representative of the 4km path Boundary Layer.*

*The absence of a "net radiometer" is explained by the high heterogeneity of the study area vegetation cover; therefore, it is not possible to measure the net radiation (Rn) of all plots or even the Rn of enough "typical" plots (with similar land cover and irrigation practice).This will be explicited in the revised version.*

8. Line 280: The terms "incoming solar radiation" and "incoming atmospheric radiation" are correct but could generate a misunderstanding. Please use the more classical "shortwave" and "longwave" terminology in eq. (9) and explain how RS data are generally used to solve balance equation of radiation (eq.9).

*Response:*

*Indeed, in equation (9) the terms "incoming shortwave radiation" and "incoming longwave radiation" are used. This terminology should be also used along all the manuscript, hence, sections 3.3.2, and 4.2.3 should be corrected. This paragraph will be added accordingly:*

*In Eq. 9, Land surface temperature (LST), surface emissivity and albedo are generally taken from remotely sensed data, whereas, incoming shortwave radiation Rg and incoming longwave radiation Ratm are meteorological data.*

9. Line 367: About the "Temporal interpolation of albedo and NDVI" some brief details could be considered.

*Response:*

*Albedo MODIS products (MCD43) are available every 8 days and come from different satellite overpasses over a period of 16 days, the day of interest is central date.. Both Terra and Aqua data are used in the generation of this product, providing the highest probability for quality input data and designating it as an MCD, which means Combined product.*

*NDVI MODIS products (MOD13A2/MYD13A2 for Terra and Aqua, respectively) come from different satellite overpasses over a period of 16 days, and they are available every 16 days and separately for Terra and Aqua. Indeed, algorithms generating this product operate on a per-pixel basis and requires multiple daily observations to generate a composite NDVI value that will represent the full period (16 days), the 1km/16days MOD13A2 (respectively MYD13A2) product is an aggregated 250m/16 days MOD13Q1 (respectively MYD13Q1) product..*

*For both products, the data is linearly interpolated over the available dates in order to get daily data. For each pixel, the best data is taken into account (based on the quality index supplied with the product). Therefore, the temporal interpolation was done pixel by pixel.*

10. Line 455: Which method has been used to evaluate the "potential conditions", please clarify.

*The half hourly potential latent heat flux is computed using the prescribed mode of the SPARSE model (see (Boulet et al., 2015)). Indeed, potential conditions are expressed through the use of the efficiencies βs and βv which are functionally equivalent to surface resistances ("s" for soil, "v" for vegetation). Their range of validity is [0, 1]. If βv= 1, then the vegetation transpires at the potential rate, and if βs= 1, the soil evaporation rate is that of a saturated surface, while βv= 0 or βs= 0 corresponds to a non-transpiring or non-evaporating surface, respectively. Therefore, in full potential conditions, βs= βv= 1, while in fully stressed conditions, βs= βv= 0.*

*The above paragraph will be added to the SPARSE model description in the manuscript.*

11. Lines 491-492: The Authors reported that . . .."An overestimation of about 15% is found between estimated and measured daily available energy. . ..and the coefficients . . .. . .were applied to remove this bias". If I well understand the above procedure (re- move of bias) is a sort of calibration of the output of modeled on the basis of observed flux station. Please clarify.

*Response: see response to comment 12.*

12. Lines 526-527: About the estimation of sensible heat flux the authors reported that "This result is of great interest considering that the SPARSE model was run with no prior calibration", but I feel a sort of contradiction with the bias removing procedure described in the above comment. Please clarify. Moreover I think that the Authors should describe the accuracy of model prior and after the bias correction.

*Responses to comments 11 and 12:*

*In fact, bias removal does concern neither the SPARSE model which was run with no prior calibration nor its estimates. Since the model provide a single instantaneous estimate of energy budget components, the global solar incoming radiation Rg was used to scale modeled AE and H from instantaneous to daily values (see section 4.2.3), the same applies to instantaneous available energy (see sections 3.3.1 and 3.3.2) computed using remote sensing and meteorological data (equation9 ) and measured H by the XLAS.*
*Indeed, the extrapolation from an instantaneous flux estimate to a daytime flux assumes that the surface energy budget is "self-preserving" i.e. the relative partitioning among components of the budget remains constant throughout the day. However, many studies (Brutsaert and Sugita, 1992; Gurney and Hsu, 1990; Sugita and Brutsaert, 1990) showed that the self-preservation method gives day- time latent heat estimates that are smaller than observed values by 5-10%. Moreover, Anderson et al. (1997) founded that the evaporative fraction computed from instantaneous measured fluxes tends to underestimate the daytime average by about 10%, hence, corrected parameterization was used and a coefficient=1.1 was applied. Similarly, Delogu et al. (2012) founded an overestimation of about 10% between estimated and measured daily component of the available energy thus, a coefficient =0.9 was applied. The Delogu et al. (2012) corrected parameterization were tested, since, in our study case also an overestimation between estimated and measured AE was found, but this coefficient did not give consistent results, therefore, we had to calibrate the extrapolation relationship in order to get accurate daily results of AE and H*
*Thereby,, the applied extrapolation method was tested using in situ Ben Salem flux station measurements. Indeed, daily measured AE (all the same for H) computed as the sum of half-hourly measured AE, was compared to daily AE computed using the extrapolation method from*

*instantaneous measured AE at Terra (equation 13) and Aqua (equation 14) over pas time. This comparison gave an overestimation of about 15% (for AE), hence, corrected parameterizations of available energy AE (coefficients summarized in Table 2) were applied to remove the bias between measured and computed AE (see sections 3.3.2 and 6.1).*

13. Line 545: (Figure 7). Looking at the scatterplot it is clear a more dispersion for H value greater than 150. Is there an explanation of this?

*Response:*

*Possible explanations are:*
    i)    *the XLAS measurement saturation; according to the "Kipp & Zonen Las and XLAS instruction manual", for a path length of 4km and a scintillometer high of 20 m, saturation measurement problem starts from H values of about 300 W.m$^{-2}$*
    ii)    *Uncertainties on the correction of stability using the universal stability function*
    iii)    *Potential inconsistencies between the area average MODIS radiative temperature and the air temperature measured locally at the meteorological station.*

14. Line 604: The Authors reported that "Daily observed and modeled ET over the whole study period were both in the range of 0-4 mm mm.day-1 which is consistent with the land use present in the XLAS pat". In my opinion this is a prosy comment, Trouble if not.

*Response:*

*We agree with the reviewer 1, and the composition of the vegetation cover over the study area (above the scintillometer) with detailed land use percentage should be added to the study area section (section 2.1), in order to show that this area is almost covered by fruit trees spaced by a lot of bare soil, with less herbaceous soil-covering crops; which lead to this range of daily ET. These ET values range was also found in (Saadi et al., 2015) dealing with the same study area.*

15. Line 616-617: The Authors reported that "Some points with little to null ET were recorded from May to July 2013 which can be explained by the very dry conditions and scattered vegetation cover with a considerable amount of bare soil". Why this behavior was not observed in the same period of 2014 ?

*Response:*

*This behavior was not observed in the same period of 2014, because 2014 was a rainy year in comparison to 2013 (more rainfall peaks), so, even supposing that the farmers have the same attitude and cultivate the same crop types between the two years (which is not true in the context of our study area and farmers always change crop types), precipitations favor the growth of spontaneous vegetation over fallows which contribute to ET rise. On the other hand, since the year is rainy, hence, piezometrical level of the water table rises, leading to more water in boreholes and wells which encourage farmers to diversify the crop types and vegetation cover is therefore becomes dense, which contribute to an overall increase in ET.*

16. Line 863: Please check the (Minacapilli and Ciraolo, 2007) reference.

*Response:*

*This reference should be corrected as follows:*

*Minacapilli, M., Ciraolo, G., D Urso, G., & Cammalleri, C. (2007). Evaluating actual evapotranspiration by means of multi-platform remote sensing data: a case study in Sicily. IAHS PUBLICATION, 316, 207.*

**References**

*Anderson, M., Norman, J., Diak, G., Kustas, W., Mecikalski, J., 1997. A two-source time-integrated model for estimating surface fluxes using thermal infrared remote sensing. Remote sensing of environment 60, 195-216.*

*Boulet, G., Mougenot, B., Lhomme, J.P., Fanise, P., Lili-Chabaane, Z., Olioso, A., Bahir, M., Rivalland, V., Jarlan, L., Merlin, O., Coudert, B., Er-Raki, S., Lagouarde, J.P., 2015. The SPARSE model for the prediction of water stress and evapotranspiration components from thermal infra-red data and its evaluation over irrigated and rainfed wheat. Hydrol. Earth Syst. Sci. 19, 4653-4672.*

*Brutsaert, W., Sugita, M., 1992. Application of self-preservation in the diurnal evolution of the surface energy budget to determine daily evaporation. Journal of Geophysical Research: Atmospheres 97, 18377-18382.*

*Courault, D., Clastre, P., Cauchi, P., Delécolle, R., 1998. Analysis of spatial variability of air temperature at regional scale using remote sensing data and a SVAT model, Proceedings of the First International Conference on Geospatial Information in Agriculture and Forestry.*

*Delogu, E., Boulet, G., Olioso, A., Coudert, B., Chirouze, J., Ceschia, E., Le Dantec, V., Marloie, O., Chehbouni, G., Lagouarde, J.P., 2012. Reconstruction of temporal variations of evapotranspiration using instantaneous estimates at the time of satellite overpass. Hydrol. Earth Syst. Sci. 16, 2995-3010.*

*Glenn, E.P., Huete, A.R., Nagler, P.L., Hirschboeck, K.K., Brown, P., 2007. Integrating remote sensing and ground methods to estimate evapotranspiration. Critical Reviews in Plant Sciences 26, 139-168.*

*Gurney, R., Hsu, A., 1990. Relating evaporative fraction to remotely sensed data at the FIFE site.*

*Saadi, S., Simonneaux, V., Boulet, G., Raimbault, B., Mougenot, B., Fanise, P., Ayari, H., Lili-Chabaane, Z., 2015. Monitoring Irrigation Consumption Using High Resolution NDVI Image Time Series: Calibration and Validation in the Kairouan Plain (Tunisia). Remote Sensing 7, 13005.*

*Sugita, M., Brutsaert, W., 1990. Regional surface fluxes from remotely sensed skin temperature and lower boundary layer measurements. Water Resources Research 26, 2937-2944.*

---

## Author Comment (AC2) · 26 Oct 2017

**Authors Response to Reviewer 2 comments**

| | General comments | Authors response |
|---|---|---|
| 1 | Depending upon editor's decision I would like to see further:
1) Figures with better accuracy in their representation. For example, some of them seems to have been the result of quick spreadsheet plots but without including accurate axis ticks, grids, labels, etc. | All figures will be improved in the revised version of the article. Particular interest is given to clarify axis ticks, grid and labels. |
| 2 | 2) Same as for the description of the figure captions and legends. The reader needs to understand a given figure by analyzing the figure and reading the information on the figure caption and legends. | Figures captions and legends will be enhanced in the revised version of the article in order to provide complete information. |
| 3 | 3) A better explanation of the SPARSE methodology is needed, steps and the set of equations in the ET and H estimates. What the assumptions are and what is the physical framework? All of that is missing and therefore theoretically this paper is very weak.

For example, from where the authors got a threshold value of 30 W/m2 to start the iteration? | This article deals with an assessment of SPARSE model accuracy and operational use in a semi arid context over a heterogeneous landscape; the theoretical framework of SPARSE is only summarized since it has been detailed in (Boulet et al., 2015) as well as in the online documentation (http://tully.ups-tlse.fr/gilles.boulet/sparse); since it is critical to have a self-understandable methodology section in the revised version of this article, we will extend the explanation of the SPARSE methodology and add a diagram showing the flowchart of the SPARSE algorithm.

There is no iteration till convergence in the SPARSE algorithm, only a decision tree with decisions made upon the sign of the retrieved soil latent heat flux component in case of invalidity of the unstressed vegetation initialisation. This will be detailed in the added figure showing the SPARSE algorithm.

The $30W.m^2$ is not a threshold to start iteration since there is not a convergence in SPARSE model but it is an arbitrary minimum positive value of soil latent heat flux (LEs) used |

| | | |
|---|---|---|
| | How convergence is achieved is a mystery here and how many iterations and how signal-to-noise ratio of RS data plays a role in that convergence? Which equation provides convergence we don't know. | as the threshold for vegetation stress detection instead of 0, in order to take into account the contribution of vapour transfer from within the topsoil porous network as shown in (Boulet et al., 1997). |
| 4 | 4) I would like the authors to provide adequate justification to the use of formulas to deduce H based on LAS or XLAS. Particularly since the indicated formulas are valid only under the similarity hypothesis of Monin-Obukhov which implies homogenous surface and stationary flows. No justification was provided as for how these conditions were tested to render valid the resulting HLAS flux. | In our study area topography is flat, and landscape is heterogeneous only from an agronomic point of view since we find different land uses (cereals, vegetables and fruit trees mainly olive trees with considerable spacing of bare soil); however, this heterogeneity in landscape features at field scale is randomly distributed and there is no drastic change in height and density of the vegetation at the scale of the XLAS transect (i.e. little heterogeneity at the km scale, most MODIS pixels have similar NDVI values for instance).Support for the MOST theory was assessed by looking at non-dimensional diagrams of normalized $Ct^2$ as the one below which will be shown in the revision as supplementary material. In this figure, we can see that most points are aligned on the theoretical curve of De Bruin et al. (1993). On that basis, we believe that MOST is valid. Points too far from the theoretical line will be excluded from our analysis. Also, on the basis of that figure, the Andreas parameterization will be replaced by the De Bruin one. We also have postprocessing selection criteria for the XLAS data, for instance, we select only H values above 50 W m$^{-2}$ and we will analyse further the XLAS and footprint data on the basis of the z/L values.
[Figure]
 |
| 5 | 5) when the authors discuss about uncertainties it is not clear what kind of uncertainties we are talking about and how have those been calculated? Moreover, uncertainties in heterogeneous terrain based on pure observations XLAS have not | Uncertainties concern mainly: i/ the instantaneous remote sensing data: there is indeed an issue with the MODIS pixel heterogeneity and notably the distribution of components at the intersection between the square pixel and the XLAS footprint. Also, MODIS products, and mainly LST which is paramount in stress coefficient computation, are assumed to be reliable since we do not have means to reprocess them; however, results |

| | | |
|---|---|---|
| | been computed. | could be checked using Landsat high resolution TIR data.
ii/ half hourly forcing and XLAS data (meteorological and flux data);
iii/ the extrapolation method from instantaneous to daily results ;
iv) unlike temperate areas in which sensible hat flux H is relatively low, in our semi-arid study area, H is mostly high leading to important difference between H and LE (which approaches zero) requiring more data post check .
v/ the empirical estimation methods of soil heat flux G (3 methods were tested) as well as the possible daily heat accumulation can lead to possible errors in available energy estimation and in turn in residual LE estimation , hence, both minimum and maximum daily observed LE were presented, the same for the modeled daily LE presented by error bars.
Despite all these possible uncertainties sources, our findings are reasonable compared to previous published results (SAMIR model,(Saadi et al., 2015)). |
| | A reference is provided so that the authors can check on that. Bai, et al., 2015. "Characterizing the Footprint of Eddy Covariance System and Large Aperture Scintillometer Measurements to Validate Satellite-Based Surface Fluxes. *Geoscience and Remote Sensing Letters, IEEE,* 12(5), 943-947, 2015. doi: 10.1109/LGRS.2014.2368580. | Thank you for this interesting reference on which we draw on to add a paragraph in the revised version discussing the uncertainties in heterogeneous terrain based on pure XLAS observations. |
| 6 | 6)  Not clear where the EC flux comes into play. Also footprint functions for the scintillometers need to be accounted for. Reference on this element is provided below. | There are two EC stations located at the top of the towers (on the side of the XLAS emitter and receiver, resp.), which are used to process the XLAS data (initialization of u* and L) and one EC station in the ground, this will be detailed in the revised manuscript:
i) The ground EC station, referred as the Ben Salem flux station measures convective fluxes exchanged between the surface and the atmosphere (H and LE) combined with measurements of the net radiation Rn and the soil heat flux G. Measured Rn and G were among the input data of XLAS derived sensible and latent heat flux computation. In addition measured Rn and G as well as measured H were used to calibrate the extrapolation relationship of the available energy and the sensible heat flux, respectively (sect. 3.3.2 and 4.2.3).
ii) EC set-ups positioned on the two water tower top |

| | | |
|---|---|---|
| | | platforms were used to initialise friction velocity u* values in the scintillometer derived flux computation.
These details will be added in the article revised version in "Experimental Setup" section. |
| 7 | 7)  I would like the authors to provide an in-depth description of physical processes explaining the results in the final figures. Description of what is being presented in the figures is fine but we need more science here. | In the revised version, more physically-based explanation dealing mainly with the outliers will be added to describe the final figures. |
| 8 | As an aside note the use of XLAS is not unique in this problem. A LAS can do 5 km max. Optical beam path and resolve the same situation. What is critical with using XLAS is beyond 5 km optical path. | |

| | **Comments in details** | Authors response |
|---|---|---|
| 1 | Line 45 –off : please put references in chronologic order. This is the proper way to recognize previous work; unless specific discussions are provided which in those cases the trail of references needs to be broken down. This note is valid through the entire paper. | References will be put in chronologic order in the revised version |
| 2 | Line 50: About the claims about water scarcity related to climate change. -or better say climate variability: I wonder how compelling are these claims? – Can the authors substantiate in more details about this problem in this area? This is an important claim and need to be fully addressed by the authors to build context to this research and the methodologies being used. | The paragraph below will be added in the revised version:
*"Indeed, the Mediterranean region is one of the most prominent "Hot-Spots" in future climate change projections (Giorgi and Lionello, 2008) due to an expected larger warming than the global average and to a pronounced increase in precipitation inter-annual variability. The major part of the southern Mediterranean countries, amongst others Tunisia, already suffer from water scarcity, and show a growing water deficit, due to the combined effect of the increasing water demand , and the limited and variable available resources (temporary drought and/or climate change)."* |
| 3 | Line 53: the use of "greatest" here tries to indicate what? "the larger" or "the most important"? This needs to be clearly understood without ambiguity and therefore we need to bring more specificity. | *"greatest"* is replaced by *"the larger"* in the revised version. |
| 4 | Line 56: I'll add complexity in. As we move from ecosystem scale to landscape scales surface heterogeneity but also dynamic of the flow, cloudiness, precipitation come into play more aggressively. This also bring more context to the need of this study. | We have already mentioned the impact of land cover heterogeneity at large scale on the land atmosphere exchange:
*"Moreover, at these scales, land cover is usually heterogeneous and this affects the land-atmosphere exchanges of heat, water and other constituents (Giorgi and Avissar, 1997)."*
However, to develop this idea further, in the revised version, we will provide some more explanation about the hydro-meteorological processes complexity and its impact on climate variables:
*"it is much more difficult at larger scales (irrigated perimeter or watershed) due to the complexity not only of the hydrological processes (Minacapilli and Ciraolo, 2007) but also of the hydro-meteorological processes. Indeed, at landscape scale, surface heterogeneity influences* |

| | | *regional and local climate inducing for example cloudiness and precipitation and temperature patterns difference at areas of higher elevation (hills and mountains surrounding the Kairouan plain) than at the lower elevation plain. Moreover, the land-atmosphere exchanges of heat, water and other constituent are affected by heterogeneous land cover (Giorgi and Avissar, 1997).”* |
|---|---|---|
| 5 | Line 61: I would disagree that "RS techniques becomes essential". Basically it has been demonstrated that plot (or ecosystem) exchanges within same complex canopies do verify consistent differences in sensible heat fluxes (the simplest and ubiquitous flux on earth) over distances that are much smaller than the RS footprint in particular MODIS. See Starkenburg et al., (2015). Starkenburg et al. 2015: "Temperature regimes and turbulent heat fluxes across a heterogeneous canopy in an Alaskan boreal forest". J. Geophys. Res. Atmos., 120: 1348–1360. doi: 10.1002/2014JD022338

 Now, I do agree that RS brings a mean to deduce, within certain ranges, an approximation of fluxes. What about mesoscale models? Or perhaps you wanted to indicated physical models using RS data as input? In any case, I think you should open this perspective here since there are other disciplines other than Remote Sensing Researchers that can also provide the same product. | Remote sensing (RS) can provide estimates of large area fluxes in remote locations, but those estimates are based on the spatial and temporal scales of the measuring systems and thus vary one from another. Hence, one solution is to upscale local micrometeorological measurements to larger spatial scales in order to acquire an optimum representation of land-atmosphere interactions (Samain et al., 2012). However, such up scaling process is not always possible and results are not reliable in comparison to the RS distributed data. In order to keep the introduction as short as possible, we will only point out in the revised version one or two examples of complex physically based LSMs using RS data as inputs to derive ET. |
| 6 | Line 63: vegetation physical properties or characteristics? | In the revised version:
 "*vegetation's physical properties*" will be replaced by "*vegetation's physical characteristics*" |
| 7 | Line 65: Authors use "plot" as one of the scales in which I assume results would be obtained. However, at no point plot-scale was defined. Please whenever plot is used for the first time in the Introduction section for example please clarify that. (excluding | We agree with Reviewer 2 and the word "plot" induces ambiguity. "*plot*" will be replaced by "*field*" in the revised version. |

| | | |
|---|---|---|
| | the abstract). | |
| 8 | Line 87: please rephrase the text between parenthesis. | In the revised version: *"(mostly derived from, say, actual water content in the root zone, wilting point and field capacity) "* will be replaced by: *"'mostly derived from the soil moisture characteristics i.e actual available water content in the root zone, wilting point and field capacity)"* |
| 9 | Line 93: Spell out FAO. If it is not being used anymore in the text, then no need to define an acronym. | In the revised version: *"FAO guidelines"* will be replaced by *"Food and Agriculture Organization-FAO guidelines"* |
| 10 | Line 98-99: get rid of parenthesis here. What is inside is part of the phrase. | Parentheses will be removed in the revised version. |
| 11 | Line 102: FAO-56 put a reference here. Or make a short phrase explanation. | The Allen et al. (1998) reference will be added in the revised version. |
| 12 | Line 103: what is "dry down"? please make sure you check consistency in all phrases. | *"Dry-down period is the period after rain or irrigation where the soil moisture is decreasing due to evapotranspiration and drainage. It is of great interest, because soil moisture has such a strong effect on nearly every aspect of the land surface (heat distribution, albedo, carbon uptake… etc.)."* This short explanation will be added to the revised version. |
| 13 | Line 114: What's the meaning of adding quotes here? If single-source means single source, then no need for quotes. Quotes are used when you use a word or combination of words but you would like to indicate a different meaning. Line 116: same as 114. | Quotes will be removed for *single-source models* and *dual-source models* |
| 14 | Line 117: comma missing before etc. | It will be rectified in the revised version. |
| 15 | Line 128: add "they provide area-averaged sensible heat flux" | *"average sensible heat estimates"* will be replaced by *"area-averaged sensible heat estimates"* in the revised version |
| 16 | Line 130-131: incomplete phrase. And, can you elaborate a little bit more here? | This phrase will be rectified in the revised version as follows: *"Scintillometry can provide sensible heat using different wavelengths (optical wavelength and microwave wavelength ranges), aperture sizes (15-30 cm) and configurations (long-path and* |

| | | *short-path scintillometry )"* |
|---|---|---|
| 17 | Line 132: delete space before comma. | It will be in the revised version. |
| 18 | Line 133: representative of the pixel? It may be the case that for a particular MODIS data your scintillometer data intersects several pixels. Then we are talking about several pixels. | Indeed, the issue of the representativity of the heterogeneity (land use and irrigation practice) at the intersection between the MODIS pixels condidered as homogeneous and the XLAS footprint was not discussed in the submitted version of the article. We will add the suggested reference and discuss the relative percentages of Land Use classes within each MODIS pixel to provide a first guess on these relative heterogeneities. |
| 19 | Line 140: **large-scale area-average** this is the proper measurement that one obtains from a scintillometer. | In the revised version: *"Since the scintillometer only provides spatially averaged sensible heat flux (...)"* will be replaced by *"Since the scintillometer only provides large-scale area-average sensible heat flux (...)"* |
| 20 | Lines 140-143: Here I need help. Are you indicating that to get ET large-scale area-average you use XLAS? But you need to assume a closure fraction or assume is 100% Energy Balance closure. As we increase surface heterogeneity and the atmospheric flow acquires an increased space-time variability then it is difficult to assume 100% energy balance closure. How you do then? Please explain how you treat and eventually circumvent this problem. See for example Foken et al., (2006; 2010) and Foken (2008). Foken, T., F. Wimmer, M. Mauder, C. Thomas, and C. Liebethal, 2006. Some aspects of the energy balance closure problem. *Atmos. Chem. Phys.,* 6, 4395– 4402. Foken, T., 2008: "The energy balance closure problem: An overview"*, Ecol. Appl.*, 18(6), 1351– 1367. Foken, T., M. Mauder, C. Liebethal, F. Wimmer, F. Beyrich, J.-P. Leps, S. Raasch, H. A. R. DeBruin, W. M. L. Meijninger, and J. Bange, 2010: "Energy balance closure for the LITFASS- 2003 experiment"*, Theor. Appl. Climatol.*, *101*(1-2), 149-160, doi: 10.1007/s00704-009-0216-8. | Please see authors' response to the general comment N°4. |

| 21 | Line 146: what is the "layer" approach? Can you be more explicit and detailed? If layer is the name of the approach, then no need to use quotes. | Indeed "*layer*" is the name of the approach, hence, the quote are removed in the revised version. More details about this approach is given in (Boulet et al., 2015) |
|----|---|---|
| 21 | Line 147: when authors normally explain the use of electrical resistance as equivalent models really are not paying attention to the details. So then now you need to explain how you transform an electrical element such as a Resistor, which is a concentrated parameter into a distributed vegetation or soil representation. What are the assumption? Hypothesis? Regions where this approximation is valid and where it fails, etc. I'll give you a hint R=V/I where V(electrical voltage: what is imposed the potential) and I(electrical current, what flows between the boundaries). Then when you say you use Rsoil and Rveg. What are the analogs of V and I here? What R actually means? And how you walk out from the Ohm's Law for concentrated electrical parameters and transition to our problem where these parameters are distributed? This comes from Norman and Kustas TSEB- way before SPARSE. For example, here it is important to remark that vegetation information has to be at much higher resolution than the radiometric information to account for vegetation/forest variations for example the existence of clear areas within the forest or cultivars. How the authors account for that needs better explanations. And, what assumptions underlain these approximations? | The resistance scheme is detailed in Boulet et al. (2015) and is similar to that used in Kustas and Norman (1999), cf. the (Monteith and Unsworth, 2007). V is either a temperature difference (soil-aerodynamic level or vegetation-aerodynamic level) or the corresponding vapour pressure difference. I is the flux component (sensible or latent) and R is the resistance to transfer (aerodynamic resistances within and above the vegetation, stomatal resistance). There is no need of specifying a soil resistance to evaporation because the evaporation rate is directly retrieved. The Series description of the electrical analogy used here is that of most LSMs following (Shuttleworth and Wallace, 1985) which describes the interactions within the soil-plant-atmosphere interface for sparse crops. The radiation interception by sparse crops might be difficult to represent with a layer approach, this will be further commented in the text. |
| 22 | Line 150: I wanted to be clear here that XLAS ONLY can deduce sensible heat not LE. Please make sure this thread is conveyed all the way through your work. | In the revised version: *"The main objective of this paper is to compare H and LE obtained using the SPARSE model and XLAS (...)"* will be replaced by: *"The main objective of this paper is to compare respectively modeled H and LE obtained using the SPARSE model and XLAS measured H and XLAS* |

| | | *derived LE (…)"* |
|---|---|---|
| 23 | Line 158: put "(" to indicate the reference the cultivars are within the phrase. | This will be rectified in the revised version |
| | Line 173: what "double device" means for you. Please be specific. | This phrase will be simplified in the revised version and "*double device*" will be removed. |
| 24 | Figure 2: it is not clear where the XLAS emitter and receiver are specifically located. Put a dot or a symbol to indicate that. Photos actually say nothing here. Now I see that the CSAT is close to the XLAS receiver. I would caution the authors here that any interpretation between XLAS fluxes and EC-CSAT fluxes would not be representative since the EC system is closer to the XLAS receiver and/or transmitter for that matter is the same. More importantly what is not clear here is what are the green contours indicating the footprint? And if these are EC footprint more likely are wrong. Please specify what SPOT5 bands 1,2,3 are in terms of wavelengths and they are used in this work. | Green contours are half-hourly XLAS footprints for selected typical wind conditions. High resolution SPOT5 image of 9[th] April 2013 was only used as background image to illustrate the land cover under the XLAS transect. Hence, figure 2 caption will be rectified in the revised version as follows: *"XLAS Set-up (XLAS transect (white), emitter and receiver are located at the extremity of each white arrow and half-hourly XLAS footprint for selected typical wind conditions (green), MODIS grid (black), orchards (blue) and the location of the Ben Salem meteorological and flux stations. This figure illustrates three colour (red, green, blue) composite of SPOT5 bands 3 (NIR), 2 (VIS-red) and 1(VIS-green) acquired on 9[th] April 2013 and showing in red the cereal plots".* On the hand, EC station flux measurements are not compared to XLAS fluxes along the article. This EC station utility has been already explained in the above responses (general comment N°6). |
| 25 | Line 196: I would write Extra Large Aperture Scintillometer (XLAS) | This will be rectified in the revised version |
| 26 | Line 198: Phrase: "Scintillometer is based on the scintillation method" what is this? | This will be rectified in the revised version |
| 27 | Line 198-200: What is the cause and what is the effect? This phrase is wrong please think about a little bit. | This will be rectified in the revised version as follows: *"Fluxes of sensible heat and momentum cause atmospheric turbulence close to the ground, and create, with surface evaporation, refractive index fluctuations due mainly to air temperature and humidity fluctuations (Hill et al., 1980)."* |
| 28 | Line 205: replace "bean" by "beam" | This will be rectified in the revised version |
| 29 | Line 204: The reference that links scintillations and Cn2 is given by Tatarskii. We need to give the proper reference here. The fact that those references have been using it doesn't mean they were the ones given the foundation for this relationship. We need to make sure we give proper value to the actual | (Tatarskii, 1961) reference is added to the revised version |

| | references. | |
|---|---|---|
| 30 | Line 206: symmetrical to what? What is that symmetry you are talking about? | This sentence will be corrected: "... follows a bell-shape curve. This means that the measured flux is more sensitive to sources located towards the center of the transect than those close to the extremeties. As transmitter and receiver apertures are equal, the sensitivity is symmetrical with respect to that center and decreases similarly towards both ends" |
| 31 | Line 208: get rid of an extra space in the phrase.
Same line: "structure parameter of temperature" by structure parameter of temperature turbulence (refractive index in the case of CN2). | This will be corrected. |
| 32 | Line 210-212: here the authors mentions very cursory a very important problem which is the variation of Cn2 because of the beam height variation across the landscape. It seems this is one point you should be more cautious in bring some references and eventually limit your study on the basis of this sensitivity parameter. | The terrain is very flat; therefore there is little beam height variation across the landscape, except for what is induced by the different roughness of the individual fields. Since the interspace between trees is large, the effective roughness of the orchard is not significantly different from that of cereal fields, esp. given the measurement height. |
| 33 | Line 213: only sensitive to temperatures. Add a period in the phrase. | This will be corrected. |
| 34 | Eq. [1] you introduce here an approximation that then you'll use as an equality. Please explain and substantiate or directly correct the equation. Also, I wonder how much beta introduce error, in this case, a semi-arid environment. | This will be corrected; an equality sign will be used in Eq. 1.
The sensible heat flux dominates the energy balance in most cases; therefore the Bowen ratio is mostly above one. The influence of the beta correction has been analysed in (Solignac et al., 2009) which shows that since the beta closure method does not rely on an exact locally observed beta it is far less sensitive to the precision on beta. |
| 35 | Line 217: iterative methods have intrinsic convergence and resolution errors. You have to specify the convergence error and also how the average of Cn2 gives you a signal with enough SNR to keep the specific convergence factor. Now recently analytical methods have been developed that integrate the set of nonlinear equations in this casa Tatarskii and Monin-Obukhov similarity hypothesis set. See Gruber and Fochesatto, (2013). | This will be verified in the revised version. |

| | | |
|---|---|---|
| | Gruber M. A. and G. J. Fochesatto. 2013: "A New Sensitivity Analysis and Solution Method for Scintillometer Measurements of Area-Average Turbulent Fluxes" *Boundary-Layer Meteorology*, 149:65– 83 DOI 10.1007/s10546-013-9835-9 | |
| 36 | Line 220: Zlas is a function where is that?

Andreas parameterization might not be valid for your site.- Can you justify here?

Zv: is the average canopy height but weighted by the extension of the plots? | $Z_{LAS}$ is not a function, since XLAS experiment is done over a flat surface, $Z_{LAS}$ is the XLAS height, "*effective*" is removed because it induces confusion.

We indeed test the De Bruin (De Bruin et al., 1993) parameterization in the revised version (cf. Figure above).

Zv estimation method is detailed by the end of section 4.1. It accounts for the various heights within the footprint selected using angular zones from the center of the transect. |
| 37 | Eq.4 contains u* but it is not clarified here from where this is taken.
Here we can conclude that XLAS ONLY measures T* as a large-scale area-average variable but u* is a local variable or at least a variable measured at the scale of the EC system which is not the same as the XLAS. Explain please? | u* is not taken from EC system it is computed based on an iteration approach in the beta closure method, only the initialization value of u* was taken from the EC station positioned on the western water tower . |
| 37 | Line 225: rho is the air density and cp here are considered constants. Do they vary across the experiment? | Indeed, air density, pressure and temperature depend on the location on the earth, on altitude and on the season of the year. However, in our study, standard values of air density ($\rho$) and air specific heat at constant pressure (cp) were used without verifying their variation across the experiment since our study concerns a limited extent (10km*8km, same earth location) with flat terrain (no altitude variation) and without a considerable temperature difference between the hot and cold seasons (average monthly temperature oscillates between 10°C and 28°C). |
| 38 | Line 227: nomenclature is Number[space]unit. please correct all the way your text. | This will be rectified in the revised version |
| 39 | Line 228: change "circa" by "near". The correct use of "circa" in English is to indicate something that happened in the past (circa, 1000 AD) for example. | This will be rectified in the revised version |

| | | |
|---|---|---|
| 40 | Line 230: how many "aberrant" values you have in the entire dataset. Please give more precision to the signal processing so that researchers can compare their work with yours in the future. | The following paragraph will be added to the revised version:
*Furthermore, half hourly H_XLAS aberrant values (measurement errors and values higher than 400 w.m$^{-2}$ arising from measurement saturation) were ruled out (3% of the total half hourly measurement throughout the experiment duration). And then daily H_XLAS was computed as the average of the half hourly H_XLAS, 9% of the daily aberrant values were ruled out following the same selection criterion as the half hourly measurement."* |
| 41 | Line 247: and also gives the major sensitivity to H. See also (Gruber et al., 2014) for the specific analytic derivation of the sensitivity to the topography height. Gruber, M. A., G.J. Fochesatto, O.K. Hartogensis, and M. Lysy. 2014: "Functional derivatives applied to error propagation of uncertainties in topography to large-aperture scintillometer-derived heat fluxes". *Atmos. Meas. Tech*., 7, 2361-2371, doi:10.5194/amt-7-2361-2014, 2014. | Again, the terrain here is very flat and does not induce any disturbance linked to topography. |
| 42 | Equations 7 and 8: assume closure of energy balance at 100% please explain how this is possible. And what are your assumptions that lead to this approximation and what is the uncertainty in this assumption. | Please see authors' response to the general comment N°4. |
| 43 | Line 271: Here the authors give an estimation of G/Rn energy partition that is known to be variable not only across a given landscape but also across landscapes. This needs to be carefully estimated. This goes from 31% to very low values in dense canopies. Please be more specific and give values of this factors across all your landscapes. | Indeed G estimation was the most uncertain variable in this study, and that's why we tested three methods to compute it since based on in situ data, we generally found a G accumulation and the daily G is rarely zero.
This part will be largely discussed in the revised version. |
| 44 | Line 284: change "meteo" by "meteorological station". | This will be rectified in the revised version |
| 45 | Lines 280-290: Here the authors bring parameterizations of G. And certainly it is appreciated this compilation. However, it would be best to have a discussion of how | We used standard relationships used in models such as SEBS (Su et al., 2001). An overview of the validity of the relationship for the sole Ben Salem EC station (cereal) will be illustrated in the |

| | | |
|---|---|---|
| | one of these parameterization is or may result more optimal for this work. It seems all the formulas were found and then tossed in this article to see what happens. – So compare your environment with the environment in which those parameterizations were developed and then decide or make some arguments about how to best use or adapt any of these parameterizations. | revision. |
| 46 | Line 294: basically with the current satellite technology we cannot estimate diurnal cycles. However, you must know that at higher latitudes Aqua and Terra have at least six-passages a day. | We agree with Reviewer 2. |
| 47 | Line 300: I don't understand why the authors propose a=1 and b=0 and then find motivation on finding that actually these are not zero. The approximation of Rn by SW (Short Wave Downwelling) is known in micrometeorology and only works to some extent in clear skies when Rn is dominated by SW downwelling. I mean Rn can be negative but never SWdown. So, the way this paragraph is written possess a problem since it is not physically correct. | This paragraph as well as the associated result section (6.1) will be rephrased in the revised version. Indeed, the extrapolation from an instantaneous flux estimate to a daytime flux assumes that the surface energy budget is "self-preserving" i.e. the relative partitioning among components of the budget remains constant throughout the day. However, many studies (Brutsaert and Sugita, 1992;Gurney and Hsu, 1990;Sugita and Brutsaert, 1990) showed that the self-preservation method gives day- time latent heat estimates that are smaller than observed values by 5-10%. Moreover, (Anderson et al., 1997) found that the evaporative fraction computed from instantaneous measured fluxes tends to underestimate the daytime average by about 10%, hence, corrected parameterization was used and a coefficient=1.1 was applied. Similarly, (Delogu et al., 2012) founded an overestimation of about 10% between estimated and measured daily component of the available energy thus, a coefficient =0.9 was applied. The (Delogu et al., 2012) corrected parameterization were tested, since, in our study case also an overestimation between estimated and measured AE was found, but this coefficient did not give consistent results, therefore, we had to calibrate the extrapolation relationship in order to get accurate daily results of AE and H. Thereby, the applied extrapolation method was tested using in situ Ben Salem flux station |

| | | measurements. Indeed, daily measured AE (all the same for H) computed as the sum of half-hourly measured AE, was compared to daily AE computed using the extrapolation method from instantaneous measured AE at Terra (equation 13) and Aqua (equation 14) over pas time. This comparison gave an overestimation of about 15% (for AE), hence, corrected parameterizations of available energy AE (coefficients summarized in Table 2) were applied to remove the bias between measured and computed AE. |
|---|---|---|
| 48 | Line 304: How you weigh the 10x8 km images data by the footprint? What kind of functions are used here to compute the footprint. Please explain. | Daily footprints were computed as a weighted sum of the half hourly footprints by the XLAS sensible heat flux.
Weighing the 10x8 km images data by the footprint means multiplying the 10x8 km result grid by the footprint (weight coefficients ranging from zero and one). |
| 49 | Line 310: replace the "temperature of soil" by "soil temperature". | This will be rectified in the revised version |
| 50 | Here you mention a "reference height" and simultaneously we are talking about a heterogeneous canopy and soil and canopy. Where is that reference height? And what are the assumptions and approximations you are taking by taking this assumption. For example, you are considering some variables at soil level but others at canopy level. How the reference height represents both? And what are the assumptions in terms of physical processes? | Reference height here is the measurement height of the meteorological forcing (2.32 m). This will be précised in the revision. |
| 51 | Eq. [15] you have here a radiative balance equation where it is assumed (without indication) that emissivity (on the left hand side ) is =1. Also this equation needs a reference level and a specific condition for the fluxes to be added and represented at the reference level. Please make sure you are accounting for all these so that the reader can fully understand what your assumptions are and where and under what conditions your analysis is valid. | Details will be added to the revised version |
| 52 | Line 319-320: is SPARSE better than TSEB? Can you give a little bit more explanation here? TSEB has modes to trait | A detailed intercomparison study between TSEB and SPARSE based on several flux stations is underway, first results indicate that bounding the |

| | | |
|---|---|---|
| | vegetation ALEXI and DIS-ALEXI. Are you saying that by incorporating aerodynamic functions makes SPARSE better than TSEB? Please clarify here what's the extent and implication of your comment on the paper. | fluxes simulated by both models by the potential rates given by SPARSE improves the performance of both models which have otherwise similar performances, though constrasted for the various cover types. In SPARSE the aerodynamic functions are those used in almost all Land Surface Models. ALEXI and DIS-ALEXI rely on coarse scale (few km) MSG data, and intercomparison of the ALEXI ET product and the scintillometer will also be carried out in the next future. |
| 53 | Line 325: from where you got the 30W/m2 minimum value? In some environments this will be three times G. Please justify this value. | Please see authors' response to the general comment N°3. |
| 54 | Line 334:335: Here we need to be more specific. What data is from bibliography and what data comes from RS? Please be specific. | After this sentence, bibliography, remote sensing and in situ data were detailed in the following paragraphs, however, in order to be more clear, this section will be rephrased in the revised version. |
| 55 | Line 343: Why you define an acronym MRT that is not used anymore? Acronyms that are not mentioned in the text anymore are unnecessary. | Rectified in the revised version |
| 56 | Line 343-347: this phrase is too long and badly constructed. | This paragraph is reworded in the revised version. |
| 57 | Line 349: We need more detail here. How many days or cases have been excluded from the entire dataset. We need to know how critical is this problem. Because if it is critical then it renders the method useless. | 360 daily data were excluded from the total daily data (1033 days), the following sentence is inserted in the revised version: *"(…) hence, days with missing data in MODIS pixels regarding the scintillometer footprint (35% of the acquired data) were excluded"* |
| 58 | Line 355: k1.15 need space. | This will be rectified in the revised version |
| 59 | Line 357: explain clump-LAI measurements. | Clump LAI is the value of the LAI of an isolated element of vegetation (tree, shrub...); if this element occupies a fraction cover f and is surrounded by bare soi, then the clum LAI value is simply equal to the area average LAI divided by f. This will be specified in the revised version. |
| 60 | Delete the word "Bibliography" from | This will be rectified in the revised version |

| | | |
|---|---|---|
| | Table 1. That column is for sources and a journal peer review is a source. | |
| 61 | Line 379: "overpasses" | This will be rectified in the revised version |
| 62 | Line 383: The second step need a more substance. How come you are running a 30 min fluxes based on a single TIR input? This will result in diurnal cycle of fluxes that are totally biased. I would say that this approximation is only valid for time-intervals in which the turbulence conditions are not too different form the TIR observations. | Indeed, the SPARSE model was run at a half hourly time step using the half hourly meteorological measurements ; assuming that the either the stress factor or the evaporative fraction are invariant during the same day, the diurnal modelled fluxes are accounted for by recovering the diurnal course of either potential ET or available energy. Running the SPARSE model at half hourly time step is only done to get half hourly latent heat flux at potential conditions LEpot wich is equivalent to a reference evapotranspiration whose calculation depends only on half hourly climatic data. This LEpot is used later when computing daily LE based on the stress factor method (section 4.2). This will be better expalined and more detailed in the revised version. |
| 63 | Line 396: please revise the following wording "…complementary part to 1…" | This will be rephrased in the revised version. |
| 64 | Section 4.2 seems to go around and around the subject without going down to the specifics. I think is necessary to simplify the description of methods. | This will be rephrased in the revised version. |
| 65 | Line 407: how you define the wet conditions here? Rain through the day, a specific amount of mm? please be more specific here. | Wet conditions are defined on the basis of a significant amount of rain recorded in the previous day (more than 5 mm). |
| 66 | Eq. [21] assume 100% energy balance closure. You need to justify the use of this condition. | Please see authors' response to the general comment N°4. |
| 67 | Line 429: "deduce" instead of "deduct". | This will be rectified in the revised version |
| 68 | Fig. 5. This figure is a very low quality without precision in the axis. Also we see | Please see authors' response to the general comment N°1. |

| | | |
|---|---|---|
| | only RS data here while it is announced XLAS data. | |
| 69 | Line 475: "convolving" Convolution has a very specific meaning in mathematics. Please verify the use of this term here. | In the revised version:

*"By convolving the XLAS footprint with the SPARSE derived H, we were able to compare compare the modelled values (H_SPARSE t-FP ) with the XLAS measurements (H_XLAS t )."*

will be replaced by

"*SPARSE derived H was weighted by the XLAS footprint in order to be able to compare the modelled values (H_SPARSE t-FP ) with the XLAS measurements (H_XLAS t )"* |
| 70 | Same for the use of modelled or modeled. Both expressions are fine however if your choice is to use words in British English (in this case modelled) you have to be consistent all the way through your paper. | This will be rectified in the revised version |
| 71 | Line 477: "dots"? seriously? | This will be rectified in the revised version |
| 72 | Line 478: Why these two days? Please give the reasons why you are specifically using those days. This is important because when scientist reading your paper would like to reproduce your results they will find no framework to produce such comparisons. | Selection criteria will be added to the revised version:
- Day 2013-86 (24 March 2013) is in the cold season and day 185-2014 (4th July 2014) is in the warm season in order to highlight the land cover impact on LST and thus on modelled H (trees and rainfed and irrigated cereals in winter vs. only irrigated trees and vegetables in summer).
- Day 2013-86 (24 March 2013) shows footprint of strong south wind wile the footprint of day 185-2014 is of a light north wind |
| 73 | Figure 6. I don't understand the coordinates (Y-axis and X-axis). Also the contours of XLAS footprint have no indications. | Figure 6 as well as its caption will be improved in the revised version |
| 74 | Line 482: what you mean by "hot pixel"? Please avoid jargon in the writing. | Hot pixel systematically means a pixel with high LST and low NDVI.
A short explanation will be added to the revised version. |
| 75 | Line 489: In general models are calibrated based on EC systems and thus the deduced large-scale area-average fluxes derived | Indeed in this study, SPARSE model was run in an operational way at landscape scale without parameters calibration, since in our study area, we do not have EC station for each crop type. |

| | | |
|---|---|---|
| | from satellite remote sensing is controlled by LAS observations. | However, SPARSE results at field scale were already compared to EC measurement in an irrigated wheat field and a rainfed wheat field in (Boulet et al., 2015) |
| 76 | Line 490-500: In general, as the heterogeneity in vegetation, soil and eventually in topography leading to variables flows increases the divergence increases. There though cases in which even EC systems that are placed together at distance shorter than the convective ABL development verify more than 50/m2 differences (Starkenburg et al, 2015). So then results expressed here are within the range of reasonable values.
The only one physical explanation why the LAS path by being longer would give different results is when the heterogeneity is such that the BL that develops integrates patches of different thermodynamic and turbulent properties. Then, the mention of issue is interesting but without a correct explanation is useless. | This part will be improved in the revised version based on this comment. |
| 77 | Figure 7. contains features that are important to discuss since there is a change in the bias as function of the flux level. I wonder the authors to discuss this aspect from the physical aspects of the processes dominating this scale integration. | This part will be improved in the revised version. Indeed, possible explanations are:
- the XLAS measurement saturation; according to the "Kipp & Zonen LAS and XLAS instruction manual", for a path length of 4km and a scintillometer height of 20 m, saturation measurement problem starts from H values of about 300 $W.m^{-2}$
- Uncertainties on the correction of stability using the universal stability function
- Potential inconsistencies between the area average MODIS radiative temperature and the air temperature measured locally at the meteorological station. |
| 78 | Figure 10. display several cases where there is a huge divergence in stress index particularly in April and July for both spacecraft. | These individual dates will be discussed in the revised version. |
| 79 | Line 562: here the authors mentioned – uncertainties- but at no point in the paper we are discussing about this. As previously mentioned uncertainties come not only in EC and XLAS observations but also in the approximation used based on 100% | Please see authors' response to the general comment N°4. |

| | | |
|---|---|---|
| | closure in the energy balance. It is confusing and not clear definitively. | |
| 80 | Line 565-570: give some explanation but actually is a description of the time-series. Can you provide a real-actual-explanation about what is the physical processes underlining this divergences and convergences. | The discussion part relating to Figure 11 will be improved in the revised version. |
| 81 | Same from 570 to 575 | Same as comment 80. |
| 82 | Line 588: is this the actual explanation of why there is such divergence or is this another speculation? | Same as comment 80. |
| 83 | Line 590-592: the error indicated here is extremely low now can you please indicate all- conditions in which this is valid and please circumvent this result to the specific interval of conditions in which this is actually valid. | Same as comment 80. |
| 84 | Figure 11. From where and how you got errorbars in blue trace? Figure caption is not clear. We need a accurate description of the contents in the figure. | Figure 11 caption is improved in the revised version.

Error bars for the SPARSE results show the minimum and the maximum daily evapotranspiration (ET) resulting from the three methods used to compute daily ET from instantaneous modelled ET at the time of Terra and Aqua overpasses: evaporative fraction, stress factor and residual methods, hence, six estimates of the daily modelled ET are produced. This will be mentioned in the caption. |
| 85 | Line 610: "valorize" I wonder what the authors wanted to indicate here? | - This word is rather vague indeed, we will precise the perspectives of this work, notably using a LSM applied at the field scale (Etchanchu et al., 2017) to analyse the scaling properties from the field to the footprint of the XLAS and the MODIS pixels similarly to the reference provided by Reviewer 2 (Bai et al., 2015). |
| 86 | SVAT seems not to have been defined earlier. | This will be rectified in the revised version. |

References:

Boulet, G., Mougenot, B., Lhomme, J. P., Fanise, P., Lili-Chabaane, Z., Olioso, A., Bahir, M., Rivalland, V., Jarlan, L., Merlin, O., Coudert, B., Er-Raki, S., and Lagouarde, J. P.: The SPARSE model for the prediction of water stress and evapotranspiration components from thermal infra-red data and its evaluation over irrigated and rainfed wheat, Hydrol. Earth Syst. Sci., 19, 4653-4672, 10.5194/hess-19-4653-2015, 2015.

Boulet, G., Braud, I., and Vauclin, M.: Study of the mechanisms of evaporation under arid conditions using a detailed model of the soil–atmosphere continuum. Application to the EFEDA I experiment, Journal of Hydrology, 193, 114-141, https://doi.org/10.1016/S0022-1694(96)03148-4, 1997.

De Bruin, H., Kohsiek, W., and Van den Hurk, B.: A verification of some methods to determine the fluxes of momentum, sensible heat, and water vapour using standard deviation and structure parameter of scalar meteorological quantities, Boundary-Layer Meteorology, 63, 231-257, 1993.

Saadi, S., Simonneaux, V., Boulet, G., Raimbault, B., Mougenot, B., Fanise, P., Ayari, H., and Lili-Chabaane, Z.: Monitoring Irrigation Consumption Using High Resolution NDVI Image Time Series: Calibration and Validation in the Kairouan Plain (Tunisia), Remote Sensing, 7, 13005, 2015.

Giorgi, F., and Lionello, P.: Climate change projections for the Mediterranean region, Global and planetary change, 63, 90-104, 2008.

Giorgi, F., and Avissar, R.: Representation of heterogeneity effects in earth system modeling: Experience from land surface modeling, Reviews of Geophysics, 35, 413-437, 1997.

Minacapilli, M., and Ciraolo, G.: Evaluating actual evapotranspiration by means of multi-platform remote sensing data: a case study in Sicily, 2007.

Samain, B., Simons, G. W., Voogt, M. P., Defloor, W., Bink, N.-J., and Pauwels, V.: Consistency between hydrological model, large aperture scintillometer and remote sensing based evapotranspiration estimates for a heterogeneous catchment, Hydrology and Earth System Sciences, 16, 2095-2107, 2012.

Kustas, W. P., and Norman, J. M.: Evaluation of soil and vegetation heat flux predictions using a simple two-source model with radiometric temperatures for partial canopy cover, Agricultural and Forest Meteorology, 94, 13-29, https://doi.org/10.1016/S0168-1923(99)00005-2, 1999.

Monteith, J., and Unsworth, M.: Principles of environmental physics, Academic Press, 2007.

Shuttleworth, W. J., and Wallace, J.: Evaporation from sparse crops an energy combination theory, Quarterly Journal of the Royal Meteorological Society, 111, 839-855, 1985.

Hill, R., Clifford, S. F., and Lawrence, R. S.: Refractive-index and absorption fluctuations in the infrared caused by temperature, humidity, and pressure fluctuations, JOSA, 70, 1192-1205, 1980.

Tatarskii, V. I.: Wave propagation in turbulent medium, Wave Propagation in Turbulent Medium, by Valerian Ilich Tatarskii. Translated by RA Silverman. 285pp. Published by McGraw-Hill, 1961., 1961.

Solignac, P. A., Brut, A., Selves, J. L., Béteille, J. P., Gastellu-Etchegorry, J. P., Keravec, P., Béziat, P., and Ceschia, E.: Uncertainty analysis of computational methods for deriving sensible heat flux values from scintillometer measurements, Atmos. Meas. Tech., 2, 741-753, 10.5194/amt-2-741-2009, 2009.

Su, Z., Schmugge, T., Kustas, W. P., and Massman, W. J.: An Evaluation of Two Models for Estimation of the Roughness Height for Heat Transfer between the Land Surface and the Atmosphere, Journal of Applied Meteorology, 40, 1933-1951, 10.1175/1520-0450(2001)040<1933:aeotmf>2.0.co;2, 2001.

Brutsaert, W., and Sugita, M.: Application of self-preservation in the diurnal evolution of the surface energy budget to determine daily evaporation, Journal of Geophysical Research: Atmospheres, 97, 18377-18382, 1992.

Gurney, R., and Hsu, A.: Relating evaporative fraction to remotely sensed data at the FIFE site, 1990.

Sugita, M., and Brutsaert, W.: Regional surface fluxes from remotely sensed skin temperature and lower boundary layer measurements, Water Resources Research, 26, 2937-2944, 1990.

Anderson, M., Norman, J., Diak, G., Kustas, W., and Mecikalski, J.: A two-source time-integrated model for estimating surface fluxes using thermal infrared remote sensing, Remote sensing of environment, 60, 195-216, 1997.

Delogu, E., Boulet, G., Olioso, A., Coudert, B., Chirouze, J., Ceschia, E., Le Dantec, V., Marloie, O., Chehbouni, G., and Lagouarde, J. P.: Reconstruction of temporal variations of evapotranspiration using instantaneous estimates at the time of satellite overpass, Hydrol. Earth Syst. Sci., 16, 2995-3010, 10.5194/hess-16-2995-2012, 2012.

Etchanchu, J., Rivalland, V., Gascoin, S., Cros, J., Brut, A., and Boulet, G.: Effects of multi-temporal high-resolution remote sensing products on simulated hydrometeorological variables in a cultivated area (southwestern France), Hydrol. Earth Syst. Sci. Discuss., 2017, 1-23, 10.5194/hess-2016-661, 2017.

Bai, J., Jia, L., Liu, S., Xu, Z., Hu, G., Zhu, M., and Song, L.: Characterizing the footprint of eddy covariance system and large aperture scintillometer measurements to validate satellite-based surface fluxes, IEEE Geoscience and Remote Sensing Letters, 12, 943-947, 2015.

---

## Author Response (AR1)

**Authors Response to Reviewer 1 comments**

The Authors present an extensive work (reinforced by experimental data) aimed to assess the operational use of the Soil Plant Atmosphere and Remote Sensing Evapotraspiration (SPARSE) model and its accuracy by a comparison to the Scintillometric technique. I think that Authors address relevant scientific questions within the scope of HESS. Furthermore the paper is generally well organized and well written and there-fore the paper could be taken into account for the final publication after a moderate revision. Particularly, The Authors should improve the part of "Results and discussion" (pag. 16-20) with a better description of the validation of SPARSE model carried out with by comparing H and AE estimations with flux station and XLAS scintillometer (see comments n 7, 11 and 12). My comments and questions are as follow:

1. Lines 33-44: The Authors corroborated "the good correspondence between instantaneous H estimates and large aperture scintillometer H measurements" reporting RMSE values expressed in W m-2. As stated by the Authors (Line 418) "For hydrological applications, daily ET is usually required: : :." and in my opinion this means that for hydrological purposes the accuracy of daily evapotranspiration should be expressed in millimeters for day (mmd-1). Therefore in the abstract and through the paper this aspect should be considered and also critically analyzed. From my calculations the accuracy obtained by SPARSE model application should be around 1.6 mmd-1. Is this value "acceptable" ?

*Response:*

*Indeed, we agree with Reviewer 1 that for hydrological purposes the accuracy of daily evapotranspiration (ET) should be expressed in millimeters per day, however, the RMSE values mentioned in the abstract and throughout the paper are instantaneous sensible and latent heat fluxes estimates at the satellite overpass time and are not daily values, therefore, they are expressed in W.m-2. Since, they are instantaneous data, it should not be converted using this formula:*
*47.2 W.m-2\*0.0864/2.45= 1.66 mm/day*
*43.2 W.m-2\*0.0864/2.45= 1.52 mm/day*
*Therefore, we get an instantaneous LE error of about 0.1 mm/0.5.hour around the satellite overpass (around midday, at the max. ET rate)*
*In the revised version of the manuscript (section 6.4), when dealing with daily ET, all values are expressed in mm.day$^{-1}$; following the reviewer's suggestion, we added the model daily ET estimates accuracy (RMSE= 0.7 mm/day) similarly to what as been done for instantaneous results.*

2. Lines 87-88: Is "irrigation requirements" (generally expressed in mmd-1) a prerogative only "of RS-based SWB models" ? Please, clarify.

*Response:*

*Irrigation requirements are mainly estimated using RS-based SWB models, since irrigation is a component of the water balance equation on which is based SWB models. Indeed, the*

*crop coefficient method (FAO56 method) is currently the main method used for scheduling irrigations around the world (Glenn et al., 2007).*

*Irrigation requirement was rarely directly estimated using SEB models. Indeed, SEB outputs are generally actual evapotranspiration (its energy equivalent LE) and if Irrigation is estimated, it should be computed as a residual term of the water balance equation. Exception exists, for example, (Courault et al., 1998) used surface temperature derived from NOAA data and a SVAT model called MAGRET to find parameters linked to the irrigation over the agricultural region "la Crau" in South-Eastern France ; the predicted parameters were the beginning and the end of irrigation, frequency and water quantity diverted.*

3. Line 108: ": : :at the beginning of the process". Please clarify.

*Response:*

*This was corrected before review, and we did not find this expression in line 108 of the last article manuscript version "hess-2017-454-manuscript-version3_discussion" to which we refer. Indeed, in this last version, the mentioned sentence was written as follows: "at the beginning of the dry down".*

4. Lines 111-112: ": : :the lack of information about the actual irrigation scheduling adopted by the farmers is the critical limitation for SWB modeling". I believe that var-ious SWB models (Swap, Cropsyst, FAO56, AcquaCrop) are able to consider both scheduled by farmer irrigation (as input) or predicted irrigation (as output). Please, clarify or modify.

*Response:*

*Indeed, several SWB models such as Swap, Cropsyst, FAO56, AcquaCrop and also the SAMIR model that we have already used (Saadi et al., 2015) are able to consider both methods to take irrigation into account: either an estimated amount provided by the farmer (as an input) or a predicted irrigation with a module to trigger irrigation according to, say, critical soil moisture levels (as an output). We clarify this part in the revised version by saying that the lack of actual irrigation scheduling information does not impact the irrigation estimation by these models, since irrigation could be simulated by SWB models, but rather the validation protocol of irrigation requirements estimates (irrigation data is usually unavailable).*

5. Line 123: Insert ". . ." in dual-source models.

*Response:*

*In the version to which we refer this expression is already put in inverted commas (line 116): "However, separate estimates of evaporation and transpiration makes the "dual-source" models more useful for agrohydrological applications*

6.      Lines 152-154: Clarify that the "layer" approach of SPARSE is essentially a "dual-source" scheme.

*Response:*

*In the revised version of the manuscript, the paragraph is simplified accordingly (line 180):*

*"In this study, (…) were obtained by the SEB method, using the Soil Plant Atmosphere and Remote Sensing Evapotranspiration (SPARSE) (...)."*

*We specify in the "model" section (section 4 line 465) that we use the "layer" approach and define it: "The SPARSE dual-source model solves the energy budgets of the soil and the vegetation. Here we use the "layer approach", for which the resistance network relating the soil and vegetation heat sources to a main reference level through a common aerodynamic level use a series electrical branching"*

7.      Line 187: The Authors should explain (also under a theoretical point of view) the choice to install Scintillometer at a 20 m height. About the experimental setup it is strange the absence of a "net radiometer" that, on the basis of the footprint analysis, could be installed in the average prevalent source area of footprint. The Authors could explain this fact.

*Response:*

*The choice to install Scintillometer at a 20 m height was based on the XLAS installation principle detailed in the "Kipp & Zonen LAS and XLAS instruction manual", indeed, the minimum installation height of the XLAS as function of the path length and for different surface conditions is graphically explained and shows that for a path length of 4km, the XLAS height of 20m is an adequate height since the XLAS is high enough to minimize measurement saturation and not too high to be representative of the 4km path Boundary Layer.*

*The absence of a "net radiometer" is explained by the high heterogeneity of the study area, especially in terms of vegetation cover; therefore, it is not possible to measure the net radiation (Rn) of all plots or even the Rn of "typical" plots (with similar land cover and irrigation practice).*
*This is clarified in the revised version (line 2014).*

8. Line 280: The terms "incoming solar radiation" and "incoming atmospheric radiation" are correct but could generate a misunderstanding. Please use the more classical "shortwave" and "longwave" terminology in eq. (9) and explain how RS data are generally used to solve balance equation of radiation (eq.9).

*Response:*

*In the revised version of the manuscript, the terms "incoming shortwave radiation" and "incoming longwave radiation" are used. This terminology is also used all along the manuscript.. The following paragraph is added accordingly (line 392):*
*"The Ben Salem meteorological station was used to provide $Rg_t$ and $R_{atm-t}$. Remote sensing variables $α$, LST, $ε_s$ and NDVI came from MODIS products"*

9. Line 367: About the "Temporal interpolation of albedo and NDVI" some brief details could be considered.

*Response:*

*Albedo MODIS products (MCD43) are available every 8 days and come from different satellite overpasses over a period of 16 days, the day of interest is central date. Both Terra and Aqua data are used in the generation of this product, providing the highest probability for quality input data and designating it as the acronym MCD, which means Combined product.*

*NDVI MODIS products (MOD13A2/MYD13A2 for Terra and Aqua, respectively) come from different satellite overpasses over a period of 16 days, and they are available every 16 days and separately for Terra and Aqua. Indeed, algorithms generating this product operate on a per-pixel basis and requires multiple daily observations to generate a composite NDVI value that will represent the full period (16 days), the 1km/16days MOD13A2 (respectively MYD13A2) product is an aggregated 250m/16 days MOD13Q1 (respectively MYD13Q1) product..*

*For both products, the data is linearly interpolated over the available dates in order to get daily data. For each pixel, the best data is taken into account (based on the quality index supplied with the product). Therefore, the temporal interpolation was done pixel by pixel.*

*This explanation is inserted in the revised version (line 248).*

10. Line 455: Which method has been used to evaluate the "potential conditions", please clarify.

*The half hourly potential latent heat flux is computed using the prescribed mode of the SPARSE model (see (Boulet et al., 2015)): " The system of equation can also be solved for Ts and Tv only if the efficiencies representing stress levels (dependent on surface soil moisture for the evaporation, and root zone soil moisture for the transpiration) are known. In that case the sole first four equations are solved. This prescribed mode allows computing all the fluxes in known limiting soil moisture levels (very dry, e.g. fully stressed, and wet enough, e.g. potential). (…) The potential evaporation and transpiration rates used later on are computed using this prescribed mode with minimum surface resistance to evaporation and transpiration, respectively."*

*The above paragraph is added to the SPARSE model description in the revised version of the manuscript (line 482).*

11. Lines 491-492: The Authors reported that . . .."An overestimation of about 15% is found between estimated and measured daily available energy. . ..and the coefficients . . .. . .were applied to remove this bias". If I well understand the above procedure (re- move of bias) is a sort of calibration of the output of modeled on the basis of observed flux station. Please clarify.

*Response: see response to comment 12.*

12. Lines 526-527: About the estimation of sensible heat flux the authors reported that "This result is of great interest considering that the SPARSE model was run with no prior calibration", but I feel a sort of contradiction with the bias removing procedure described in the above comment. Please clarify. Moreover I think that the Authors should describe the accuracy of model prior and after the bias correction.

*Responses to comments 11 and 12:*

*In fact, bias removal does concern neither the SPARSE model which was run with no prior calibration nor its estimates. Since the model provide a single instantaneous estimate of energy budget components, the global solar incoming radiation Rg was used to scale modeled AE and H from instantaneous to daily values (see section 4.2.3), the same applies to instantaneous available energy (see sections 3.3.1 and 3.3.2) computed using remote sensing and meteorological data (equation 9 ) and measured H by the XLAS.*

*Indeed, the extrapolation from an instantaneous flux estimate to a daytime flux assumes that the surface energy budget is "self-preserving" i.e. the relative partitioning among components of the budget remains constant throughout the day. However, many studies (Brutsaert and Sugita, 1992; Gurney and Hsu, 1990; Sugita and Brutsaert, 1990) showed that the self-preservation method gives day- time latent heat estimates that are smaller than observed values by 5-10%. Moreover, (Anderson et al., 1997) found that the evaporative fraction computed from instantaneous measured fluxes tends to underestimate the daytime average by about 10%, hence, corrected parameterization was used and a coefficient=1.1 was applied. Similarly, (Delogu et al., 2012) founded an overestimation of about 10% between estimated and measured daily component of the available energy thus, a coefficient =0.9 was applied. The (Delogu et al., 2012) corrected parameterization were tested, since, in our study case also an overestimation between estimated and measured AE was found, but this coefficient did not give consistent results, therefore, we had to calibrate the extrapolation relationship in order to get accurate daily results of AE and H*

*Thereby, the applied extrapolation method was tested using in situ Ben Salem flux station measurements. Indeed, Daily measured available energy $AE_{BS-day}$ (all the same for $H_{BS}$) computed as the average of half-hourly measured $AE_{BS-30}$, was compared to daily available energy ($AE_{BS-day-Terra}$ and $AE_{BS-day-Aqua}$) computed using the extrapolation method from instantaneous measured $AE_{BS-t-Terra}$ and $AE_{BS-t-Aqua}$ at Terra and Aqua overpass time, respectively (Equation 14). Results gave an overestimation of about 15 %. The corrected parameterizations of AE (Table 1), needed to remove the bias between measured ($AE_{BS-day}$) and computed AE ($AE_{BS}$-$_{day-Terra}$ and $AE_{BS-day-Aqua}$), were applied to compute daily remotely sensed AE ($AE_{day}$) from instantaneous AE ($AE_t$) following the extrapolation method shown in equation 14.*

*This explanation is inserted in the revised version (lines 419 to 450 and lines 542 to 554).*

13. Line 545: (Figure 7). Looking at the scatterplot it is clear a more dispersion for H value greater than 150. Is there an explanation of this?

*Response:*

*Possible explanations of the scatter observed or high H values are (revised version line):*

  i)      *the XLAS measurement saturation; according to the "Kipp & Zonen Las and XLAS instruction manual", for a path length of 4km and a scintillometer high of 20 m, saturation measurement problem might be present from H values of about 300 W.m$^{-2}$*
  ii)     *Uncertainties on the correction of stability using the universal stability function*
  iii)    *Potential inconsistencies between the area average MODIS radiative temperature and the air temperature measured locally at the meteorological station.*

14. Line 604: The Authors reported that "Daily observed and modeled ET over the whole study period were both in the range of 0-4 mm mm.day-1 which is consistent with the land use present in the XLAS pat". In my opinion this is a prosy comment, Trouble if not.

*Response:*

*We agree with the reviewer 1, and the composition of the vegetation cover over the study area (above the scintillometer) with detailed land use percentage is added (section 3.2), in order to show that this area is almost covered by fruit trees spaced by a lot of bare soil, with less herbaceous soil-covering crops; which lead to this range of daily ET. These ET values range was also found in (Saadi et al., 2015) dealing with the same study area. This is precised in the revised version (Figure 4).*

15. Line 616-617: The Authors reported that "Some points with little to null ET were recorded from May to July 2013 which can be explained by the very dry conditions and scattered vegetation cover with a considerable amount of bare soil". Why this behavior was not observed in the same period of 2014 ?

*Response:*

*This behavior was not observed in the same period of 2014, because 2014 was a rainy year in comparison to 2013 (more rainfall peaks), so, even supposing that the farmers have the same attitude and cultivate the same crop types between the two years (which is not true in the context of our study area and farmers always change crop types), precipitations favor the growth of spontaneous vegetation over fallows which contribute to ET rise. On the other hand, since the year experiences more rain, farmers cultivate a larger part of the land diversify the crop types and the vegetation cover is denser, this contributes to an overall increase in ET.*
*This explanation is inserted in the revised version (line 693).*

16. Line 863: Please check the (Minacapilli and Ciraolo, 2007) reference.

*Response:*

*This reference should be corrected as follows:*

*Minacapilli, M., Ciraolo, G., D Urso, G., and Cammalleri, C.: Evaluating actual evapotranspiration by means of multi-platform remote sensing data: a case study in Sicily, IAHS PUBLICATION, 316, 207., 2007.*

**References**

Anderson, M., Norman, J., Diak, G., Kustas, W., Mecikalski, J., 1997. A two-source time-integrated model for estimating surface fluxes using thermal infrared remote sensing. Remote sensing of environment 60, 195-216.

Boulet, G., Mougenot, B., Lhomme, J.P., Fanise, P., Lili-Chabaane, Z., Olioso, A., Bahir, M., Rivalland, V., Jarlan, L., Merlin, O., Coudert, B., Er-Raki, S., Lagouarde, J.P., 2015. The SPARSE model for the prediction of water stress and evapotranspiration components from thermal infra-red data and its evaluation over irrigated and rainfed wheat. Hydrol. Earth Syst. Sci. 19, 4653-4672.

Brutsaert, W., Sugita, M., 1992. Application of self-preservation in the diurnal evolution of the surface energy budget to determine daily evaporation. Journal of Geophysical Research: Atmospheres 97, 18377-18382.

Courault, D., Clastre, P., Cauchi, P., Delécolle, R., 1998. Analysis of spatial variability of air temperature at regional scale using remote sensing data and a SVAT model, Proceedings of the First International Conference on Geospatial Information in Agriculture and Forestry.

Delogu, E., Boulet, G., Olioso, A., Coudert, B., Chirouze, J., Ceschia, E., Le Dantec, V., Marloie, O., Chehbouni, G., Lagouarde, J.P., 2012. Reconstruction of temporal variations of evapotranspiration using instantaneous estimates at the time of satellite overpass. Hydrol. Earth Syst. Sci. 16, 2995-3010.

Glenn, E.P., Huete, A.R., Nagler, P.L., Hirschboeck, K.K., Brown, P., 2007. Integrating remote sensing and ground methods to estimate evapotranspiration. Critical Reviews in Plant Sciences 26, 139-168.

Gurney, R., Hsu, A., 1990. Relating evaporative fraction to remotely sensed data at the FIFE site.

Saadi, S., Simonneaux, V., Boulet, G., Raimbault, B., Mougenot, B., Fanise, P., Ayari, H., Lili-Chabaane, Z., 2015. Monitoring Irrigation Consumption Using High Resolution NDVI Image Time Series: Calibration and Validation in the Kairouan Plain (Tunisia). Remote Sensing 7, 13005.

Sugita, M., Brutsaert, W., 1990. Regional surface fluxes from remotely sensed skin temperature and lower boundary layer measurements. Water Resources Research 26, 2937-2944.

**Authors Response to Reviewer 2 comments**

|   | General comments | Authors response |
|---|------------------|------------------|
| 1 | Depending upon editor's decision I would like to see further:
1) Figures with better accuracy in their representation. For example, some of them seems to have been the result of quick spreadsheet plots but without including accurate axis ticks, grids, labels, etc. | All figures are improved in the revised version. Particular attention is paid to axis ticks, grid and labels. |
| 2 | 2) Same as for the description of the figure captions and legends. The reader needs to understand a given figure by analyzing the figure and reading the information on the figure caption and legends. | Figures captions and legends are enhanced in the revised version of the article in order to provide complete information. |
| 3 | 3) A better explanation of the SPARSE methodology is needed, steps and the set of equations in the ET and H estimates. What the assumptions are and what is the physical framework? All of that is missing and therefore theoretically this paper is very weak.

For example, from where the authors got a threshold value of 30 W/m2 to start the iteration? How convergence is achieved is a mystery here and how many iterations and how signal-to-noise ratio of RS data plays a role in that convergence? Which equation provides convergence we don't know. | This article deals with an assessment of the SPARSE model accuracy and operational use in a semi arid context over a heterogeneous landscape; the theoretical framework of SPARSE is only summarized since it has been detailed in (Boulet et al., 2015) as well as in the online documentation (Boulet, 2017); since it is critical to have a self-understandable methodology section in the revised version of this article, we extend the explanation of the SPARSE methodology and add a diagram showing the flowchart of the SPARSE algorithm (Figure 5).

There is no iteration till convergence in the SPARSE algorithm, only a decision tree with decisions made upon the sign of the retrieved soil latent heat flux component: if negative, the assumption of unstressed vegetation is considered as invalid and the stress of the vegetation is retrieved. This is detailed in the added figure.

The $30 \mathrm{Wm}^{-2}$ is not a threshold to start iteration since there is not a convergence in SPARSE model, but it is a minimum positive threshold for vegetation stress detection which accounts for the small but non negligible vapor flow reaching the surface (Boulet et al., 1997). (Revised version line 492) |

| 4 | 4) I would like the authors to provide adequate justification to the use of formulas to deduce H based on LAS or XLAS. Particularly since the indicated formulas are valid only under the similarity hypothesis of Monin-Obukhov which implies homogenous surface and stationary flows. No justification was provided as for how these conditions were tested to render valid the resulting HLAS flux. | In our study area topography is flat, and landscape is heterogeneous only from an agronomic point of view since we find different land uses (cereals, vegetables and fruit trees mainly small olive trees with considerable spacing of bare soil); however, this heterogeneity in landscape features at field scale is randomly distributed and there is no drastic change in height and density of the vegetation at the scale of the XLAS transect (i.e. little heterogeneity at the km scale, most MODIS pixels have similar NDVI values for instance). In these conditions, considering the size of the surface changes in roughness (mean vegetation height ~1.5m), we assumed that the XLAS measurement height was close to the blending height, or either higher. Thus, the fluxes measured by scintillometry are area-averaged and MOST theory can be applied in the flux algorithm computation. In addition, support for the MOST theory was assessed by looking at non-dimensional diagrams of normalized $Ct^2$ and most points are aligned on the theoretical curves of Andreas and (De Bruin et al., 1993). On that basis, we believe that MOST is valid. |
|---|---|---|
| 5 | 5) when the authors discuss about uncertainties it is not clear what kind of uncertainties we are talking about and how have those been calculated? Moreover, uncertainties in heterogeneous terrain based on pure observations XLAS have not been computed. | Uncertainties concern mainly:
i/ the instantaneous remote sensing data: there is indeed an issue with the MODIS pixel heterogeneity and notably the distribution of components at the intersection between the square pixel and the XLAS footprint. Also, MODIS products, and mainly LST which is paramount in stress coefficient computation, are assumed to be reliable since we do not have means to reprocess them; however, results could be checked using Landsat high resolution TIR data.
ii/ half hourly forcing and XLAS data (meteorological and flux data);
iii/ the extrapolation method from instantaneous to daily results ;
iv) unlike temperate areas in which sensible hat flux H is relatively low, in our semi-arid study area, H is mostly high leading to important difference between H and LE (which approaches zero) requiring more data postchecking in the residual derivation of LE from XLAS.
v/ the empirical estimation methods of soil heat flux G (3 methods were tested) as well as the possible |

| | | daily heat accumulation can lead to possible errors in available energy estimation and in turn in residual LE estimation, hence, both minimum and maximum daily observed LE were presented, the same for the modeled daily LE presented by error bars. |
|---|---|---|
| | | Despite all these possible uncertainty sources, our findings are reasonable compared to previous published results (SAMIR model,(Saadi et al., 2015). |
| | A reference is provided so that the authors can check on that. Bai, et al., 2015. "Characterizing the Footprint of Eddy Covariance System and Large Aperture Scintillometer Measurements to Validate Satellite-Based Surface Fluxes. *Geoscience and Remote Sensing Letters, IEEE,* 12(5), 943-947, 2015. doi: 10.1109/LGRS.2014.2368580. | Thank you for this interesting reference on which we draw on to add a paragraph in the revised version discussing the uncertainties in heterogeneous terrain based on pure XLAS observations. |
| 6 | 6) Not clear where the EC flux comes into play. Also footprint functions for the scintillometers need to be accounted for. Reference on this element is provided below. | There are two EC stations located at the top of the towers (on the side of the XLAS emitter and receiver, respectively), which are used to process the XLAS data (initialization of friction velocity u* values and the Obukhov length Lo) and one EC station on the ground. This is detailed in the revised manuscript: *i)Line 218: "two automatic Campbell Scientific (Logan, USA) eddy covariance (EC) flux stations were also positioned at the same level on the two water tower top platforms. Half hourly turbulent fluxes in the western and the eastern EC stations were measured used a sonic anemometer CSAT3 (Campbell Scientific, USA) at a rate of 20 Hz and a sonic anemometer RM 81000 (Young, USA) at a rate of 10 Hz, respectively. The western station data were more reliable with less measurement errors and gaps, hence, the western EC set-up was used initialise friction velocity u* values and the Obukhov length Lo in the scintillometer flux computation".* ii) Line 232: "In addition, an EC flux station, referred as the Ben Salem flux station (few tens of meters away from the meteorological station) was installed from November 2012 to June 2013 in an irrigated wheat field (Figure 2) measuring half hourly convective fluxes exchanged between the* |

| | | surface and the atmosphere ($H_{BS-30}$ and $LE_{BS-30}$) combined with measurements of the net radiation $Rn_{BS-30}$ and the soil heat flux $G_{BS-30}$. Net radiation and soil heat flux measurements were transferred to the meteorological station from June 2013 till June 2015. Since, there are no Rn and G measurements in the two water towers EC stations, $Rn_{BS}$ and $G_{BS}$ measurements were among the inputs data to derive sensible and latent heat fluxes from the XLAS measurements. In addition, measured available energy ($AE_{BS}=Rn_{BS}$—$G_{BS}$) and $H_{BS}$ were used to calibrate the extrapolation relationship of the available energy and the sensible heat flux, respectively" |
|---|---|---|
| 7 | 7)  I would like the authors to provide an in-depth description of physical processes explaining the results in the final figures. Description of what is being presented in the figures is fine but we need more science here. | In the revised version, more physically-based explanation dealing mainly with the outliers is added to describe the final figures. |
| 8 | As an aside note the use of XLAS is not unique in this problem. A LAS can do 5 km max. Optical beam path and resolve the same situation. What is critical with using XLAS is beyond 5 km optical path. | |

| | **Detailed comments** | Authors response |
|---|---|---|
| 1 | Line 45 –off : please put references in chronologic order. This is the proper way to recognize previous work; unless specific discussions are provided which in those cases the trail of references needs to be broken down. This note is valid through the entire paper. | References are put in chronologic order in the revised version. |
| 2 | Line 50: About the claims about water scarcity related to climate change. -or better say climate variability: I wonder how compelling are these claims? – Can the authors substantiate in more details about this problem in this area? This is an important claim and need to be fully addressed by the authors to build context to this research and the methodologies being used. | The paragraph below is added in the revised version (line 50): *"Indeed, the Mediterranean region is one of the most prominent "hot spots" in future climate change projections (Giorgi and Lionello, 2008) due to an expected larger warming than the global average and to a pronounced increase in precipitation inter-annual variability. The major part of the southern Mediterranean countries, among others Tunisia, already suffer from water scarcity, and show a growing water deficit, due to the combined effect of the water needs growth (soaring demography and irrigated areas extension), and the reduction of resources (temporary drought and/or climate change)"* |
| 3 | Line 53: the use of "greatest" here tries to indicate what? "the larger" or "the most important"? This needs to be clearly understood without ambiguity and therefore we need to bring more specificity. | *"greatest"* is replaced by *"the largest"* in the revised version (line 59) |
| 4 | Line 56: I'll add complexity in. As we move from ecosystem scale to landscape scales surface heterogeneity but also dynamic of the flow, cloudiness, precipitation come into play more aggressively. This also bring more context to the need of this study. | We have already mentioned the impact of land cover heterogeneity at large scale on the land atmosphere exchange: *"Moreover, at these scales, land cover is usually heterogeneous and this affects the land-atmosphere exchanges of heat, water and other constituents (Giorgi and Avissar, 1997)."* However, to develop this idea further, in the revised version, we provide some more explanation about the hydro-meteorological processes complexity and its impact on climate variables (line 61): *"(...)it is much more difficult at larger scales (irrigated perimeter or watershed) due to the complexity not only of the hydrological processes* |

| | | |
|---|---|---|
| | | *(Minacapilli et al., 2007) but also of the hydro-meteorological processes. Indeed, at landscape scale, surface heterogeneity influences regional and local climate, inducing for example cloudiness, precipitation and temperature patterns differences between areas of higher elevation (hills and mountains surrounding the Kairouan plain) and the plain downstream. Moreover, at these scales, land cover is usually heterogeneous and this affects the land-atmosphere exchanges of heat, water and other constituents (Giorgi and Avissar, 1997).* |
| 5 | Line 61: I would disagree that "RS techniques becomes essential". Basically it has been demonstrated that plot (or ecosystem) exchanges within same complex canopies do verify consistent differences in sensible heat fluxes (the simplest and ubiquitous flux on earth) over distances that are much smaller than the RS footprint in particular MODIS. See Starkenburg et al., (2015). Starkenburg et al. 2015: "Temperature regimes and turbulent heat fluxes across a heterogeneous canopy in an Alaskan boreal forest". J. Geophys. Res. Atmos., 120: 1348–1360. doi: 10.1002/2014JD022338

 Now, I do agree that RS brings a mean to deduce, within certain ranges, an approximation of fluxes. What about mesoscale models? Or perhaps you wanted to indicated physical models using RS data as input? In any case, I think you should open this perspective here since there are other disciplines other than Remote Sensing Researchers that can also provide the same product. | Remote sensing (RS) can provide estimates of large area fluxes in remote locations, but those estimates are based on the spatial and temporal scales of the measuring systems and thus vary one from another. Hence, one solution is to upscale local micrometeorological measurements to larger spatial scales in order to acquire an optimum representation of land-atmosphere interactions (Samain et al., 2012). However, such upscaling is not always possible and results might not be reliable in comparison to the RS distributed products.
 In order to keep the introduction as short as possible, in the revised version, two examples of complex physically based LSMs using RS data as inputs to derive ET are mentioned (line 76) |
| 6 | Line 63: vegetation physical properties or characteristics? | In the revised version:
 "*vegetation's physical properties*" is replaced by "*vegetation physical characteristics*" (line 72) |

| 7 | Line 65: Authors use "plot" as one of the scales in which I assume results would be obtained. However, at no point plot-scale was defined. Please whenever plot is used for the first time in the Introduction section for example please clarify that. (excluding the abstract). | We agree with Reviewer 2 and the word "plot" induces ambiguity. "*plot*" is replaced by "*field*" in the revised version. (line 75) |
|---|---|---|
| 8 | Line 87: please rephrase the text between parenthesis. | In the revised version:
"*(mostly derived from, say, actual water content in the root zone, wilting point and field capacity)*"
is replaced by:
"*mostly derived from the soil moisture characteristics: actual available water content in the root zone, wilting point and field capacity*"(line 107) |
| 9 | Line 93: Spell out FAO. If it is not being used anymore in the text, then no need to define an acronym. | In the revised version:
"*FAO guidelines*" is replaced by
"*Food and Agriculture Organization-FAO guidelines*" (line 113) |
| 10 | Line 98-99: get rid of parenthesis here. What is inside is part of the phrase. | Parentheses are removed in the revised version. |
| 11 | Line 102: FAO-56 put a reference here. Or make a short phrase explanation. | The Allen et al. (1998) reference is added in the revised version. |
| 12 | Line 103: what is "dry down"? please make sure you check consistency in all phrases. | "*Dry-down period is the period after rain or irrigation where the soil moisture is decreasing due to evapotranspiration and drainage. It is of great interest, because soil moisture has such a strong effect on nearly every aspect of the land surface (heat distribution, albedo, carbon uptake… etc.).*"

This short explanation is added to the revised version (line 123). |
| 13 | Line 114: What's the meaning of adding quotes here? If single-source means single source, then no need for quotes. Quotes are used when you use a word or combination of words but you would like to indicate a different meaning. | Quotes are removed for *single-source models* and *dual-source models*. |

| | | |
|---|---|---|
| | Line 116: same as 114. | |
| 14 | Line 117: comma missing before etc. | It is rectified in the revised version. |
| 15 | Line 128: add "they provide area-averaged sensible heat flux" | "*average sensible heat estimates*" is replaced by "*area-averaged sensible heat flux*" in the revised version (line 154). |
| 16 | Line 130-131: incomplete phrase. And, can you elaborate a little bit more here? | This phrase is rectified in the revised version as follows (line 156): "*Scintillometry can provide sensible heat using different wavelengths (optical wavelength and microwave wavelength ranges), aperture sizes (15-30 cm) and configurations (long-path and short-path scintillometry )*" . |
| 17 | Line 132: delete space before comma. | This is rectified in the revised version. |
| 18 | Line 133: representative of the pixel? It may be the case that for a particular MODIS data your scintillometer data intersects several pixels. Then we are talking about several pixels. | Indeed, the issue of the representativity of the heterogeneity (land use and irrigation practice) at the intersection between the MODIS pixels considered as homogeneous and the XLAS footprint was not discussed in the submitted version of the article. We add the suggested reference and discuss the relative percentages of Land Use classes within each MODIS pixel to provide a first guess on these relative heterogeneities. (line 329) |
| 19 | Line 140: **large-scale area-average** this is the proper measurement that one obtains from a scintillometer. | In the revised version: "*Since the scintillometer only provides spatially averaged sensible heat flux (…)*" is replaced by "*Since the scintillometer only provides large-scale area-average sensible heat flux (…)*" |
| 20 | Lines 140-143: Here I need help. Are you indicating that to get ET large-scale area-average you use XLAS? But you need to assume a closure fraction or assume is 100% Energy Balance closure. As we increase surface heterogeneity and the atmospheric flow acquires an increased space-time variability then it is difficult to assume 100% energy balance closure. How you do then? Please explain how you treat and eventually circumvent this problem. See for example Foken et al., (2006; 2010) and Foken (2008). Foken, T., F. Wimmer, M. Mauder, C. Thomas, and C. Liebethal, 2006. Some aspects of the energy balance | Please see authors' response to the general comment N°4. |

| | | |
|---|---|---|
| | closure problem. *Atmos. Chem. Phys.*, 6, 4395–4402.
Foken, T., 2008: "The energy balance closure problem: An overview", *Ecol. Appl.*, 18(6), 1351– 1367.
Foken, T., M. Mauder, C. Liebethal, F. Wimmer, F. Beyrich, J.-P. Leps, S. Raasch, H. A. R. DeBruin, W. M. L. Meijninger, and J. Bange, 2010: "Energy balance closure for the LITFASS- 2003 experiment", *Theor. Appl. Climatol.*, *101*(1-2), 149-160, doi: 10.1007/s00704-009-0216-8. | |
| 21 | Line 146: what is the "layer" approach? Can you be more explicit and detailed? If layer is the name of the approach, then no need to use quotes. | Indeed "*layer*" is the name of the approach, hence, the quote are removed in the revised version. More details about this approach is given in (Boulet et al., 2015) |
| 21 | Line 147: when authors normally explain the use of electrical resistance as equivalent models really are not paying attention to the details. So then now you need to explain how you transform an electrical element such as a Resistor, which is a concentrated parameter into a distributed vegetation or soil representation. What are the assumption? Hypothesis? Regions where this approximation is valid and where it fails, etc. I'll give you a hint R=V/I where V(electrical voltage: what is imposed the potential) and I(electrical current, what flows between the boundaries). Then when you say you use Rsoil and Rveg. What are the analogs of V and I here? What R actually means? And how you walk out from the Ohm's Law for concentrated electrical parameters and transition to our problem where these parameters are distributed?
This comes from Norman and Kustas TSEB- way before SPARSE.
For example, here it is important to remark that vegetation information has to be at much higher resolution | The resistance scheme is detailed in Boulet et al. (2015) and is similar to that used in (Kustas and Norman, 1999), cf. (Monteith and Unsworth, 2007). V is either a temperature difference (soil-aerodynamic level or vegetation-aerodynamic level) or the corresponding vapour pressure difference. I is the flux component (sensible or latent) and R is the resistance to transfer (aerodynamic resistances within and above the vegetation, stomatal resistance). There is no need of specifying a soil resistance to evaporation because the evaporation rate is directly retrieved. The Series description of the electrical analogy used here is that of most LSMs following (Shuttleworth and Wallace, 1985) which describes the interactions within the soil-plant-atmosphere interface for sparse crops. The radiation interception by sparse crops might be difficult to represent with a layer approach, this will be further commented in the text. |

| | | |
|---|---|---|
| | than the radiometric information to account for vegetation/forest variations for example the existence of clear areas within the forest or cultivars. How the authors account for that needs better explanations. And, what assumptions underlain these approximations? | |
| 22 | Line 150: I wanted to be clear here that XLAS ONLY can deduce sensible heat not LE. Please make sure this thread is conveyed all the way through your work. | In the revised version (line 183): *"The main objective of this paper is to compare H and LE obtained using the SPARSE model and XLAS (…)"* is replaced by: *"The main objective of this paper is to compare the modeled H and LE simulated by the SPARSE model with, respectively, the H measured by the XLAS and the LE reconstructed from the XLAS measurements acquired during two years over a large, heterogeneous area."* |
| 23 | Line 158: put "(" to indicate the reference the cultivars are within the phrase. | This is rectified in the revised version. |
| 24 | Line 173: what "double device" means for you. Please be specific. | This phrase is simplified in the revised version and "*double device*" is removed. (line 205) |
| 25 | Figure 2: it is not clear where the XLAS emitter and receiver are specifically located. Put a dot or a symbol to indicate that. Photos actually say nothing here. Now I see that the CSAT is close to the XLAS receiver. I would caution the authors here that any interpretation between XLAS fluxes and EC-CSAT fluxes would not be representative since the EC system is closer to the XLAS receiver and/or transmitter for that matter is the same. More importantly what is not clear here is what are the green contours indicating the footprint? And if these are EC footprint more likely are wrong. Please specify what SPOT5 bands 1,2,3 are in terms of wavelengths and they are used in this work. | Green contours are half-hourly XLAS footprints for selected typical wind conditions. High resolution SPOT5 image of 9th April 2013 was only used as background image to illustrate the land cover under the XLAS transect. Hence, figure 2 caption is modified in the revised version as follows: *"XLAS set up: XLAS transect (white), for which the emitter and the receiver are located at the extremity of each white arrow, half-hourly XLAS footprint for selected typical wind conditions (green), MODIS grid (black), orchards (blue) and the location of the Ben Salem meteorological and flux stations. Background is a three colour (red, green, blue) composite of SPOT5 bands 3 (NIR), 2 (VIS-red) and 1(VIS-green) acquired on 9th April 2013 and showing in red the cereal plots".* On the other hand, EC station flux measurements are not compared to XLAS fluxes along the article. This EC station utility has been already explained in the above responses (general comment N°6). |

| 26 | Line 196: I would write Extra Large Aperture Scintillometer (XLAS) | This is rectified in the revised version. |
|---|---|---|
| 27 | Line 198: Phrase: "Scintillometer is based on the scintillation method" what is this? | This is rectified in the revised version. |
| 28 | Line 198-200: What is the cause and what is the effect? This phrase is wrong please think about a little bit. | This is rectified in the revised version as follows (line 269): *"Scintillometer measurements are based on the scintillation theory; fluxes of sensible heat and momentum cause atmospheric turbulence close to the ground, and create, with surface evaporation, refractive index fluctuations due mainly to air temperature and humidity fluctuations (Hill et al., 1980)""* |
| 29 | Line 205: replace "bean" by "beam" | This is rectified in the revised version (line) |
| 30 | Line 204: The reference that links scintillations and Cn2 is given by Tatarskii. We need to give the proper reference here. The fact that those references have been using it doesn't mean they were the ones given the foundation for this relationship. We need to make sure we give proper value to the actual references. | (Tatarskii, 1961) reference is added to the revised version (line 275) |
| 31 | Line 206: symmetrical to what? What is that symmetry you are talking about? | *This sentence is corrected (line 275): "The sensitivity of the scintillometer to $C_{n^2}$ along the beam is not uniform and follows a bell-shape curve due to the symmetry of the devices. This means that the measured flux is more sensitive to sources located towards the transect centre and is less affected by those close to the transect extremities."* |
| 32 | Line 208: get rid of an extra space in the phrase. Same line: "structure parameter of temperature" by structure parameter of temperature turbulence (refractive index in the case of CN2). | This is corrected in the revised version (line). |
| 33 | Line 210-212: here the authors mentions very cursory a very important problem which is the variation of Cn2 because of the beam height variation across the landscape. It seems this is one point you should be more cautious in bring some | The terrain is very flat; therefore there is little beam height variation across the landscape, except for what is induced by the various roughness heights of the individual fields. Since the interspace between trees is large, the effective roughness of the orchard is not significantly different from that of cereal fields, and far below |

| | | |
|---|---|---|
| | references and eventually limit your study on the basis of this sensitivity parameter. | the measurement height. |
| 34 | Line 213: only sensitive to temperatures. Add a period in the phrase. | This is corrected in the revised version. |
| 35 | Eq. [1] you introduce here an approximation that then you'll use as an equality. Please explain and substantiate or directly correct the equation. Also, I wonder how much beta introduce error, in this case, a semi-arid environment. | This is corrected; an equality sign is used in Eq. 1. The sensible heat flux dominates the energy balance in most cases; therefore the Bowen ratio is mostly above one. The influence of the beta correction has been analyzed in (Solignac et al., 2009) which shows that since the beta closure method does not rely on an exact locally observed beta it is far less sensitive to the precision on beta. |
| 36 | Line 217: iterative methods have intrinsic convergence and resolution errors. You have to specify the convergence error and also how the average of Cn2 gives you a signal with enough SNR to keep the specific convergence factor. Now recently analytical methods have been developed that integrate the set of nonlinear equations in this casa Tatarskii and Monin-Obukhov similarity hypothesis set. See Gruber and Fochesatto, (2013). Gruber M. A. and G. J. Fochesatto. 2013: "A New Sensitivity Analysis and Solution Method for Scintillometer Measurements of Area-Average Turbulent Fluxes" *Boundary-Layer Meteorology*, 149:65– 83 DOI 10.1007/s10546-013-9835-9 | I'm not sure to fully understand the reviewer's remark. Actually, as shown by (Gruber and Fochesatto, 2013), the height z at which $C_T^2$ is sampled can substantially affect the sensible heat flux (20%), but in our study, the *in situ* G measurement (used to initialize the energy budget closure) has also an impact on the estimate of H_XLAS throughout the convergence algorithm. Since XLAS measurements were processed at the beginning of the project, no sensitivity analysis of theses variables, *e.g.* effective height z, initial guess of the iterative algorithm (local *vs* integrated *via* remote sensing or modeling) was performed. As it is not the scope of the paper, we didn't achieve any sensitivity analysis on XLAS fluxes computation to determine which parameter has the strongest influence on the flux uncertainty. |
| 37 | Line 220: Zlas is a function where is that?

Andreas parameterization might not be valid for your site.- Can you justify here?

Zv: is the average canopy height but | $Z_{LAS}$ is not a function, since the XLAS experiment took place over a flat surface, $Z_{LAS}$ is the XLAS height; the word "*effective*" is therefore removed because it induces confusion.

We indeed test the De Bruin (De Bruin et al., 1993) parameterization in the revised version (cf. Figure above). |

| | weighted by the extension of the plots? | Zv estimation method is detailed by the end of section 4.1. It accounts for the various heights within the footprint selected using angular zones originating from the centre of the transect, and supported by high resolution remote sensing data. |
|---|---|---|
| 37 | Eq.4 contains u* but it is not clarified here from where this is taken. Here we can conclude that XLAS ONLY measures T* as a large-scale area-average variable but u* is a local variable or at least a variable measured at the scale of the EC system which is not the same as the XLAS. Explain please? | u* is not taken from EC system, it is computed based on an iteration approach in the beta closure method, only the initialization value of u* was taken from the EC station positioned on the western water tower. |
| 38 | Line 225: rho is the air density and cp here are considered constants. Do they vary across the experiment? | Indeed, air density, pressure and temperature depend on the location on the earth, on altitude and on the season of the year. However, in our study, standard values of air density ($\rho$) and air specific heat at constant pressure (cp) were used without verifying their variation across the experiment since our study concerns a limited extent (10 km*8 km, same earth location) with flat terrain (no altitude variation) and without a considerable temperature difference between the hot and cold seasons (average monthly temperature oscillates between 10°C and 28°C). |
| 39 | Line 227: nomenclature is Number[space]unit. please correct all the way your text. | This is rectified in the revised version. |
| 40 | Line 228: change "circa" by "near". The correct use of "circa" in English is to indicate something that happened in the past (circa, 1000 AD) for example. | This is rectified in the revised version. |
| 41 | Line 230: how many "aberrant" values you have in the entire dataset. Please give more precision to the signal processing so that researchers can compare their work with yours in the future. | The following paragraph is added to the revised version (line 306): *"Furthermore, half hourly H_XLAS aberrant values due to measurement errors and values higher than 400 $Wm^{-2}$, arising from measurement saturation, were ruled out (3% of the total measurement throughout the experiment duration)"* |
| 42 | Line 247: and also gives the major sensitivity to H. See also (Gruber et | Again, the terrain here is very flat and does not induce any disturbance linked to topography. |

| | | |
|---|---|---|
| | al., 2014) for the specific analytic derivation of the sensitivity to the topography height.
Gruber, M. A., G.J. Fochesatto, O.K. Hartogensis, and M. Lysy. 2014: "Functional derivatives applied to error propagation of uncertainties in topography to large-aperture scintillometer-derived heat fluxes".
*Atmos. Meas. Tech.*, 7, 2361-2371, doi:10.5194/amt-7-2361-2014, 2014. | |
| 43 | Equations 7 and 8: assume closure of energy balance at 100% please explain how this is possible. And what are your assumptions that lead to this approximation and what is the uncertainty in this assumption. | Please see authors' response to the general comment N°4. There is no large scale advection of heat and the XLAS is located above the blending height, therefore we expect that the 100% energy closure assumption is valid. |
| 44 | Line 271: Here the authors give an estimation of G/Rn energy partition that is known to be variable not only across a given landscape but also across landscapes. This needs to be carefully estimated. This goes from 31% to very low values in dense canopies. Please be more specific and give values of this factors across all your landscapes. | Indeed G estimation was the most uncertain variable in this study, and that's why we tested three methods to compute it since based on in situ data, we generally found an accumulation of G and the daily G is rarely zero.
This part is discussed in the revised version (line 365). |
| 45 | Line 284: change "meteo" by "meteorological station". | This is rectified in the revised version |
| 46 | Lines 280-290: Here the authors bring parameterizations of G. And certainly it is appreciated this compilation. However, it would be best to have a discussion of how one of these parameterization is or may result more optimal for this work. It seems all the formulas were found and then tossed in this article to see what happens. – So compare your environment with the environment in which those | We used standard relationships used in models such as SEBS (Su et al., 2001). An overview of the validity of the relationship for the sole Ben Salem EC station (cereal) is illustrated in the revision (line 384). |

| | | |
|---|---|---|
| | parameterizations were developed and then decide or make some arguments about how to best use or adapt any of these parameterizations. | |
| 47 | Line 294: basically with the current satellite technology we cannot estimate diurnal cycles. However, you must know that at higher latitudes Aqua and Terra have at least six-passages a day. | We agree with Reviewer 2. |
| 48 | Line 300: I don't understand why the authors propose a=1 and b=0 and then find motivation on finding that actually these are not zero. The approximation of Rn by SW (Short Wave Downwelling) is known in micrometeorology and only works to some extent in clear skies when Rn is dominated by SW downwelling. I mean Rn can be negative but never SWdown. So, the way this paragraph is written possess a problem since it is not physically correct. | This paragraph as well as the associated result section (6.1) is rephrased in the revised version (lines 419 to 450 and lines 542 to 554) |
| 49 | Line 304: How you weigh the 10x8 km images data by the footprint? What kind of functions are used here to compute the footprint. Please explain. | Daily footprints were computed as a weighted sum of the half hourly footprints by the XLAS sensible heat flux. Weighing the 10x8 km images data by the footprint means multiplying the 10x8 km result grid by the footprint (weight coefficients ranging from zero and one). |
| 50 | Line 310: replace the "temperature of soil" by "soil temperature". | This is rectified in the revised version |
| 51 | Here you mention a "reference height" and simultaneously we are talking about a heterogeneous canopy and soil and canopy. Where is that reference height? And what are the assumptions and approximations you are taking by taking this assumption. For example, you are considering some variables at soil level but others at canopy level. How the reference height represents | Reference height here is the measurement height of the meteorological forcing (2.32 m). This is precised in the revision. |

| | both? And what are the assumptions in terms of physical processes? | |
|---|---|---|
| 52 | Eq. [15] you have here a radiative balance equation where it is assumed (without indication) that emissivity (on the left hand side ) is =1. Also this equation needs a reference level and a specific condition for the fluxes to be added and represented at the reference level. Please make sure you are accounting for all these so that the reader can fully understand what your assumptions are and where and under what conditions your analysis is valid. | Details are added to the revised version (line 467). |
| 53 | Line 319-320: is SPARSE better than TSEB? Can you give a little bit more explanation here? TSEB has modes to trait vegetation ALEXI and DIS-ALEXI. Are you saying that by incorporating aerodynamic functions makes SPARSE better than TSEB? Please clarify here what's the extent and implication of your comment on the paper. | A detailed intercomparison study between TSEB and SPARSE based on several flux stations is underway, first results indicate that bounding the fluxes simulated by both models by the potential rates given by SPARSE improves the performance of both models which have otherwise similar performances, though constrasted for the various cover types. In SPARSE the aerodynamic functions are those used in almost all Land Surface Models. ALEXI and DIS-ALEXI rely on coarse scale (few km) MSG data, and intercomparison of the ALEXI ET product and the scintillometer will also be carried out in the next future. |
| 54 | Line 325: from where you got the 30W/m2 minimum value? In some environments this will be three times G. Please justify this value. | Please see authors' response to the general comment N°3. |
| 55 | Line 334:335: Here we need to be more specific. What data is from bibliography and what data comes from RS? Please be specific. | After this sentence, bibliography, remote sensing and in situ data are detailed in the following paragraphs, however, in order to be more clear, this section will be rephrased in the revised version. |
| 56 | Line 343: Why you define an acronym MRT that is not used anymore? Acronyms that are not mentioned in the text anymore are unnecessary. | Rectified in the revised version |

| 57 | Line 343-347: this phrase is too long and badly constructed. | This paragraph is reworded in the revised version. |
|---|---|---|
| 58 | Line 349: We need more detail here. How many days or cases have been excluded from the entire dataset. We need to know how critical is this problem. Because if it is critical then it renders the method useless. | 360 daily data were excluded from the total daily data (1033 days), the following sentence is inserted in the revised version: *"(...) hence, days with missing data in MODIS pixels regarding the scintillometer footprint (35% of the acquired data) were excluded"* |
| 59 | Line 355: k1.15 need space. | Rectified in the revised version |
| 60 | Line 357: explain clump-LAI measurements. | Clump LAI is the value of the LAI of an isolated element of vegetation (tree, shrub...); if this element occupies a fraction cover f and is surrounded by bare soil, then the clump LAI value is simply equal to the area average LAI divided by f. This is specified in the revised version (Line 402). |
| 61 | Delete the word "Bibliography" from Table 1. That column is for sources and a journal peer review is a source. | Rectified in the revised version |
| 62 | Line 379: "overpasses" | Rectified in the revised version |
| 63 | Line 383: The second step need a more substance. How come you are running a 30 min fluxes based on a single TIR input? This will result in diurnal cycle of fluxes that are totally biased. I would say that this approximation is only valid for time-intervals in which the turbulence conditions are not too different form the TIR observations. | Indeed, the SPARSE model was run at a half hourly time step using the half hourly meteorological measurements ; assuming that either the stress factor or the evaporative fraction are invariant during the same day, the diurnal modelled fluxes are accounted for by recovering the diurnal course of either potential ET or available energy AE. Running the SPARSE model at half hourly time step is only done to get half hourly latent heat flux in potential conditions LEpot which is equivalent to a reference evapotranspiration whose calculation depends only on half hourly climatic data. This LEpot is used later when computing daily LE based on the stress factor method (section 4.2). This is better explained and more detailed in the revised version (line 508). |
| 64 | Line 396: please revise the following | Rephrased in the revised version. |

| | wording "…complementary part to 1…" | |
|---|---|---|
| 65 | Section 4.2 seems to go around and around the subject without going down to the specifics. I think is necessary to simplify the description of methods. | Rephrased in the revised version. |
| 66 | Line 407: how you define the wet conditions here? Rain through the day, a specific amount of mm? please be more specific here. | Wet conditions are defined on the basis of a significant amount of rain recorded in the previous day (more than 5 mm). This is clarified. |
| 67 | Eq. [21] assume 100% energy balance closure. You need to justify the use of this condition. | Please see authors' response to the general comment N°4. |
| 68 | Line 429: "deduce" instead of "deduct". | Rectified in the revised version |
| 69 | Fig. 5. This figure is a very low quality without precision in the axis. Also we see only RS data here while it is announced XLAS data. | Please see authors' response to the general comment N°1. |
| 70 | Line 475: "convolving" Convolution has a very specific meaning in mathematics. Please verify the use of this term here. | In the revised version: *"By convolving the XLAS footprint with the SPARSE derived H, we were able to compare the modelled values ($H\_SPARSE_{t\text{-}FP}$ ) with the XLAS measurements ($H\_XLAS_t$ )."* is replaced by "*SPARSE derived H was weighted by the XLAS footprint in order to be able to compare the modeled values ($H\_SPARSE_{t\text{-}FP}$) with the XLAS measurements ($H\_XLAS_t$)*" |
| 71 | Same for the use of modelled or modeled. Both expressions are fine however if your choice is to use words in British English (in this case | Rectified in the revised version |

| | | |
|---|---|---|
| | modelled) you have to be consistent all the way through your paper. | |
| 72 | Line 477: "dots"? seriously? | Rectified in the revised version |
| 73 | Line 478: Why these two days? Please give the reasons why you are specifically using those days. This is important because when scientist reading your paper would like to reproduce your results they will find no framework to produce such comparisons. | Selection criteria are added to the revised version (line 578):
- Day 2013-86 (24 March 2013) is in the cold season and day 185-2014 (4th July 2014) is in the warm season in order to highlight the land cover impact on LST and thus on modelled H (trees and rainfed and irrigated cereals in winter vs. only irrigated trees and vegetables in summer).
- Day 2013-86 (24 March 2013) shows footprint of strong south wind while the footprint of day 185-2014 is of a light north wind |
| 74 | Figure 6. I don't understand the coordinates (Y-axis and X-axis). Also the contours of XLAS footprint have no indications. | Figure 6 as well as its caption is improved in the revised version |
| 75 | Line 482: what you mean by "hot pixel"? Please avoid jargon in the writing. | Hot pixel systematically means a pixel with high LST and low NDVI.
A short explanation is added to the revised version. |
| 76 | Line 489: In general models are calibrated based on EC systems and thus the deduced large-scale area-average fluxes derived from satellite remote sensing is controlled by LAS observations. | Indeed in this study, SPARSE model was run in an operational way at landscape scale without parameters calibration, since in our study area, we do not have EC station for each crop type. However, SPARSE results at field scale were already compared to EC measurement in an irrigated wheat field and a rainfed wheat field in (Boulet et al., 2015) |
| 77 | Line 490-500: In general, as the heterogeneity in vegetation, soil and eventually in topography leading to variables flows increases the divergence increases. There though cases in which even EC systems that are placed together at distance shorter than the convective ABL development verify more than 50/m2 differences (Starkenburg et al, 2015). So then results expressed here are within the range of reasonable values.
The only one physical explanation | Please see authors' response to the general comment N°4. |

| | | |
|---|---|---|
| | why the LAS path by being longer would give different results is when the heterogeneity is such that the BL that develops integrates patches of different thermodynamic and turbulent properties. Then, the mention of issue is interesting but without a correct explanation is useless. | |
| 78 | Figure 7. contains features that are important to discuss since there is a change in the bias as function of the flux level. I wonder the authors to discuss this aspect from the physical aspects of the processes dominating this scale integration. | This part is improved in the revised version. Indeed, possible explanations are:
- the XLAS measurement saturation; according to the "Kipp & Zonen LAS and XLAS instruction manual", for a path length of 4km and a scintillometer height of 20 m, saturation measurement problem starts from H values of about 300 W.m$^{-2}$
- Uncertainties on the correction of stability using the universal stability function
- Potential inconsistencies between the area average MODIS radiative temperature and the air temperature measured locally at the meteorological station. |
| 79 | Figure 10. display several cases where there is a huge divergence in stress index particularly in April and July for both spacecraft. | These individual dates are discussed in the revised version. |
| 80 | Line 562: here the authors mentioned –uncertainties- but at no point in the paper we are discussing about this. As previously mentioned uncertainties come not only in EC and XLAS observations but also in the approximation used based on 100% closure in the energy balance. It is confusing and not clear definitively. | Please see authors' response to the general comment N°4. |
| 81 | Line 565-570: give some explanation but actually is a description of the time-series. Can you provide a real-actual-explanation about what is the physical processes underlining this divergences and convergences. | The discussion part relating to Figure 11 is improved in the revised version. |
| 82 | Same from 570 to 575 | Same as comment 80. |

| 83 | Line 588: is this the actual explanation of why there is such divergence or is this another speculation? | Same as comment 80. |
|----|----|----|
| 84 | Line 590-592: the error indicated here is extremely low now can you please indicate all- conditions in which this is valid and please circumvent this result to the specific interval of conditions in which this is actually valid. | Same as comment 80. |
| 85 | Figure 11. From where and how you got errorbars in blue trace? Figure caption is not clear. We need a accurate description of the contents in the figure. | Figure 11 caption is improved in the revised version.

Error bars for the SPARSE results show the minimum and the maximum daily evapotranspiration (ET) resulting from the three methods used to compute daily ET from instantaneous modeled ET at the time of Terra and Aqua overpasses: evaporative fraction, stress factor and residual methods, hence, six estimates of the daily modelled ET are produced. |
| 86 | Line 610: "valorize" I wonder what the authors wanted to indicate here? | This word is rather vague indeed, we precise the perspectives of this work, notably using a LSM applied at the field scale (Etchanchu et al., 2017) to analyse the scaling properties from the field to the footprint of the XLAS and the MODIS pixels similarly to the reference provided by Reviewer 2 (Bai et al., 2015). |
| 87 | SVAT seems not to have been defined earlier. | Rectified in the revised version. |

*Correspondence to*: Sameh Saadi (saadi_sameh@hotmail.fr)

**Abstract.**

In semi-arid areas, agricultural production is restricted by water availability; hence efficient agricultural water management is a major issue. The design of tools providing regional estimates of evapotranspiration (ET), one of the most relevant water balance fluxes, may help the sustainable management of water resources.

Remote sensing provides periodic data about actual vegetation temporal dynamics (through the Normalized Difference Vegetation Index NDVI) and water availability under water stress (through the land surface temperature LST) which are crucial factors controlling ET.

In this study, spatially distributed estimates of ET (or its energy equivalent, the latent heat flux LE) in the Kairouan plain (Central Tunisia) were computed by applying the Soil Plant Atmosphere and Remote Sensing Evapotranspiration (SPARSE) model fed by low resolution remote sensing data (Terra and Aqua MODIS). The work goal was to assess the operational use of the SPARSE model and the accuracy of the modeled i) sensible heat flux (H) and ii) daily ET over a heterogeneous semi-arid landscape with a complex land cover (*i.e.* trees, winter cereals, summer vegetables).

SPARSE was run to compute instantaneous estimates of H and LE fluxes at the satellite overpass time. The good correspondence ($R^2$= 0.60 and 0.63 and RMSE=57.89 Wm$^{-2}$ and 53.85 Wm$^{-2}$; for Terra and Aqua, respectively) between instantaneous H estimates and large aperture scintillometer (XLAS) H measurements along a path length of 4 km over the study area showed that the SPARSE model presents satisfactory accuracy. Results showed that, despite the fairly large scatter, the instantaneous LE can be suitably estimated at large scale (RMSE=47.20 Wm$^{-2}$ and 43.20 Wm$^{-2}$; for Terra and Aqua, respectively and $R^2$= 0.55 for both satellites). Additionally, water stress was investigated by comparing modeled (SPARSE) and observed (XLAS) water stress values; we found that most points were located within a 0.2 confidence interval, thus the general tendencies are well reproduced. Even though extrapolation of instantaneous latent heat flux values to daily totals was less obvious, daily ET estimates are deemed acceptable.

KEYWORDS: Evapotranspiration, Remote sensing, SPARSE model, scintillometer, water stress.

[revised manuscript text omitted]

360

**Figure 3 : MODIS pixels partially or totally covered by XLAS source area**

The percentage of land use classes was computed for i) the part of each pixel that lies within the footprint, and ii) the complementary part of the pixel located outside of the footprint (Figure 4). Results show that difference in percentages of each land use classes for the pixel fractions located within or outside the footprint is low with 1.8%, 1.7%, 1.0% and 3.5% for cereals, market gardening, trees and bare soil, respectively. Moreover, the major part of the area above transect is covered by fallow and orchards. The land use classes' partition inside the 13 MODIS pixels totally covered by the average footprint is comparable.

365

[Figure]

**Figure 4: Land use classes' percentage of the MODIS pixels within or outside the footprint**

370 ### 3.3 XLAS derived latent heat flux

Instantaneous (LE_residual_XLAS$_{t-FP}$) and daily (LE_residual_XLAS$_{day-FP}$) XLAS derived latent heat flux (*i.e.* residual latent heat flux) of the XLAS upwind area were computed using the energy budget closure of the XLAS measured sensible heat flux (H_XLAS) with additional estimations of remotely sensed net surface radiation Rn combined withand soil heat flux G, as available energy (AE=Rn-G), as follows:

$$\text{LE\_residual\_XLAS}_{t-FP} = \text{AE}_{t-FP} - \text{H\_XLAS}_t H\_XLAS_t \qquad (7)$$

$$\text{LE\_residual\_XLAS}_{day-FP} = \text{AE}_{day-FP} - \text{H\_XLAS}_{day} \qquad (8)$$

375 H_XLAS$_t$ isand H_XLAS$_{day}$ are respectively the scintillometer sensible heat fluxinstantaneous and daily measured H at the time of the satellite overpass interpolated from the half hourly fluxes measurements. Daily H (H_XLAS$_{day}$) was computed as the average of the half hourly XLAS-measured H. Daily available energy within

the footprint ($AE_{day-FP}$) was computed from instantaneous available energy ($AE_{t-FP}$) as detailed in Sect. 3.3.1 and Sect. 3.3.2. 3.3.2. The subscripts "30", "day" and "t" refer to half hourly, daily and instantaneous (at the time of Terra and Aqua overpasses) variables, respectively; while the subscript "FP" means that the footprint is taken into account *i.e.* instantaneous or the daily (depending on time scale) footprint was multiplied by the variable.

**3.3.1 Instantaneous available energy**

Net surface radiation is the balance of energy between incoming and outgoing shortwave and longwave radiation fluxes at the land-atmosphere interface. RemoteRemotely sensed surface radiative budget components provide unparalleled spatial and temporal information, thus several studies have attempted to estimate net radiation by combining remote sensing observations with surface and atmospheric data. Net radiation equation can be written as follows:

$$Rn = (1 - \alpha)Rg + \varepsilon_s \times R_{atm} R_{atm} - \varepsilon_s \ast \sigma \ast LST^4 LST^4 \tag{9}$$

where $Rg$ is the incoming shortwave radiation (W.m$^{-2}$), $R_{atm}$ is the incoming longwave radiation (W.m$^{-2}$), $\varepsilon_s$ is the surface emissivity, $\sigma$ is Stefan-Boltzmann coefficient (W.m$^{-2}$.K$^4$) , $\alpha$ is the albedo, and LST is the land-surface temperature (°K).

The soil heat flux G depends on the soil type and water content as well as the vegetation type (Allen et al., 2005).The direct estimation of G by remote sensing data is not possible (Allen et al., 2011), however, empirical relations couldcan estimate the fraction $\xi$=G/Rn as a function of soil and vegetation characteristics using satellite image data, such as the LAI, NDVI, α and LST. In order to estimate the Generally, G/Rn ratio, several methods have been tested for various types represents 5-20% of surfaces at different locations (Bastiaanssen, 1995; Burba et al., 1999; Choudhury et al., 1987; Jackson et al., 1987; Kustas and Daughtry, 1990; Kustas et al., 1993; Ma et al., 2002; Payero et al., 2001).

Rn during daylight hours Danelichen(Kalma et al.(2014., 2008) evaluated the parameterization of these different models in three sites in Mato Grosso state in Brazil and found that the model proposed by Bastiaanssen (2005) showed the best performance for all sites, followed by the model from Choudhury et al. (1987) and Jackson et al. (1987). Hence to estimate G, we tested three methods:

. In order to estimate the G/Rn ratio, several methods have been tested for various types of surfaces at different locations. The most common methods parameterize ξ as a constant for the entire day or at satellite overpass time Bastiaanssen (2005)(Ventura et al., 1999) :

, according to NDVI (Jackson et al., 1987; Kustas and Daughtry, 1990), LAI (Choudhury et al., 1987; Kustas et al., 1993; Tasumi et al., 2005), vegetation fraction (fc ) (Su, 2002), LST and α (Bastiaanssen, 1995), or only LST (Santanello Jr and Friedl, 2003). These empirical methods are suitable for specific conditions; therefore, estimating G, especially in this type of environment where NDVI values are low and thus G/Rn values are large, is a critical issue. The approach adopted here was drawn on Danelichen et al. (2014) who evaluated the parameterization of these different models in three sites in Mato Grosso state in Brazil and found that the model proposed by (Bastiaanssen, 1995) showed the best performance for all sites, followed by the model from Choudhury et al. (1987) and Jackson et al. (1987):

Bastiaanssen (1995):

$$G = Rn \times \overline{(LST - 273.16) \times (0.0038 + 0.0074\alpha) \times} ((LST - 273.16)(0.0038 + 0.0074\alpha)(1 - 0.98NDVI^4) \tag{10}$$

Choudhury et al. (1987):

[revised manuscript text omitted]

$$\sout{EF = \frac{LE}{Rn\text{-}G}} \quad SF = 1 - \frac{LE\_SPARSE_{t-FP}}{LE_{p-t-FP}} \quad\quad \sout{(17)}\ (16)$$

$$\sout{SF = 1 - \frac{LE}{LEpot}} \quad EF = \frac{LE\_SPARSE_{t-FP}}{AE\_SPARSE_{Ft-P}} \quad\quad (\sout{14}18)$$

All daily ET estimates were done for the 10 km × 8 km sub-image (LE_SPARSE~day~) and then were weighted by the corresponding daily footprint to get the daily ET of the upwind area (LE_SPARSE~day-FP~).

***Stress Factor (SF) method***

Assuming that the stress factor is constant during the day, the daily modeled ET (LE_SPARSE~day-FP~) can be expressed as the product of the instantaneous estimate of SF at the satellite overpass time and the daily potential evapotranspiration :

$$LE\_SPARSE_{day-FP} = (1 - SF)LE_{p-day-FP} \tag{15}$$

LE~p-day-FP~ was calculated as the sum of the half hourly modeled latent heat fluxes at potential conditions LE~p-30-FP~.

**4.2.1 *Evaporative Fraction method**

Under clear sky days, EF self preservation was revised by several studies. Hoedjes et al. (2008) showed that EF is almost constant during daytime under dry conditions whereas it follows a concave up shape under wet conditions. Hence, EF depends strongly on soil moisture as well as canopy fraction cover, but, it is nearly unrelated to solar radiation and wind speed, as shown by Gentine et al. (2007).

Consequently, theThe daily modeled ET total (i.e. (LE_SPARSE~day-FP~) can be expressed as the product of the instantaneous estimate of EF at the satellite overpass time and the daily modeled available energy AE_SPARSE~day~:

$$LE\_SPARSE_{day} = EF \tag{(19)(19)}$$

$$\times AE\_SPARSE_{day}LE\_SPARSE_{day-FP}$$

$$= EF \times AE\_SPARSE_{day-FP}$$

Daily cumulative available energy AE_SPARSE~day~ was computed from instantaneous modeled available energy (AE_SPARSE~t~) at the two satellite overpass times using the same approach detailed in Sect. 3.3.2 (Eq. (13) and Eq.applying equation (14)). Instantaneous estimates of Rn and G with the SPARSE model were used.

**4.2.2 Stress Factor (SF) method**

Assuming that the stress factor (SF) is constant during the day, the daily ET (LE). AE_SPARSE~day~) can be expressed as the product of the instantaneous estimate of SF at the satellite overpass time and was weighted by the corresponding daily footprint to get the daily potential evapotranspiration LEpot~day~:modeled AE of the upwind area AE_SPARSE~day-FP~.

$$LE\_SPARSE_{day} = (1 - SF) \times LEpot_{day} \tag{20}$$

LEpot~day~ was calculated as the sum of the half hourly modelled latent heat fluxes at potential conditions. The SF method is more complex than the EF method since inputs for the SF method have to be computed from a potential evapotranspiration model while inputs used for EF method can be derived from remote sensing.

**4.2.3 *Residual method**

Daily modelled latent heat fluxBesides, daily modeled ET (LE_residual_SPARSE~day-FP~) was also estimated as a residual term of the surface energy budget using daily modelledmodeled sensible heat flux (H_SPARSE~day-FP~) and available energy (AE_SPARSE~day~) totals~-FP~) as shown in Eq. (21).follows:

$$LE\_residual\_SPARSE_{day} \qquad\qquad (21)(20)$$
$$= AE\_SPARSE_{day}$$
$$- H\_SPARSE_{day} \; LE\_SPARSE_{day-FP}$$
$$= AE\_SPARSE_{day-FP} - H\_SPARSE_{day-FP}$$

H_SPARSE$_{day}$ was computed from modeled sensible heat flux (H_SPARSE$_t$) following the same extrapolation method used for the available energy (see Sect. 3.3.2). The corrected parameterizations of H were got from the comparison of daily measured sensible heat flux H$_{BS-day}$ computed as the average of half-hourly measured H$_{BS-30}$ and daily sensible heat flux (H$_{BS-day-Terra}$ and H$_{BS-day-Aqua}$) computed using the extrapolation method from instantaneous measured H$_{BS-t-Terra}$ and H$_{BS-t-Aqua}$ at Terra and Aqua overpass time, respectively (Equation 21).

$$H\_SPARSE_{day_{Aqua}} = a'_{Terra} \times Rg_{day} \frac{H\_SPARSE_{t_{Aqua}}}{Rg_t} + b'_{Terra} \; H_{BS-day-Terra} \qquad (22)(21)$$

$$= a'_{Terra} Rg_{day} \frac{H_{BS-t-Terra}}{Rg_{t-Terra}} + b'_{Terra}$$

$$H\_SPARSE_{day_{Terra}} = a'_{Aqua} \times Rg_{day} \frac{H\_SPARSE_{t_{Terra}}}{Rg_t} + b'_{Aqua} \qquad (23)$$

$$H_{BS-day-Aqua} = a'_{Aqua} Rg_{day} \frac{H_{BS-t-Aqua}}{Rg_{t-Aqua}} + b'_{Aqua}$$

where H$_{BS-t-Terra}$ and H$_{BS-t-Aqua}$ are  the instantaneous measured sensible heat flux in the Ben Salem flux station.

Therefore, the corrected parameterizations of H (Table 3), needed to remove the bias between measured (H$_{BS-day}$) and computed H (H$_{BS-day-Terra}$ and AE$_{BS-day-Aqua}$), were applied to compute daily  modeled H ( H_SPARSE$_{day}$) from instantaneous modeled H (H_SPARSE$_t$) following the extrapolation method shown in equation 21. Finally, H_SPARSE$_{day}$ was weighted by the corresponding daily footprint to get the daily modeled H of the upwind area H_SPARSE$_{day-FP}$.

**Table 3: Corrected parameterizations of sensible heat flux for the diurnal reconstitution**

| Terra | a'$_{Terra}$ | 1.02 |
|---|---|---|
|  | b'$_{Terra}$ | -17.31 |
| Aqua | a'$_{Aqua}$ | 1.00 |
|  | b'$_{Aqua}$ | -14.83 |

**5 Water stress estimates**

Water stress estimation is crucial to deduce the root zone soil moisture level using remote sensing data, (Hain et al., 2009). Water stress results in a drop of actual evapotranspiration below the potential rate. Its intensity is usually represented by a stress factor  as defined in Sect. 4.2, ranging between 0 (unstressed surface) and 1 (fully stressed surface).

Modeled values of SF at the time of Terra and Aqua overpass (SF$_{mod}$) have been computed from modeled potential LE (LEp$_{-t-FP}$) as follows:

$$SF_{mod} = 1 - \frac{LE}{LE_{pot}} = \frac{LST - Trad_{pot}}{Trad_{stress} - Trad_{pot}} \; \frac{LE\_SPARSE_{t-FP}}{LE_{p-t-FP}} \qquad (16 \, 24)$$

where LE_SPARSE~t-FP~ and ~~LE~pot~~LE~p-tFP~ are the modeled latent heat fluxes in actual and potential conditions, respectively~~, and Trad~stress~ and Trad~pot~ are simulated radiative temperature in actual and potential conditions, respectively; and LST is the MODIS land surface temperature~~.

Furthermore, surface water stress factor derived from XLAS measurement, named SF~obs~, at the time of Terra and Aqua overpass was computed as follows (Su, 2002):

$$SF_{obs} = \frac{\cancel{H\_XLAS_t - H_{pot}}\ H\_XLAS_t - H_{p-t-FP}}{\cancel{H_{stress} - H_{pot}}\ H_{s-t-FP} - H_{p-t-FP}} \tag{17\cancel{25}}$$

[revised manuscript text omitted]

---

## Referee Report (RR1)

Reviewer #2
Second Review:

**To the authors:**
Please, you need to understand that theory is one thing and hypothesis is a different scientific element. The so called MO similarity hypothesis is just a hypothesis not a THEORY. This hypothesis proposes that the set of non-linear equations you are using are valid only under homogenous surfaces and stationary flows. – please correct all instances in which MOST is mistaken and replace by MO hypothesis.
Going to your response #4 basically your answer is circumventing the question and what you are indicating is that the blending height will give you a safe interval to indicate the you are above all internal boundary layers that can be generated in the XLAS footprint and therefore you use either way the equations. But you had never demonstrated that this is the case. More importantly blending heights are defined including aerodynamic characteristics and no stratification is accounted for. So please make sure or provide example calculations to ensure you are above the blending height assuming this blending height is defined under conditions in which mechanical turbulence is larger than buoyancy driven turbulence.

**Response #8 is missing.**

**Detailed comments:**
**21: your answer still does not address the problem of distributed parameters. We agree on all those references you mentioned but you are still not given a convincing argument about using a model with concentrated parameters (R, V, I) over a problem that contains distributed parameters (roughness, soil temperatures, canopy temperatures, air temp, vapor pressure etc). Please give a reasoning to convince the reader that this model despite being a concentrate parameters model it represent distributed parameters of the land surface characteristics.**

**36 here I refer to the numerical convergence of the set of MO + CN2 equations that you solve at each step when you input CN2, Tamb, Pamb and Bowen Ratio to deduce H_XLAS. Now this is called iterative method. The article mentioned is an advanced version of how to articulate all the equations MO + CN2 and obtain H_XLAS through an analytical computation. This has nothing to do with the G. Also what I mentioned about SNR of CN2 has to do with the convergence factor in the traditional iterative calculation method for H_XLAS retrieval. You are right about CN2(z) this is explained in the article of Paulson that gives the sensitivity to terrain height since CN2 varies strongly and non-linearly with height. But my question here is about the numerical methodology you are using to integrate all equations each 30 min. based on the set of MO + CN2 equations.**

**37  please then clarify this in the text.**

**43: this needs to be demonstrated. You should provide at least an example analysis that no advection is present. Remember that advective flows or the presence on site of submeso flows can erase the scale gap in which all micromet observations are based on. And that this condition will put atmospheric surface layer flows outside the conditions of MO Hypothesis in which all your deductions are based. Please take a look at papers from Foken as I mentioned before.**

---

## Author Response (AR3)

**Authors Response to Reviewer 1 comments**

The Authors present an extensive work (reinforced by experimental data) aimed to assess the operational use of the Soil Plant Atmosphere and Remote Sensing Evapotraspiration (SPARSE) model and its accuracy by a comparison to the Scintillometric technique. I think that Authors address relevant scientific questions within the scope of HESS. Furthermore the paper is generally well organized and well written and there-fore the paper could be taken into account for the final publication after a moderate revision. Particularly, The Authors should improve the part of "Results and discussion" (pag. 16-20) with a better description of the validation of SPARSE model carried out with by comparing H and AE estimations with flux station and XLAS scintillometer (see comments n 7, 11 and 12). My comments and questions are as follow:

1. Lines 33-44: The Authors corroborated "the good correspondence between instantaneous H estimates and large aperture scintillometer H measurements" reporting RMSE values expressed in W m-2. As stated by the Authors (Line 418) "For hydrological applications, daily ET is usually required: : :." and in my opinion this means that for hydrological purposes the accuracy of daily evapotranspiration should be expressed in millimeters for day (mmd-1). Therefore in the abstract and through the paper this aspect should be considered and also critically analyzed. From my calculations the accuracy obtained by SPARSE model application should be around 1.6 mmd-1. Is this value "acceptable" ?

*Response:*

*Indeed, we agree with Reviewer 1 that for hydrological purposes the accuracy of daily evapotranspiration (ET) should be expressed in millimeters per day, however, the RMSE values mentioned in the abstract and throughout the paper are instantaneous sensible and latent heat fluxes estimates at the satellite overpass time and are not daily values, therefore, they are expressed in W.m-2. Since, they are instantaneous data, it should not be converted using this formula:*
*47.2 W.m-2\*0.0864/2.45= 1.66 mm/day*
*43.2 W.m-2\*0.0864/2.45= 1.52 mm/day*
*Therefore, we get an instantaneous LE error of about 0.1 mm/0.5.hour around the satellite overpass (around midday, at the max. ET rate)*
*In the revised version of the manuscript (section 6.4), when dealing with daily ET, all values are expressed in mm.day$^{-1}$; following the reviewer's suggestion, we added the model daily ET estimates accuracy (RMSE= 0.7 mm/day) similarly to what as been done for instantaneous results.*

2. Lines 87-88: Is "irrigation requirements" (generally expressed in mmd-1) a prerogative only "of RS-based SWB models" ? Please, clarify.

*Response:*

*Irrigation requirements are mainly estimated using RS-based SWB models, since irrigation is a component of the water balance equation on which is based SWB models. Indeed, the*

*crop coefficient method (FAO56 method) is currently the main method used for scheduling irrigations around the world (Glenn et al., 2007).*
*Irrigation requirement was rarely directly estimated using SEB models. Indeed, SEB outputs are generally actual evapotranspiration (its energy equivalent LE) and if Irrigation is estimated, it should be computed as a residual term of the water balance equation. Exception exists, for example, (Courault et al., 1998) used surface temperature derived from NOAA data and a SVAT model called MAGRET to find parameters linked to the irrigation over the agricultural region "la Crau" in South-Eastern France ; the predicted parameters were the beginning and the end of irrigation, frequency and water quantity diverted.*

3. Line 108: ": : :at the beginning of the process". Please clarify.

*Response:*

*This was corrected before review, and we did not find this expression in line 108 of the last article manuscript version "hess-2017-454-manuscript-version3_discussion" to which we refer. Indeed, in this last version, the mentioned sentence was written as follows: "at the beginning of the dry down".*

4. Lines 111-112: ": : :the lack of information about the actual irrigation scheduling adopted by the farmers is the critical limitation for SWB modeling". I believe that var-ious SWB models (Swap, Cropsyst, FAO56, AcquaCrop) are able to consider both scheduled by farmer irrigation (as input) or predicted irrigation (as output). Please, clarify or modify.

*Response:*

*Indeed, several SWB models such as Swap, Cropsyst, FAO56, AcquaCrop and also the SAMIR model that we have already used (Saadi et al., 2015) are able to consider both methods to take irrigation into account: either an estimated amount provided by the farmer (as an input) or a predicted irrigation with a module to trigger irrigation according to, say, critical soil moisture levels (as an output). We clarify this part in the revised version by saying that the lack of actual irrigation scheduling information does not impact the irrigation estimation by these models, since irrigation could be simulated by SWB models, but rather the validation protocol of irrigation requirements estimates (irrigation data is usually unavailable).*

5. Line 123: Insert ". . ." in dual-source models.

*Response:*

*In the version to which we refer this expression is already put in inverted commas (line 116): "However, separate estimates of evaporation and transpiration makes the "dual-source" models more useful for agrohydrological applications*

6. Lines 152-154: Clarify that the "layer" approach of SPARSE is essentially a "dual-source" scheme.

*Response:*

*In the revised version of the manuscript, the paragraph is simplified accordingly (line 180):*

*"In this study, (…) were obtained by the SEB method, using the Soil Plant Atmosphere and Remote Sensing Evapotranspiration (SPARSE) (...)."*

*We specify in the "model" section (section 4 line 465) that we use the "layer" approach and define it: "The SPARSE dual-source model solves the energy budgets of the soil and the vegetation. Here we use the "layer approach", for which the resistance network relating the soil and vegetation heat sources to a main reference level through a common aerodynamic level use a series electrical branching"*

7.    Line 187: The Authors should explain (also under a theoretical point of view) the choice to install Scintillometer at a 20 m height. About the experimental setup it is strange the absence of a "net radiometer" that, on the basis of the footprint analysis, could be installed in the average prevalent source area of footprint. The Authors could explain this fact.

*Response:*

*The choice to install Scintillometer at a 20 m height was based on the XLAS installation principle detailed in the "Kipp & Zonen LAS and XLAS instruction manual", indeed, the minimum installation height of the XLAS as function of the path length and for different surface conditions is graphically explained and shows that for a path length of 4km, the XLAS height of 20m is an adequate height since the XLAS is high enough to minimize measurement saturation and not too high to be representative of the 4km path Boundary Layer.*

*The absence of a "net radiometer" is explained by the high heterogeneity of the study area, especially in terms of vegetation cover; therefore, it is not possible to measure the net radiation (Rn) of all plots or even the Rn of "typical" plots (with similar land cover and irrigation practice).*
*This is clarified in the revised version (line 2014).*

8. Line 280: The terms "incoming solar radiation" and "incoming atmospheric radiation" are correct but could generate a misunderstanding. Please use the more classical "shortwave" and "longwave" terminology in eq. (9) and explain how RS data are generally used to solve balance equation of radiation (eq.9).

*Response:*

*In the revised version of the manuscript, the terms "incoming shortwave radiation" and "incoming longwave radiation" are used. This terminology is also used all along the manuscript.. The following paragraph is added accordingly (line 392):*
*"The Ben Salem meteorological station was used to provide $Rg_t$ and $R_{atm\text{-}t}$. Remote sensing variables α, LST, $ε_s$ and NDVI came from MODIS products"*

9. Line 367: About the "Temporal interpolation of albedo and NDVI" some brief details could be considered.

*Response:*

*Albedo MODIS products (MCD43) are available every 8 days and come from different satellite overpasses over a period of 16 days, the day of interest is central date. Both Terra and Aqua data are used in the generation of this product, providing the highest probability for quality input data and designating it as the acronym MCD, which means Combined product.*

*NDVI MODIS products (MOD13A2/MYD13A2 for Terra and Aqua, respectively) come from different satellite overpasses over a period of 16 days, and they are available every 16 days and separately for Terra and Aqua. Indeed, algorithms generating this product operate on a per-pixel basis and requires multiple daily observations to generate a composite NDVI value that will represent the full period (16 days), the 1km/16days MOD13A2 (respectively MYD13A2) product is an aggregated 250m/16 days MOD13Q1 (respectively MYD13Q1) product..*

*For both products, the data is linearly interpolated over the available dates in order to get daily data. For each pixel, the best data is taken into account (based on the quality index supplied with the product). Therefore, the temporal interpolation was done pixel by pixel.*

*This explanation is inserted in the revised version (line 248).*

10. Line 455: Which method has been used to evaluate the "potential conditions", please clarify.

*The half hourly potential latent heat flux is computed using the prescribed mode of the SPARSE model (see (Boulet et al., 2015)): " The system of equation can also be solved for Ts and Tv only if the efficiencies representing stress levels (dependent on surface soil moisture for the evaporation, and root zone soil moisture for the transpiration) are known. In that case the sole first four equations are solved. This prescribed mode allows computing all the fluxes in known limiting soil moisture levels (very dry, e.g. fully stressed, and wet enough, e.g. potential). (…) The potential evaporation and transpiration rates used later on are computed using this prescribed mode with minimum surface resistance to evaporation and transpiration, respectively."*

*The above paragraph is added to the SPARSE model description in the revised version of the manuscript (line 482).*

11. Lines 491-492: The Authors reported that . . .."An overestimation of about 15% is found between estimated and measured daily available energy. . ..and the coefficients . . .. . .were applied to remove this bias". If I well understand the above procedure (re- move of bias) is a sort of calibration of the output of modeled on the basis of observed flux station. Please clarify.

*Response: see response to comment 12.*

12. Lines 526-527: About the estimation of sensible heat flux the authors reported that "This result is of great interest considering that the SPARSE model was run with no prior calibration", but I feel a sort of contradiction with the bias removing procedure described in the above comment. Please clarify. Moreover I think that the Authors should describe the accuracy of model prior and after the bias correction.

*Responses to comments 11 and 12:*

*In fact, bias removal does concern neither the SPARSE model which was run with no prior calibration nor its estimates. Since the model provide a single instantaneous estimate of energy budget components, the global solar incoming radiation Rg was used to scale modeled AE and H from instantaneous to daily values (see section 4.2.3), the same applies to instantaneous available energy (see sections 3.3.1 and 3.3.2) computed using remote sensing and meteorological data (equation 9 ) and measured H by the XLAS.*

*Indeed, the extrapolation from an instantaneous flux estimate to a daytime flux assumes that the surface energy budget is "self-preserving" i.e. the relative partitioning among components of the budget remains constant throughout the day. However, many studies (Brutsaert and Sugita, 1992; Gurney and Hsu, 1990; Sugita and Brutsaert, 1990) showed that the self-preservation method gives day- time latent heat estimates that are smaller than observed values by 5-10%. Moreover, (Anderson et al., 1997) found that the evaporative fraction computed from instantaneous measured fluxes tends to underestimate the daytime average by about 10%, hence, corrected parameterization was used and a coefficient=1.1 was applied. Similarly, (Delogu et al., 2012) founded an overestimation of about 10% between estimated and measured daily component of the available energy thus, a coefficient =0.9 was applied. The (Delogu et al., 2012) corrected parameterization were tested, since, in our study case also an overestimation between estimated and measured AE was found, but this coefficient did not give consistent results, therefore, we had to calibrate the extrapolation relationship in order to get accurate daily results of AE and H*

*Thereby, the applied extrapolation method was tested using in situ Ben Salem flux station measurements. Indeed, Daily measured available energy $AE_{BS\text{-}day}$ (all the same for $H_{BS}$) computed as the average of half-hourly measured $AE_{BS\text{-}30}$, was compared to daily available energy ($AE_{BS\text{-}day\text{-}Terra}$ and $AE_{BS\text{-}day\text{-}Aqua}$) computed using the extrapolation method from instantaneous measured $AE_{BS\text{-}t\text{-}Terra}$ and $AE_{BS\text{-}t\text{-}Aqua}$ at Terra and Aqua overpass time, respectively (Equation 14). Results gave an overestimation of about 15 %. The corrected parameterizations of AE (Table 1), needed to remove the bias between measured ($AE_{BS\text{-}day}$) and computed AE ($AE_{BS}\text{-}_{day\text{-}Terra}$ and $AE_{BS\text{-}day\text{-}Aqua}$), were applied to compute daily remotely sensed AE ($AE_{day}$) from instantaneous AE ($AE_t$) following the extrapolation method shown in equation 14.*

*This explanation is inserted in the revised version (lines 419 to 450 and lines 542 to 554).*

13. Line 545: (Figure 7). Looking at the scatterplot it is clear a more dispersion for H value greater than 150. Is there an explanation of this?

*Response:*

*Possible explanations of the scatter observed or high H values are (revised version line):*

i) *the XLAS measurement saturation; according to the "Kipp & Zonen Las and XLAS instruction manual", for a path length of 4km and a scintillometer high of 20 m, saturation measurement problem might be present from H values of about 300 $W.m^{-2}$*

ii) *Uncertainties on the correction of stability using the universal stability function*

iii) *Potential inconsistencies between the area average MODIS radiative temperature and the air temperature measured locally at the meteorological station.*

14. Line 604: The Authors reported that "Daily observed and modeled ET over the whole study period were both in the range of 0-4 mm mm.day-1 which is consistent with the land use present in the XLAS pat". In my opinion this is a prosy comment, Trouble if not.

*Response:*

*We agree with the reviewer 1, and the composition of the vegetation cover over the study area (above the scintillometer) with detailed land use percentage is added (section 3.2), in order to show that this area is almost covered by fruit trees spaced by a lot of bare soil, with less herbaceous soil-covering crops; which lead to this range of daily ET. These ET values range was also found in (Saadi et al., 2015) dealing with the same study area. This is precised in the revised version (Figure 4).*

15. Line 616-617: The Authors reported that "Some points with little to null ET were recorded from May to July 2013 which can be explained by the very dry conditions and scattered vegetation cover with a considerable amount of bare soil". Why this behavior was not observed in the same period of 2014 ?

*Response:*

*This behavior was not observed in the same period of 2014, because 2014 was a rainy year in comparison to 2013 (more rainfall peaks), so, even supposing that the farmers have the same attitude and cultivate the same crop types between the two years (which is not true in the context of our study area and farmers always change crop types), precipitations favor the growth of spontaneous vegetation over fallows which contribute to ET rise. On the other hand, since the year experiences more rain, farmers cultivate a larger part of the land diversify the crop types and the vegetation cover is denser, this contributes to an overall increase in ET.*
*This explanation is inserted in the revised version (line 693).*

16. Line 863: Please check the (Minacapilli and Ciraolo, 2007) reference.

*Response:*

*This reference should be corrected as follows:*

*Minacapilli, M., Ciraolo, G., D Urso, G., and Cammalleri, C.: Evaluating actual evapotranspiration by means of multi-platform remote sensing data: a case study in Sicily, IAHS PUBLICATION, 316, 207., 2007.*

**References**

Anderson, M., Norman, J., Diak, G., Kustas, W., Mecikalski, J., 1997. A two-source time-integrated model for estimating surface fluxes using thermal infrared remote sensing. Remote sensing of environment 60, 195-216.

Boulet, G., Mougenot, B., Lhomme, J.P., Fanise, P., Lili-Chabaane, Z., Olioso, A., Bahir, M., Rivalland, V., Jarlan, L., Merlin, O., Coudert, B., Er-Raki, S., Lagouarde, J.P., 2015. The SPARSE model for the prediction of water stress and evapotranspiration components from thermal infra-red data and its evaluation over irrigated and rainfed wheat. Hydrol. Earth Syst. Sci. 19, 4653-4672.

Brutsaert, W., Sugita, M., 1992. Application of self-preservation in the diurnal evolution of the surface energy budget to determine daily evaporation. Journal of Geophysical Research: Atmospheres 97, 18377-18382.

Courault, D., Clastre, P., Cauchi, P., Delécolle, R., 1998. Analysis of spatial variability of air temperature at regional scale using remote sensing data and a SVAT model, Proceedings of the First International Conference on Geospatial Information in Agriculture and Forestry.

Delogu, E., Boulet, G., Olioso, A., Coudert, B., Chirouze, J., Ceschia, E., Le Dantec, V., Marloie, O., Chehbouni, G., Lagouarde, J.P., 2012. Reconstruction of temporal variations of evapotranspiration using instantaneous estimates at the time of satellite overpass. Hydrol. Earth Syst. Sci. 16, 2995-3010.

Glenn, E.P., Huete, A.R., Nagler, P.L., Hirschboeck, K.K., Brown, P., 2007. Integrating remote sensing and ground methods to estimate evapotranspiration. Critical Reviews in Plant Sciences 26, 139-168.

Gurney, R., Hsu, A., 1990. Relating evaporative fraction to remotely sensed data at the FIFE site.

Saadi, S., Simonneaux, V., Boulet, G., Raimbault, B., Mougenot, B., Fanise, P., Ayari, H., Lili-Chabaane, Z., 2015. Monitoring Irrigation Consumption Using High Resolution NDVI Image Time Series: Calibration and Validation in the Kairouan Plain (Tunisia). Remote Sensing 7, 13005.

Sugita, M., Brutsaert, W., 1990. Regional surface fluxes from remotely sensed skin temperature and lower boundary layer measurements. Water Resources Research 26, 2937-2944.

**Authors Response to Reviewer 2 comments**

| | General comments | Authors response |
|---|---|---|
| 1 | Depending upon editor's decision I would like to see further:
1) Figures with better accuracy in their representation. For example, some of them seems to have been the result of quick spreadsheet plots but without including accurate axis ticks, grids, labels, etc. | All figures are improved in the revised version. Particular attention is paid to axis ticks, grid and labels. |
| 2 | 2) Same as for the description of the figure captions and legends. The reader needs to understand a given figure by analyzing the figure and reading the information on the figure caption and legends. | Figures captions and legends are enhanced in the revised version of the article in order to provide complete information. |
| 3 | 3) A better explanation of the SPARSE methodology is needed, steps and the set of equations in the ET and H estimates. What the assumptions are and what is the physical framework? All of that is missing and therefore theoretically this paper is very weak.

For example, from where the authors got a threshold value of 30 W/m2 to start the iteration? How convergence is achieved is a mystery here and how many iterations and how signal-to-noise ratio of RS data plays a role in that convergence? Which equation provides convergence we don't know. | This article deals with an assessment of the SPARSE model accuracy and operational use in a semi arid context over a heterogeneous landscape; the theoretical framework of SPARSE is only summarized since it has been detailed in (Boulet et al., 2015) as well as in the online documentation (Boulet, 2017); since it is critical to have a self-understandable methodology section in the revised version of this article, we extend the explanation of the SPARSE methodology and add a diagram showing the flowchart of the SPARSE algorithm (Figure 5).

There is no iteration till convergence in the SPARSE algorithm, only a decision tree with decisions made upon the sign of the retrieved soil latent heat flux component: if negative, the assumption of unstressed vegetation is considered as invalid and the stress of the vegetation is retrieved. This is detailed in the added figure.

The $30 \mathrm{Wm}^{-2}$ is not a threshold to start iteration since there is not a convergence in SPARSE model, but it is a minimum positive threshold for vegetation stress detection which accounts for the small but non negligible vapor flow reaching the surface (Boulet et al., 1997). (Revised version line 492) |

| 4 | 4) I would like the authors to provide adequate justification to the use of formulas to deduce H based on LAS or XLAS. Particularly since the indicated formulas are valid only under the similarity hypothesis of Monin-Obukhov which implies homogenous surface and stationary flows. No justification was provided as for how these conditions were tested to render valid the resulting HLAS flux. | In our study area topography is flat, and landscape is heterogeneous only from an agronomic point of view since we find different land uses (cereals, vegetables and fruit trees mainly small olive trees with considerable spacing of bare soil); however, this heterogeneity in landscape features at field scale is randomly distributed and there is no drastic change in height and density of the vegetation at the scale of the XLAS transect (i.e. little heterogeneity at the km scale, most MODIS pixels have similar NDVI values for instance). In these conditions, considering the size of the surface changes in roughness (mean vegetation height ~1.5m), we assumed that the XLAS measurement height was close to the blending height, or either higher. Thus, the fluxes measured by scintillometry are area-averaged and MOST theory can be applied in the flux algorithm computation. In addition, support for the MOST theory was assessed by looking at non-dimensional diagrams of normalized $C_t^2$ and most points are aligned on the theoretical curves of Andreas and (De Bruin et al., 1993). On that basis, we believe that MOST is valid. |
|---|---|---|
| 5 | 5) when the authors discuss about uncertainties it is not clear what kind of uncertainties we are talking about and how have those been calculated? Moreover, uncertainties in heterogeneous terrain based on pure observations XLAS have not been computed. | Uncertainties concern mainly:
i/ the instantaneous remote sensing data: there is indeed an issue with the MODIS pixel heterogeneity and notably the distribution of components at the intersection between the square pixel and the XLAS footprint. Also, MODIS products, and mainly LST which is paramount in stress coefficient computation, are assumed to be reliable since we do not have means to reprocess them; however, results could be checked using Landsat high resolution TIR data.
ii/ half hourly forcing and XLAS data (meteorological and flux data);
iii/ the extrapolation method from instantaneous to daily results ;
iv) unlike temperate areas in which sensible hat flux H is relatively low, in our semi-arid study area, H is mostly high leading to important difference between H and LE (which approaches zero) requiring more data postchecking in the residual derivation of LE from XLAS.
v/ the empirical estimation methods of soil heat flux G (3 methods were tested) as well as the possible |

| | | |
|---|---|---|
| | | daily heat accumulation can lead to possible errors in available energy estimation and in turn in residual LE estimation, hence, both minimum and maximum daily observed LE were presented, the same for the modeled daily LE presented by error bars.
Despite all these possible uncertainty sources, our findings are reasonable compared to previous published results (SAMIR model,(Saadi et al., 2015). |
| | A reference is provided so that the authors can check on that.
Bai, et al., 2015. "Characterizing the Footprint of Eddy Covariance System and Large Aperture Scintillometer Measurements to Validate Satellite-Based Surface Fluxes. *Geoscience and Remote Sensing Letters, IEEE,* 12(5), 943-947, 2015. doi: 10.1109/LGRS.2014.2368580. | Thank you for this interesting reference on which we draw on to add a paragraph in the revised version discussing the uncertainties in heterogeneous terrain based on pure XLAS observations. |
| 6 | 6) Not clear where the EC flux comes into play. Also footprint functions for the scintillometers need to be accounted for. Reference on this element is provided below. | There are two EC stations located at the top of the towers (on the side of the XLAS emitter and receiver, respectively), which are used to process the XLAS data (initialization of friction velocity u* values and the Obukhov length Lo) and one EC station on the ground. This is detailed in the revised manuscript:
*i)Line 218: "two automatic Campbell Scientific (Logan, USA) eddy covariance (EC) flux stations were also positioned at the same level on the two water tower top platforms. Half hourly turbulent fluxes in the western and the eastern EC stations were measured used a sonic anemometer CSAT3 (Campbell Scientific, USA) at a rate of 20 Hz and a sonic anemometer RM 81000 (Young, USA) at a rate of 10 Hz, respectively. The western station data were more reliable with less measurement errors and gaps, hence, the western EC set-up was used initialise friction velocity u* values and the Obukhov length Lo in the scintillometer flux computation".*
*ii) Line 232: "In addition, an EC flux station, referred as the Ben Salem flux station (few tens of meters away from the meteorological station) was installed from November 2012 to June 2013 in an irrigated wheat field (Figure 2) measuring half hourly convective fluxes exchanged between the* |

| | | surface and the atmosphere ($H_{BS\text{-}30}$ and $LE_{BS\text{-}30}$) combined with measurements of the net radiation $Rn_{BS\text{-}30}$ and the soil heat flux $G_{BS\text{-}30}$. Net radiation and soil heat flux measurements were transferred to the meteorological station from June 2013 till June 2015. Since, there are no Rn and G measurements in the two water towers EC stations, $Rn_{BS}$ and $G_{BS}$ measurements were among the inputs data to derive sensible and latent heat fluxes from the XLAS measurements. In addition, measured available energy ($AE_{BS}=Rn_{BS}—G_{BS}$) and $H_{BS}$ were used to calibrate the extrapolation relationship of the available energy and the sensible heat flux, respectively" |
|---|---|---|
| 7 | 7)  I would like the authors to provide an in-depth description of physical processes explaining the results in the final figures. Description of what is being presented in the figures is fine but we need more science here. | In the revised version, more physically-based explanation dealing mainly with the outliers is added to describe the final figures. |
| 8 | As an aside note the use of XLAS is not unique in this problem. A LAS can do 5 km max. Optical beam path and resolve the same situation. What is critical with using XLAS is beyond 5 km optical path. | |

| | **Detailed comments** | Authors response |
|---|---|---|
| 1 | Line 45 –off : please put references in chronologic order. This is the proper way to recognize previous work; unless specific discussions are provided which in those cases the trail of references needs to be broken down. This note is valid through the entire paper. | References are put in chronologic order in the revised version. |
| 2 | Line 50: About the claims about water scarcity related to climate change. -or better say climate variability: I wonder how compelling are these claims? – Can the authors substantiate in more details about this problem in this area? This is an important claim and need to be fully addressed by the authors to build context to this research and the methodologies being used. | The paragraph below is added in the revised version (line 50): *"Indeed, the Mediterranean region is one of the most prominent "hot spots" in future climate change projections (Giorgi and Lionello, 2008) due to an expected larger warming than the global average and to a pronounced increase in precipitation inter-annual variability. The major part of the southern Mediterranean countries, among others Tunisia, already suffer from water scarcity, and show a growing water deficit, due to the combined effect of the water needs growth (soaring demography and irrigated areas extension), and the reduction of resources (temporary drought and/or climate change)"* |
| 3 | Line 53: the use of "greatest" here tries to indicate what? "the larger" or "the most important"? This needs to be clearly understood without ambiguity and therefore we need to bring more specificity. | *"greatest"* is replaced by *"the largest"* in the revised version (line 59) |
| 4 | Line 56: I'll add complexity in. As we move from ecosystem scale to landscape scales surface heterogeneity but also dynamic of the flow, cloudiness, precipitation come into play more aggressively. This also bring more context to the need of this study. | We have already mentioned the impact of land cover heterogeneity at large scale on the land atmosphere exchange: *"Moreover, at these scales, land cover is usually heterogeneous and this affects the land-atmosphere exchanges of heat, water and other constituents (Giorgi and Avissar, 1997)."* However, to develop this idea further, in the revised version, we provide some more explanation about the hydro-meteorological processes complexity and its impact on climate variables (line 61): *"(…)it is much more difficult at larger scales (irrigated perimeter or watershed) due to the complexity not only of the hydrological processes* |

| | | |
|---|---|---|
| | | *(Minacapilli et al., 2007) but also of the hydro-meteorological processes. Indeed, at landscape scale, surface heterogeneity influences regional and local climate, inducing for example cloudiness, precipitation and temperature patterns differences between areas of higher elevation (hills and mountains surrounding the Kairouan plain) and the plain downstream. Moreover, at these scales, land cover is usually heterogeneous and this affects the land-atmosphere exchanges of heat, water and other constituents (Giorgi and Avissar, 1997).* |
| 5 | Line 61: I would disagree that "RS techniques becomes essential". Basically it has been demonstrated that plot (or ecosystem) exchanges within same complex canopies do verify consistent differences in sensible heat fluxes (the simplest and ubiquitous flux on earth) over distances that are much smaller than the RS footprint in particular MODIS. See Starkenburg et al., (2015). Starkenburg et al. 2015: "Temperature regimes and turbulent heat fluxes across a heterogeneous canopy in an Alaskan boreal forest". J. Geophys. Res. Atmos., 120: 1348–1360. doi: 10.1002/2014JD022338

Now, I do agree that RS brings a mean to deduce, within certain ranges, an approximation of fluxes. What about mesoscale models? Or perhaps you wanted to indicated physical models using RS data as input? In any case, I think you should open this perspective here since there are other disciplines other than Remote Sensing Researchers that can also provide the same product. | Remote sensing (RS) can provide estimates of large area fluxes in remote locations, but those estimates are based on the spatial and temporal scales of the measuring systems and thus vary one from another. Hence, one solution is to upscale local micrometeorological measurements to larger spatial scales in order to acquire an optimum representation of land-atmosphere interactions (Samain et al., 2012). However, such upscaling is not always possible and results might not be reliable in comparison to the RS distributed products.
In order to keep the introduction as short as possible, in the revised version, two examples of complex physically based LSMs using RS data as inputs to derive ET are mentioned (line 76) |
| 6 | Line 63: vegetation physical properties or characteristics? | In the revised version:
"*vegetation's physical properties*" is replaced by "*vegetation physical characteristics*" (line 72) |

| 7 | Line 65: Authors use "plot" as one of the scales in which I assume results would be obtained. However, at no point plot-scale was defined. Please whenever plot is used for the first time in the Introduction section for example please clarify that. (excluding the abstract). | We agree with Reviewer 2 and the word "plot" induces ambiguity. "*plot*" is replaced by "*field*" in the revised version. (line 75) |
|---|---|---|
| 8 | Line 87: please rephrase the text between parenthesis. | In the revised version:
"*(mostly derived from, say, actual water content in the root zone, wilting point and field capacity)*"
is replaced by:
"*mostly derived from the soil moisture characteristics: actual available water content in the root zone, wilting point and field capacity*"(line 107) |
| 9 | Line 93: Spell out FAO. If it is not being used anymore in the text, then no need to define an acronym. | In the revised version:
"*FAO guidelines*" is replaced by
"*Food and Agriculture Organization-FAO guidelines*" (line 113) |
| 10 | Line 98-99: get rid of parenthesis here. What is inside is part of the phrase. | Parentheses are removed in the revised version. |
| 11 | Line 102: FAO-56 put a reference here. Or make a short phrase explanation. | The Allen et al. (1998) reference is added in the revised version. |
| 12 | Line 103: what is "dry down"? please make sure you check consistency in all phrases. | "*Dry-down period is the period after rain or irrigation where the soil moisture is decreasing due to evapotranspiration and drainage. It is of great interest, because soil moisture has such a strong effect on nearly every aspect of the land surface (heat distribution, albedo, carbon uptake… etc.).*"

This short explanation is added to the revised version (line 123). |
| 13 | Line 114: What's the meaning of adding quotes here? If single-source means single source, then no need for quotes. Quotes are used when you use a word or combination of words but you would like to indicate a different meaning. | Quotes are removed for *single-source models* and *dual-source models*. |

| | Line 116: same as 114. | |
|---|---|---|
| 14 | Line 117: comma missing before etc. | It is rectified in the revised version. |
| 15 | Line 128: add "they provide area-averaged sensible heat flux" | "*average sensible heat estimates*" is replaced by "*area-averaged sensible heat flux*" in the revised version (line 154). |
| 16 | Line 130-131: incomplete phrase. And, can you elaborate a little bit more here? | This phrase is rectified in the revised version as follows (line 156): "*Scintillometry can provide sensible heat using different wavelengths (optical wavelength and microwave wavelength ranges), aperture sizes (15-30 cm) and configurations (long-path and short-path scintillometry )*" . |
| 17 | Line 132: delete space before comma. | This is rectified in the revised version. |
| 18 | Line 133: representative of the pixel? It may be the case that for a particular MODIS data your scintillometer data intersects several pixels. Then we are talking about several pixels. | Indeed, the issue of the representativity of the heterogeneity (land use and irrigation practice) at the intersection between the MODIS pixels considered as homogeneous and the XLAS footprint was not discussed in the submitted version of the article. We add the suggested reference and discuss the relative percentages of Land Use classes within each MODIS pixel to provide a first guess on these relative heterogeneities. (line 329) |
| 19 | Line 140: **large-scale area-average** this is the proper measurement that one obtains from a scintillometer. | In the revised version: "*Since the scintillometer only provides spatially averaged sensible heat flux (…)*" is replaced by "*Since the scintillometer only provides large-scale area-average sensible heat flux (…)*" |
| 20 | Lines 140-143: Here I need help. Are you indicating that to get ET large-scale area-average you use XLAS? But you need to assume a closure fraction or assume is 100% Energy Balance closure. As we increase surface heterogeneity and the atmospheric flow acquires an increased space-time variability then it is difficult to assume 100% energy balance closure. How you do then? Please explain how you treat and eventually circumvent this problem. See for example Foken et al., (2006; 2010) and Foken (2008). Foken, T., F. Wimmer, M. Mauder, C. Thomas, and C. Liebethal, 2006. Some aspects of the energy balance | Please see authors' response to the general comment N°4. |

| | | |
|---|---|---|
| | closure problem. *Atmos. Chem. Phys.*, 6, 4395–4402.
Foken, T., 2008: "The energy balance closure problem: An overview", *Ecol. Appl.*, 18(6), 1351– 1367.
Foken, T., M. Mauder, C. Liebethal, F. Wimmer, F. Beyrich, J.-P. Leps, S. Raasch, H. A. R. DeBruin, W. M. L. Meijninger, and J. Bange, 2010: "Energy balance closure for the LITFASS- 2003 experiment", *Theor. Appl. Climatol.*, *101*(1-2), 149-160, doi: 10.1007/s00704-009-0216-8. | |
| 21 | Line 146: what is the "layer" approach? Can you be more explicit and detailed? If layer is the name of the approach, then no need to use quotes. | Indeed "*layer*" is the name of the approach, hence, the quote are removed in the revised version. More details about this approach is given in (Boulet et al., 2015) |
| 21 | Line 147: when authors normally explain the use of electrical resistance as equivalent models really are not paying attention to the details. So then now you need to explain how you transform an electrical element such as a Resistor, which is a concentrated parameter into a distributed vegetation or soil representation. What are the assumption? Hypothesis? Regions where this approximation is valid and where it fails, etc. I'll give you a hint R=V/I where V(electrical voltage: what is imposed the potential) and I(electrical current, what flows between the boundaries). Then when you say you use Rsoil and Rveg. What are the analogs of V and I here? What R actually means? And how you walk out from the Ohm's Law for concentrated electrical parameters and transition to our problem where these parameters are distributed?
This comes from Norman and Kustas TSEB- way before SPARSE.
For example, here it is important to remark that vegetation information has to be at much higher resolution | The resistance scheme is detailed in Boulet et al. (2015) and is similar to that used in (Kustas and Norman, 1999), cf. (Monteith and Unsworth, 2007). V is either a temperature difference (soil-aerodynamic level or vegetation-aerodynamic level) or the corresponding vapour pressure difference. I is the flux component (sensible or latent) and R is the resistance to transfer (aerodynamic resistances within and above the vegetation, stomatal resistance). There is no need of specifying a soil resistance to evaporation because the evaporation rate is directly retrieved. The Series description of the electrical analogy used here is that of most LSMs following (Shuttleworth and Wallace, 1985) which describes the interactions within the soil-plant-atmosphere interface for sparse crops. The radiation interception by sparse crops might be difficult to represent with a layer approach, this will be further commented in the text. |

| | | |
|---|---|---|
| | than the radiometric information to account for vegetation/forest variations for example the existence of clear areas within the forest or cultivars. How the authors account for that needs better explanations. And, what assumptions underlain these approximations? | |
| 22 | Line 150: I wanted to be clear here that XLAS ONLY can deduce sensible heat not LE. Please make sure this thread is conveyed all the way through your work. | In the revised version (line 183): *"The main objective of this paper is to compare H and LE obtained using the SPARSE model and XLAS (...)"* is replaced by: *"The main objective of this paper is to compare the modeled H and LE simulated by the SPARSE model with, respectively, the H measured by the XLAS and the LE reconstructed from the XLAS measurements acquired during two years over a large, heterogeneous area."* |
| 23 | Line 158: put "(" to indicate the reference the cultivars are within the phrase. | This is rectified in the revised version. |
| 24 | Line 173: what "double device" means for you. Please be specific. | This phrase is simplified in the revised version and "*double device*" is removed. (line 205) |
| 25 | Figure 2: it is not clear where the XLAS emitter and receiver are specifically located. Put a dot or a symbol to indicate that. Photos actually say nothing here. Now I see that the CSAT is close to the XLAS receiver. I would caution the authors here that any interpretation between XLAS fluxes and EC-CSAT fluxes would not be representative since the EC system is closer to the XLAS receiver and/or transmitter for that matter is the same. More importantly what is not clear here is what are the green contours indicating the footprint? And if these are EC footprint more likely are wrong. Please specify what SPOT5 bands 1,2,3 are in terms of wavelengths and they are used in this work. | Green contours are half-hourly XLAS footprints for selected typical wind conditions. High resolution SPOT5 image of 9[th] April 2013 was only used as background image to illustrate the land cover under the XLAS transect. Hence, figure 2 caption is modified in the revised version as follows: *"XLAS set up: XLAS transect (white), for which the emitter and the receiver are located at the extremity of each white arrow, half-hourly XLAS footprint for selected typical wind conditions (green), MODIS grid (black), orchards (blue) and the location of the Ben Salem meteorological and flux stations. Background is a three colour (red, green, blue) composite of SPOT5 bands 3 (NIR), 2 (VIS-red) and 1(VIS-green) acquired on 9th April 2013 and showing in red the cereal plots".* On the other hand, EC station flux measurements are not compared to XLAS fluxes along the article. This EC station utility has been already explained in the above responses (general comment N°6). |

| | | |
|---|---|---|
| 26 | Line 196: I would write Extra Large Aperture Scintillometer (XLAS) | This is rectified in the revised version. |
| 27 | Line 198: Phrase: "Scintillometer is based on the scintillation method" what is this? | This is rectified in the revised version. |
| 28 | Line 198-200: What is the cause and what is the effect? This phrase is wrong please think about a little bit. | This is rectified in the revised version as follows (line 269): *"Scintillometer measurements are based on the scintillation theory; fluxes of sensible heat and momentum cause atmospheric turbulence close to the ground, and create, with surface evaporation, refractive index fluctuations due mainly to air temperature and humidity fluctuations (Hill et al., 1980)""* |
| 29 | Line 205: replace "bean" by "beam" | This is rectified in the revised version (line) |
| 30 | Line 204: The reference that links scintillations and Cn2 is given by Tatarskii. We need to give the proper reference here. The fact that those references have been using it doesn't mean they were the ones given the foundation for this relationship. We need to make sure we give proper value to the actual references. | (Tatarskii, 1961) reference is added to the revised version (line 275) |
| 31 | Line 206: symmetrical to what? What is that symmetry you are talking about? | *This sentence is corrected (line 275): "The sensitivity of the scintillometer to $C_{n^2}$ along the beam is not uniform and follows a bell-shape curve due to the symmetry of the devices. This means that the measured flux is more sensitive to sources located towards the transect centre and is less affected by those close to the transect extremities."* |
| 32 | Line 208: get rid of an extra space in the phrase. Same line: "structure parameter of temperature" by structure parameter of temperature turbulence (refractive index in the case of CN2). | This is corrected in the revised version (line). |
| 33 | Line 210-212: here the authors mentions very cursory a very important problem which is the variation of Cn2 because of the beam height variation across the landscape. It seems this is one point you should be more cautious in bring some | The terrain is very flat; therefore there is little beam height variation across the landscape, except for what is induced by the various roughness heights of the individual fields. Since the interspace between trees is large, the effective roughness of the orchard is not significantly different from that of cereal fields, and far below |

| | | |
|---|---|---|
| | references and eventually limit your study on the basis of this sensitivity parameter. | the measurement height. |
| 34 | Line 213: only sensitive to temperatures. Add a period in the phrase. | This is corrected in the revised version. |
| 35 | Eq. [1] you introduce here an approximation that then you'll use as an equality. Please explain and substantiate or directly correct the equation. Also, I wonder how much beta introduce error, in this case, a semi-arid environment. | This is corrected; an equality sign is used in Eq. 1. The sensible heat flux dominates the energy balance in most cases; therefore the Bowen ratio is mostly above one. The influence of the beta correction has been analyzed in (Solignac et al., 2009) which shows that since the beta closure method does not rely on an exact locally observed beta it is far less sensitive to the precision on beta. |
| 36 | Line 217: iterative methods have intrinsic convergence and resolution errors. You have to specify the convergence error and also how the average of Cn2 gives you a signal with enough SNR to keep the specific convergence factor. Now recently analytical methods have been developed that integrate the set of nonlinear equations in this casa Tatarskii and Monin-Obukhov similarity hypothesis set. See Gruber and Fochesatto, (2013). Gruber M. A. and G. J. Fochesatto. 2013: "A New Sensitivity Analysis and Solution Method for Scintillometer Measurements of Area-Average Turbulent Fluxes" *Boundary-Layer Meteorology*, 149:65– 83 DOI 10.1007/s10546-013-9835-9 | I'm not sure to fully understand the reviewer's remark. Actually, as shown by (Gruber and Fochesatto, 2013), the height z at which $C_T^2$ is sampled can substantially affect the sensible heat flux (20%), but in our study, the *in situ* G measurement (used to initialize the energy budget closure) has also an impact on the estimate of H_XLAS throughout the convergence algorithm. Since XLAS measurements were processed at the beginning of the project, no sensitivity analysis of theses variables, *e.g.* effective height z, initial guess of the iterative algorithm (local *vs* integrated *via* remote sensing or modeling) was performed. As it is not the scope of the paper, we didn't achieve any sensitivity analysis on XLAS fluxes computation to determine which parameter has the strongest influence on the flux uncertainty. |
| 37 | Line 220: Zlas is a function where is that?

Andreas parameterization might not be valid for your site.- Can you justify here?

Zv: is the average canopy height but | $Z_{LAS}$ is not a function, since the XLAS experiment took place over a flat surface, $Z_{LAS}$ is the XLAS height; the word "*effective*" is therefore removed because it induces confusion.

We indeed test the De Bruin (De Bruin et al., 1993) parameterization in the revised version (cf. Figure above). |

| | weighted by the extension of the plots? | Zv estimation method is detailed by the end of section 4.1. It accounts for the various heights within the footprint selected using angular zones originating from the centre of the transect, and supported by high resolution remote sensing data. |
|---|---|---|
| 37 | Eq.4 contains u* but it is not clarified here from where this is taken. Here we can conclude that XLAS ONLY measures T* as a large-scale area-average variable but u* is a local variable or at least a variable measured at the scale of the EC system which is not the same as the XLAS. Explain please? | u* is not taken from EC system, it is computed based on an iteration approach in the beta closure method, only the initialization value of u* was taken from the EC station positioned on the western water tower. |
| 38 | Line 225: rho is the air density and cp here are considered constants. Do they vary across the experiment? | Indeed, air density, pressure and temperature depend on the location on the earth, on altitude and on the season of the year. However, in our study, standard values of air density ($\rho$) and air specific heat at constant pressure (cp) were used without verifying their variation across the experiment since our study concerns a limited extent (10 km*8 km, same earth location) with flat terrain (no altitude variation) and without a considerable temperature difference between the hot and cold seasons (average monthly temperature oscillates between 10°C and 28°C). |
| 39 | Line 227: nomenclature is Number[space]unit. please correct all the way your text. | This is rectified in the revised version. |
| 40 | Line 228: change "circa" by "near". The correct use of "circa" in English is to indicate something that happened in the past (circa, 1000 AD) for example. | This is rectified in the revised version. |
| 41 | Line 230: how many "aberrant" values you have in the entire dataset. Please give more precision to the signal processing so that researchers can compare their work with yours in the future. | The following paragraph is added to the revised version (line 306): *"Furthermore, half hourly H_XLAS aberrant values due to measurement errors and values higher than 400 $Wm^{-2}$, arising from measurement saturation, were ruled out (3% of the total measurement throughout the experiment duration)"* |
| 42 | Line 247: and also gives the major sensitivity to H. See also (Gruber et | Again, the terrain here is very flat and does not induce any disturbance linked to topography. |

| | | |
|---|---|---|
| | al., 2014) for the specific analytic derivation of the sensitivity to the topography height. Gruber, M. A., G.J. Fochesatto, O.K. Hartogensis, and M. Lysy. 2014: "Functional derivatives applied to error propagation of uncertainties in topography to large-aperture scintillometer-derived heat fluxes". *Atmos. Meas. Tech.*, 7, 2361-2371, doi:10.5194/amt-7-2361-2014, 2014. | |
| 43 | Equations 7 and 8: assume closure of energy balance at 100% please explain how this is possible. And what are your assumptions that lead to this approximation and what is the uncertainty in this assumption. | Please see authors' response to the general comment N°4. There is no large scale advection of heat and the XLAS is located above the blending height, therefore we expect that the 100% energy closure assumption is valid. |
| 44 | Line 271: Here the authors give an estimation of G/Rn energy partition that is known to be variable not only across a given landscape but also across landscapes. This needs to be carefully estimated. This goes from 31% to very low values in dense canopies. Please be more specific and give values of this factors across all your landscapes. | Indeed G estimation was the most uncertain variable in this study, and that's why we tested three methods to compute it since based on in situ data, we generally found an accumulation of G and the daily G is rarely zero. This part is discussed in the revised version (line 365). |
| 45 | Line 284: change "meteo" by "meteorological station". | This is rectified in the revised version |
| 46 | Lines 280-290: Here the authors bring parameterizations of G. And certainly it is appreciated this compilation. However, it would be best to have a discussion of how one of these parameterization is or may result more optimal for this work. It seems all the formulas were found and then tossed in this article to see what happens. – So compare your environment with the environment in which those | We used standard relationships used in models such as SEBS (Su et al., 2001). An overview of the validity of the relationship for the sole Ben Salem EC station (cereal) is illustrated in the revision (line 384). |

| | | |
|---|---|---|
| | parameterizations were developed and then decide or make some arguments about how to best use or adapt any of these parameterizations. | |
| 47 | Line 294: basically with the current satellite technology we cannot estimate diurnal cycles. However, you must know that at higher latitudes Aqua and Terra have at least six-passages a day. | We agree with Reviewer 2. |
| 48 | Line 300: I don't understand why the authors propose a=1 and b=0 and then find motivation on finding that actually these are not zero. The approximation of Rn by SW (Short Wave Downwelling) is known in micrometeorology and only works to some extent in clear skies when Rn is dominated by SW downwelling. I mean Rn can be negative but never SWdown. So, the way this paragraph is written possess a problem since it is not physically correct. | This paragraph as well as the associated result section (6.1) is rephrased in the revised version (lines 419 to 450 and lines 542 to 554) |
| 49 | Line 304: How you weigh the 10x8 km images data by the footprint? What kind of functions are used here to compute the footprint. Please explain. | Daily footprints were computed as a weighted sum of the half hourly footprints by the XLAS sensible heat flux. Weighing the 10x8 km images data by the footprint means multiplying the 10x8 km result grid by the footprint (weight coefficients ranging from zero and one). |
| 50 | Line 310: replace the "temperature of soil" by "soil temperature". | This is rectified in the revised version |
| 51 | Here you mention a "reference height" and simultaneously we are talking about a heterogeneous canopy and soil and canopy. Where is that reference height? And what are the assumptions and approximations you are taking by taking this assumption. For example, you are considering some variables at soil level but others at canopy level. How the reference height represents | Reference height here is the measurement height of the meteorological forcing (2.32 m). This is precised in the revision. |

| | both? And what are the assumptions in terms of physical processes? | |
|---|---|---|
| 52 | Eq. [15] you have here a radiative balance equation where it is assumed (without indication) that emissivity (on the left hand side ) is =1. Also this equation needs a reference level and a specific condition for the fluxes to be added and represented at the reference level. Please make sure you are accounting for all these so that the reader can fully understand what your assumptions are and where and under what conditions your analysis is valid. | Details are added to the revised version (line 467). |
| 53 | Line 319-320: is SPARSE better than TSEB? Can you give a little bit more explanation here? TSEB has modes to trait vegetation ALEXI and DIS-ALEXI. Are you saying that by incorporating aerodynamic functions makes SPARSE better than TSEB? Please clarify here what's the extent and implication of your comment on the paper. | A detailed intercomparison study between TSEB and SPARSE based on several flux stations is underway, first results indicate that bounding the fluxes simulated by both models by the potential rates given by SPARSE improves the performance of both models which have otherwise similar performances, though constrasted for the various cover types. In SPARSE the aerodynamic functions are those used in almost all Land Surface Models. ALEXI and DIS-ALEXI rely on coarse scale (few km) MSG data, and intercomparison of the ALEXI ET product and the scintillometer will also be carried out in the next future. |
| 54 | Line 325: from where you got the 30W/m2 minimum value? In some environments this will be three times G. Please justify this value. | Please see authors' response to the general comment N°3. |
| 55 | Line 334:335: Here we need to be more specific. What data is from bibliography and what data comes from RS? Please be specific. | After this sentence, bibliography, remote sensing and in situ data are detailed in the following paragraphs, however, in order to be more clear, this section will be rephrased in the revised version. |
| 56 | Line 343: Why you define an acronym MRT that is not used anymore? Acronyms that are not mentioned in the text anymore are unnecessary. | Rectified in the revised version |

| 57 | Line 343-347: this phrase is too long and badly constructed. | This paragraph is reworded in the revised version. |
|----|----|----|
| 58 | Line 349: We need more detail here. How many days or cases have been excluded from the entire dataset. We need to know how critical is this problem. Because if it is critical then it renders the method useless. | 360 daily data were excluded from the total daily data (1033 days), the following sentence is inserted in the revised version: *"(...) hence, days with missing data in MODIS pixels regarding the scintillometer footprint (35% of the acquired data) were excluded"* |
| 59 | Line 355: k1.15 need space. | Rectified in the revised version |
| 60 | Line 357: explain clump-LAI measurements. | Clump LAI is the value of the LAI of an isolated element of vegetation (tree, shrub...); if this element occupies a fraction cover f and is surrounded by bare soil, then the clump LAI value is simply equal to the area average LAI divided by f. This is specified in the revised version (Line 402). |
| 61 | Delete the word "Bibliography" from Table 1. That column is for sources and a journal peer review is a source. | Rectified in the revised version |
| 62 | Line 379: "overpasses" | Rectified in the revised version |
| 63 | Line 383: The second step need a more substance. How come you are running a 30 min fluxes based on a single TIR input? This will result in diurnal cycle of fluxes that are totally biased. I would say that this approximation is only valid for time-intervals in which the turbulence conditions are not too different form the TIR observations. | Indeed, the SPARSE model was run at a half hourly time step using the half hourly meteorological measurements ; assuming that either the stress factor or the evaporative fraction are invariant during the same day, the diurnal modelled fluxes are accounted for by recovering the diurnal course of either potential ET or available energy AE. Running the SPARSE model at half hourly time step is only done to get half hourly latent heat flux in potential conditions LEpot which is equivalent to a reference evapotranspiration whose calculation depends only on half hourly climatic data. This LEpot is used later when computing daily LE based on the stress factor method (section 4.2). This is better explained and more detailed in the revised version (line 508). |
| 64 | Line 396: please revise the following | Rephrased in the revised version. |

| | | |
|---|---|---|
| | wording "…complementary part to 1…" | |
| 65 | Section 4.2 seems to go around and around the subject without going down to the specifics. I think is necessary to simplify the description of methods. | Rephrased in the revised version. |
| 66 | Line 407: how you define the wet conditions here? Rain through the day, a specific amount of mm? please be more specific here. | Wet conditions are defined on the basis of a significant amount of rain recorded in the previous day (more than 5 mm). This is clarified. |
| 67 | Eq. [21] assume 100% energy balance closure. You need to justify the use of this condition. | Please see authors' response to the general comment N°4. |
| 68 | Line 429: "deduce" instead of "deduct". | Rectified in the revised version |
| 69 | Fig. 5. This figure is a very low quality without precision in the axis. Also we see only RS data here while it is announced XLAS data. | Please see authors' response to the general comment N°1. |
| 70 | Line 475: "convolving" Convolution has a very specific meaning in mathematics. Please verify the use of this term here. | In the revised version:

 *"By convolving the XLAS footprint with the SPARSE derived H, we were able to compare the modelled values ($H\_SPARSE_{t\text{-}FP}$ ) with the XLAS measurements ($H\_XLAS_t$ )."*

 is replaced by

 "*SPARSE derived H was weighted by the XLAS footprint in order to be able to compare the modeled values ($H\_SPARSE_{t\text{-}FP}$) with the XLAS measurements ($H\_XLAS_t$)*" |
| 71 | Same for the use of modelled or modeled. Both expressions are fine however if your choice is to use words in British English (in this case | Rectified in the revised version |

| | modelled) you have to be consistent all the way through your paper. | |
|---|---|---|
| 72 | Line 477: "dots"? seriously? | Rectified in the revised version |
| 73 | Line 478: Why these two days? Please give the reasons why you are specifically using those days. This is important because when scientist reading your paper would like to reproduce your results they will find no framework to produce such comparisons. | Selection criteria are added to the revised version (line 578):
- Day 2013-86 (24 March 2013) is in the cold season and day 185-2014 (4th July 2014) is in the warm season in order to highlight the land cover impact on LST and thus on modelled H (trees and rainfed and irrigated cereals in winter vs. only irrigated trees and vegetables in summer).
- Day 2013-86 (24 March 2013) shows footprint of strong south wind while the footprint of day 185-2014 is of a light north wind |
| 74 | Figure 6. I don't understand the coordinates (Y-axis and X-axis). Also the contours of XLAS footprint have no indications. | Figure 6 as well as its caption is improved in the revised version |
| 75 | Line 482: what you mean by "hot pixel"? Please avoid jargon in the writing. | Hot pixel systematically means a pixel with high LST and low NDVI.
A short explanation is added to the revised version. |
| 76 | Line 489: In general models are calibrated based on EC systems and thus the deduced large-scale area-average fluxes derived from satellite remote sensing is controlled by LAS observations. | Indeed in this study, SPARSE model was run in an operational way at landscape scale without parameters calibration, since in our study area, we do not have EC station for each crop type. However, SPARSE results at field scale were already compared to EC measurement in an irrigated wheat field and a rainfed wheat field in (Boulet et al., 2015) |
| 77 | Line 490-500: In general, as the heterogeneity in vegetation, soil and eventually in topography leading to variables flows increases the divergence increases. There though cases in which even EC systems that are placed together at distance shorter than the convective ABL development verify more than 50/m2 differences (Starkenburg et al, 2015). So then results expressed here are within the range of reasonable values.
The only one physical explanation | Please see authors' response to the general comment N°4. |

| | | |
|---|---|---|
| | why the LAS path by being longer would give different results is when the heterogeneity is such that the BL that develops integrates patches of different thermodynamic and turbulent properties. Then, the mention of issue is interesting but without a correct explanation is useless. | |
| 78 | Figure 7. contains features that are important to discuss since there is a change in the bias as function of the flux level. I wonder the authors to discuss this aspect from the physical aspects of the processes dominating this scale integration. | This part is improved in the revised version. Indeed, possible explanations are:
- the XLAS measurement saturation; according to the "Kipp & Zonen LAS and XLAS instruction manual", for a path length of 4km and a scintillometer height of 20 m, saturation measurement problem starts from H values of about 300 W.m$^{-2}$
- Uncertainties on the correction of stability using the universal stability function
- Potential inconsistencies between the area average MODIS radiative temperature and the air temperature measured locally at the meteorological station. |
| 79 | Figure 10. display several cases where there is a huge divergence in stress index particularly in April and July for both spacecraft. | These individual dates are discussed in the revised version. |
| 80 | Line 562: here the authors mentioned –uncertainties- but at no point in the paper we are discussing about this. As previously mentioned uncertainties come not only in EC and XLAS observations but also in the approximation used based on 100% closure in the energy balance. It is confusing and not clear definitively. | Please see authors' response to the general comment N°4. |
| 81 | Line 565-570: give some explanation but actually is a description of the time-series. Can you provide a real-actual-explanation about what is the physical processes underlining this divergences and convergences. | The discussion part relating to Figure 11 is improved in the revised version. |
| 82 | Same from 570 to 575 | Same as comment 80. |

| 83 | Line 588: is this the actual explanation of why there is such divergence or is this another speculation? | Same as comment 80. |
|----|----|----|
| 84 | Line 590-592: the error indicated here is extremely low now can you please indicate all- conditions in which this is valid and please circumvent this result to the specific interval of conditions in which this is actually valid. | Same as comment 80. |
| 85 | Figure 11. From where and how you got errorbars in blue trace? Figure caption is not clear. We need a accurate description of the contents in the figure. | Figure 11 caption is improved in the revised version.

Error bars for the SPARSE results show the minimum and the maximum daily evapotranspiration (ET) resulting from the three methods used to compute daily ET from instantaneous modeled ET at the time of Terra and Aqua overpasses: evaporative fraction, stress factor and residual methods, hence, six estimates of the daily modelled ET are produced. |
| 86 | Line 610: "valorize" I wonder what the authors wanted to indicate here? | This word is rather vague indeed, we precise the perspectives of this work, notably using a LSM applied at the field scale (Etchanchu et al., 2017) to analyse the scaling properties from the field to the footprint of the XLAS and the MODIS pixels similarly to the reference provided by Reviewer 2 (Bai et al., 2015). |
| 87 | SVAT seems not to have been defined earlier. | Rectified in the revised version. |

**Authors Response to Reviewer 1 comments**
**Second Review**

This is my second revision of the paper. The Authors improved the quality of the manuscript and all recommendations and questions of my first revision were filled. Therefore now the paper can be accepted for final publication. Only small corrections should be performed in the following figures:

Figure 2: Please use the same format of figure 6 (excepted the size), with Latitude and longitude in y and x axes and with the line of XLAS.

Figure 6: Please reduce the size.

Figure 10a: Use the same size and x/y ratio of Figure 10b.

*Response:*

Figures 2, 6 and 10 were improved according to the Reviewer 1's suggestions

**Authors Response to Reviewer 2 comments**
**Second Review**

**To the authors:**

Please, you need to understand that theory is one thing and hypothesis is a different scientific element. The so called MO similarity hypothesis is just a hypothesis not a THEORY. This hypothesis proposes that the set of non-linear equations you are using are valid only under homogenous surfaces and stationary flows. Please correct all instances in which MOST is mistaken and replace by MO hypothesis.

*Response:*

It is rectified in the revised version.
* * *
Going to your response #4 basically your answer is circumventing the question and what you are indicating is that the blending height will give you a safe interval to indicate the you are above all internal boundary layers that can be generated in the XLAS footprint and therefore you use either way the equations. But you had never demonstrated that this is the case. More importantly blending heights are defined including aerodynamic characteristics and no stratification is accounted for. So please make sure or provide example calculations to ensure you are above the blending height assuming this blending height is defined under conditions in which mechanical turbulence is larger than buoyancy driven turbulence.

*Response:*

We agree with the Reviewer; the blending height depends on aerodynamic characteristics and no stratification is accounted for. The blending height can be estimated with the equation of Mason (1988):

$$l_b = 2 * \left(\frac{u_*}{U}\right)^2 L_x$$

where $L_x$ the horizontal length scale, u* the friction velocity and U the wind speed.

We chose a horizontal length $L_x$ of about 200 m which is the maximum length of the cultivated plots (orchards or others) in our study area.

Below is displayed the histogram of $l_b$ computed with Mason's formula for half-hourly data when the stability index ($Z_{LAS}/L_O$) is comprised between -2 and 0 i.e. during unstable atmospheric conditions. It is noticeable that most measurements are performed under conditions for which the blending height is lower than 25 m. When $l_b$ is greater than 25 m, scintillometer measurements are removed from the dataset.

[Figure]
* * *
Response #8 is missing.

**8 : As an aside note the use of XLAS is not unique in this problem. A LAS can do 5 km max. Optical beam path and resolve the same situation. What is critical with using XLAS is beyond 5 km optical path.**

*Response:*

In our case, even for a path length less than 5 km, we cannot install a LAS instead of an XLAS. Based on the XLAS installation principle detailed in the "Kipp & Zonen LAS and XLAS instruction manual", the minimum installation height of a LAS as function of the path length and for different surface conditions is graphically explained and shows that for a path length of 4 km, the LAS height must be greater than 50 m to minimize measurement saturation. Operationally, in our study site, we cannot install a LAS at this height. Hence, the choice to install an XLAS.
* * *
**Detailed comments:**

**#21:** your answer still does not address the problem of distributed parameters. We agree on all those references you mentioned but you are still not given a convincing argument about using a model with concentrated parameters (R, V, I) over a problem that contains distributed parameters (roughness, soil temperatures, canopy temperatures, air temp, vapor pressure etc). Please give a reasoning to convince the reader that this model despite being a concentrate parameters model it represent distributed parameters of the land surface characteristics.

*Response:*

Scaling of the parameters in the flux/gradient relationship is indeed a crucial issue in applying energy balance models at low resolution. The gradient U is defined by the atmospheric

forcing representative of the Ben Salem area and the surface temperature acquired at the kilometre scale, with potential small but significant mismatch between both areas of representativity (the meteorological station vs. the radiometric FOV). This discrepancy has been mostly circumvented when applying those models at much larger scales than the perimeter scale for which meteorological data are available (cf. ALEXI/disALEXI frameworks coupling geostationary and heliosynchronous remote sensing data). On the other hand, scaling the resistance R is a much trickier issue. However, this issue has been tackled by Bahir et al. (2017) over a semi-arid agricultural land in Mexico. This work has shown that there were little differences between the total ET computed by TSEB at low resolution when applying the resistance formulation at that scale and the total ET summed from applying TSEB at high resolution over a homogeneous pixel. In our case, we have aggregated the roughness length from high or very high resolution data to the footprint or the pixel scale using the aggregation laws presented in Bahir et al. (2017).
* * *
**#36** here I refer to the numerical convergence of the set of MO + CN2 equations that you solve at each step when you input CN2, Tamb, Pamb and Bowen Ratio to deduce H_XLAS. Now this is called iterative method. The article mentioned is an advanced version of how to articulate all the equations MO + CN2 and obtain H_XLAS through an analytical computation. This has nothing to do with the G. Also what I mentioned about SNR of CN2 has to do with the convergence factor in the traditional iterative calculation method for H_XLAS retrieval. You are right about CN2 (z) this is explained in the article of Paulson that gives the sensitivity to terrain height since CN2 varies strongly and non-linearly with height. But my question here is about the numerical methodology you are using to integrate all equations each 30 min. based on the set of MO + CN2 equations.

*Response:*

In the iterative algorithm ($\beta$-closure method) used in the study, the input parameters are $Cn^2$, T, u, P, $z_0$, d, $z_{LAS}$, Rn, G. Then, the following variables (H, $L_{MO}$, $T_*$, $u_*$, $\beta$) are computed iteratively and are considered to be spatially representative of the scintillometer path. Below is the computation algorithm of the scintillometer fluxes.

```
%calculate H from scintillometer
for i=1: size(G,1)
    beeta(i)=0.01;
    L=9999;          %dummy value to avoid /0 in first iteration
    Lnew=-9999;      %dummy value to avoid /0 in first iteration
    disp(TIMESTAMP{i})
    count=0;
    %iteration to obtain H fluxes
    while (abs(L-Lnew) > 0.001) && count<=50
        L=Lnew;
        CT2=CT2nobow(i)*(1/(1+0.03/beeta(i)))^2;
        Tstar=-sqrt((CT2*z^(2/3))/(ct_1*(1-ct_2*z/L)^(-2/3)));
        xz=(1-16*z/L)^(1/4);
        xz0=(1-16*z0/L)^(1/4);
        psi_z=2*log((1+xz)/2)+log((1+xz^2)/2)-2*atan(xz)+pi/2;
        psi_z0=2*log((1+xz0)/2)+log((1+xz0^2)/2)-2*atan(xz0)+pi/2;
        ustar=(k*u(i))/(log(z/z0)-psi_z+psi_z0);
        Lnew=(ustar^2*T(i))/(g*k*Tstar);
        Tstar=-sqrt((CT2*z^(2/3))/(ct_1*(1-ct_2*z/Lnew)^(-2/3)));
        xz=(1-16*z/Lnew)^(1/4);
```

```
        xz0=(1-16*z0/Lnew)^(1/4);
        psi_z=2*log((1+xz)/2)+log((1+xz^2)/2)-2*atan(xz)+pi/2;
        psi_z0=2*log((1+xz0)/2)+log((1+xz0^2)/2)-2*atan(xz0)+pi/2;
        ustar=(k*u(i))/(log(z/z0)-psi_z+psi_z0);
        Hscint(i)=-rho*cp*ustar*Tstar;
        LEscint(i)=Rn(i)-G(i)-Hscint(i);
        beeta(i)=Hscint(i)/LEscint(i);
        count=count+1;
        if count==10
            Hscint(i)=NaN; LEscint(i)=NaN; ustar=NaN;
        end
    end
    Lstabilityscint(i)=(ustar^2*T(i))/(g*k*Tstar);
    ustarlas(i)=ustar;
end
```

With the β-closure method, the iterative algorithm fails (doesn't converge) in less than 1% of the cases. In addition, in most cases (~73%, see figure below), XLAS measurements are limited to unstable atmospheric conditions with a stability index $Z_{LAS}/L_O$ comprised between -2 and 0. Furthermore, Gruber and Fochesatto (2013) showed that the sensitivity functions for u* remain similar to those of Andreas (1992) while $Z_{LAS}/L_O$ <-1, but even if discrepancies increase for -2< $Z_{LAS}/L_O$ <-1, they remain acceptable.

[Figure]
* * *
**# 37:** please then clarify this in the text.

*Response:*

It is clarified in the revised version (section 3.1)
* * *
**#43**: this needs to be demonstrated. You should provide at least an example analysis that no advection is present. Remember that advective flows or the presence on site of submeso flows can erase the scale gap in which all micromet observations are based on. And that this condition will put atmospheric surface layer flows outside the conditions of MO Hypothesis in which all your deductions are based. Please take a look at papers from Foken as I mentioned before.

*Response:*

We agree with the reviewer's remark concerning the effect of advective flows or submeso flows (secondary circulations due to the heterogeneity of the surface which induce low-frequency vertical transport) and its impact on the use of Monin-Obukhov hypothesis. But, according to the previous answers, we showed that most of the time, measurements are well performed above the blending height in unstable conditions, where Monin-Obukhov hypothesis applies. The papers of Foken (Foken, 2008; Foken et al., 2010; Foken et al., 2006) you mentioned are focused on the difficulty to close the energy budget with Eddy covariance methods. With this micro-metereological set-up, you can not estimate the horizontal advection and the mean vertical mass flow. But, on the contrary, Foken et al. (2010) quote spatial averaging techniques such as scintillometry (Meijninger et al., 2006) or even airborne measurements (Mauder et al., 2007) as methods which seems more likely to be able to capture these secondary circulations and better close the energy budget.

[revised manuscript text omitted]

**4.2.1 • *Evaporative Fraction method**

Under clear sky days, The evaporative fraction (EF) self- -preservation was revised by several studies. Hoedjes et al. (2008) showed that EF is almost constant during daytime a valid assumption under dry conditions whereas it follows a concave up shape but no longer under wet conditions. Hence, (Hoedjes et al., 2008). For these conditions, assuming a constant EF underestimates actual EF depends strongly on soil moisture as well as canopy fraction cover, but, it is nearly unrelated to solar radiation and wind speed, as shown by therefore ET (Lhomme and Elguero, 1999). Indeed, according to Gentine et al. (2007).

[revised manuscript text omitted]

705    potential LE  to a combination of radiative temperatures($LE_{p-t-FP}$) as follows:

$$SF_{mod} = 1 - \frac{LE\_SPARSE_{t-FP}}{LE_{p-t-FP}}$$

 (25)

where $LE\_SPARSE_{t-FP}$ and $LE_{p-tFP}$ are the modeled latent heat fluxes in actual and potential conditions, respectively.

710    Furthermore, surface water stress factor derived from XLAS measurement, named $SF_{obs}$, at the time of Terra and Aqua overpass was computed as follows (Su, 2002):

$$SF_{obs} = \frac{H\_XLAS_t - H_{p-t-FP}}{H_{s-t-FP} - H_{p-t-FF}}$$

 (26)

[revised manuscript text omitted]